# On Trajectory Augmentations for Off-Policy Evaluation

**Ge Gao**[*]    **Qitong Gao**[†]    **Xi Yang**[‡]    **Song Ju**[*]    **Miroslav Pajic**[†]    **Min Chi**[*]

## Abstract

In the realm of reinforcement learning (RL), off-policy evaluation (OPE) holds a pivotal position, especially in high-stake human-centric scenarios such as e-learning and healthcare. Applying OPE to these domains is often challenging with scarce and underrepresentative offline training trajectories. Data augmentation has been a successful technique to enrich training data. However, directly employing existing data augmentation methods to OPE may not be feasible, due to the Markovian nature within the offline trajectories and the desire for generalizability across diverse target policies. In this work, we propose an offline trajectory augmentation approach, named **OAT**, to specifically facilitate OPE in human-involved scenarios. We propose sub-trajectory mining to extract potentially valuable sub-trajectories from offline data, and diversify the behaviors within those sub-trajectories by varying coverage of the state-action space. Our work was empirically evaluated in a wide array of environments, encompassing both simulated scenarios and real-world domains like robotic control, healthcare, and e-learning, where the training trajectories include varying levels of coverage of the state-action space. By enhancing the performance of a variety of OPE methods, our work offers a promising path forward for tackling OPE challenges in situations where human-centric data may be limited or underrepresentative.

## 1 Introduction

Off-policy evaluation (OPE) has been recognized as an important part of reinforcement learning (RL), especially for human-involved RLs (Wu et al., 2022; Gao et al., 2023a;b;c), in which evaluations of online policies can have high stakes (Levine et al., 2020). The objective of OPE is to evaluate target policies based on offline trajectories collected from behavioral policies different from the target ones. One major barrier often lies in the fact that the offline trajectories in human-involved tasks often only provide limited coverage of the entire state-action space (Chang et al., 2021; Schweighofer et al., 2021). This can be caused by *homogeneous behavioral* policies; for example, during clinical procedures, physicians need to follow certain standardized guidelines. However, a sub-optimal autonomous control agent (*e.g.*, surgical robots under training) may deviate from such guidelines, and thus result in trajectories where the state-action space may not be fully covered by the offline trajectories collected, which introduces great challenges for OPE, as illustrated in Figure 1. Therefore, to improve the OPE performance, it is essential to enrich the offline trajectories.

Data augmentation is a powerful tool for data enrichment by artificially generating new data points from existing data. It has shown effectiveness in facilitating learning more robust supervised and unsupervised models (Iwana & Uchida, 2021a; Xie et al., 2020). Specifically, generative methods such as variational autoencoder (VAE) have achieved superior performance in time-series augmentation (Yoon et al., 2019; Barak et al., 2022). However, an important characteristic of OPE training data is the Markovian nature, as the environments are usually formulated as a Markov decision process (MDP) (Thomas & Brunskill, 2016; Fu et al., 2021). As a result, prior works on time-series augmentation may not be directly applicable to MDP trajectory augmentation. Recently, though data augmentation methods have been extended to facilitate RL policy optimization, most existing

[*]North Carolina State University, USA. Emails: {ggao5, mchi}@ncsu.edu.

[†]Duke University, USA. Emails: {qitong.gao, miroslav.pajic}@duke.edu.

[‡]IBM Research, USA. Email: xi.yang@ibm.com.

works focus on enriching the state space, such as adding noise to input images to generate sufficient data and improve the generality of agents (Laskin et al., 2020a; Raileanu et al., 2021), but overlook the coverage of the joint state-action distribution over time. More importantly, the goal of data augmentation towards OPE is different from RL policy optimization. Data augmentation in RL generally aims to quickly facilitate identifying and learning from high-reward regions of the state-action space (Liu et al., 2021; Park et al., 2022). In contrast, the *evaluation* policies considered by OPE can be *heterogeneous* and lead to varied performance, *i.e.*, the policies to be evaluated by OPE do not necessarily perform well; therefore, it is equally important to allow the agent learning from trajectories resulted from high- and low-reward regions. As a result, OPE methods prefer training data that provides comprehensive coverage of the state-action space, including the trajectories resulting from low-performing and sub-optimal policies. To the best of our knowledge, there does not exist a method that augments historical trajectories specific to OPE.

In this paper, we propose a framework to facilitate **O**PE using **A**ugmented **T**rajectories (**OAT**). Specifically, motivated by the intrinsic nature that human-involved systems (HIS) are often provided limited coverage of the state-action space, while human may behave diversely when following different policies (Yang et al., 2020b; Wang et al., 2022), we propose potential sub-trajectories (PSTs) mining to identify sub-trajectories of historical trajectories whose state-action space is less covered but have great potential to enrich the space. Then a generative modeling framework is used to capture the dynamic underlying the PSTs and induce augmented sub-trajectories. Based on that, we design the fuse process by simultaneously taking the augmented sub-trajectories while maintaining the part of the states and actions associated with

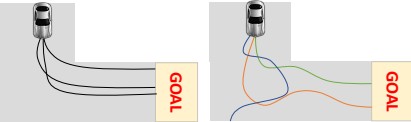

Figure 1: A conceptual illustration of the discrepancy between human demonstrations (left) versus the empirical trajectories resulted from a sub-optimal policy (right) to be evaluated by OPE. It can be observed that the autonomous agent may perform maneuvers unseen from the training (demonstration) trajectories, and thus can potentially hinder OPE's performance.

non-PSTs. The key contributions of this work are summarized as follows: (*i*) To the best of our knowledge, OAT is the first method augmenting historical trajectories to facilitate OPE in HIS. (*ii*) We conduct extensive experiments to validate OAT in a variety of simulation and real-world environments, including robotics, healthcare, and e-learning. (*iii*) The experimental results present that OAT can significantly facilitate OPE performance and outperform all data augmentation baselines.

## 2 OPE WITH AUGMENTED TRAJECTORIES (OAT)

We propose a framework to facilitate OPE with augmented trajectories (OAT). Specifically, we first introduce offline trajectories and OPE. Then we propose a sub-trajectory mining method that identifies the sub-trajectories of trajectories that have great potential to increase the offline trajectories' coverage over the state-action space, *i.e., potential sub-trajectories (PSTs)*. A generative modeling framework is used to capture the dynamics underlying the selected PSTs, followed by a fuse process that generates augmented trajectories with augmented sub-trajectories which will be used to train the OPE methods.

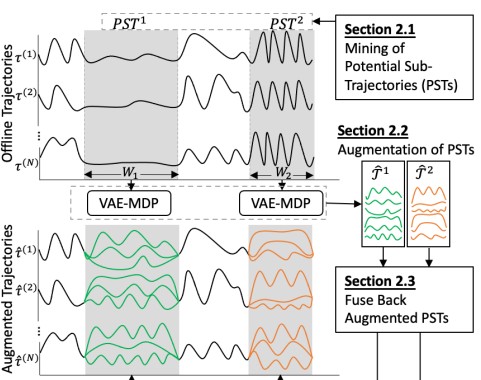

Figure 2: The illustration of OAT. It consists of three steps: (*i*) Mining of potential sub-trajectories (PSTs), where human behave similarly under behavioral policies at the grey-shaded area and may have more potential to enrich the coverage of its state-action space; (*ii*) VAE-MDP for augmenting PSTs; (*iii*) Fuse augmented PSTs back to their origins.

**Offline Trajectories.** We consider framing an agent's interaction with the environment over a sequence of decision-making steps as a Markov decision process (MDP), which is formulated as a 6-tuple $(\mathcal{S}, \mathcal{A}, \mathcal{P}, \mathcal{S}_0, r, \gamma)$. $\mathcal{S}$ is the state space. $\mathcal{A}$ is the action space. $\mathcal{P}$ defines transition dynamics from the current state and action to the next state. $\mathcal{S}_0$ defines the initial state distribution. $r$ is the reward function. $\gamma \in (0, 1]$ is discount factor. Episodes are of finite horizon $T$. At each time-step $t$, the agent observes the state $s_t \in \mathcal{S}$ of the environment, then

chooses an action $a_t \in \mathcal{A}$ following a policy $\pi$. The environment accordingly provides a reward $r_t = r(s_t, a_t)$, and the agent observes the next state $s_{t+1}$ determined by $\mathcal{P}$. $\tau^{(i)}$ is defined as a *trajectory* where $\tau^{(i)} = [..., (s_t, a_t, r_t, s'_t), ...]_{t=1}^T$.

**Offline Policy Evaluation (OPE).** The goal of OPE is to estimate the expected total return over the *evaluation (target)* policy $\pi$, $V^\pi = \mathbb{E}[\sum_{t=1}^T \gamma^{t-1} r_t | a_t \sim \pi]$, using set of historical trajectories $\mathcal{D}$ collected over a *behavioral* policy $\beta \neq \pi$. The historical trajectories $\mathcal{D} = \{..., \tau^{(i)}, ...\}_{i=1}^N$ consist of a set of $N$ trajectories.

## 2.1 Mining of Potential Sub-trajectories (PSTs)

The historical trajectories $\mathcal{D}$ collected from HIS are often provided with limited coverage of the state-action space, due to the intrinsic nature that human may follow homogeneous behavioral policies or specific guidelines when performing their professions (Yang et al., 2020b; Wang et al., 2022). For example, a surgeon could perform appendectomy in various ways across patients depending on each patient's specific condition; however, they may strictly follow similar steps at the beginning (*e.g.*, disinfection) and the end of surgeries (*e.g.*, stitching). Therefore, the resulting trajectories may lead to limited coverage for part of the state-action space representing similar scenarios. However, a sub-optimal autonomous agent, subject to be evaluated by OPE, may visit states unseen from the trajectories collected from the surgeon, *e.g.*, towards the beginning/end of the surgery. As a result, we consider augmenting the part of trajectories, *i.e.*, the PSTs, that are more likely to be insufficiently covered by the historical trajectories $\mathcal{D}$. Moreover, the downstream generative models, such as VAEs, do not necessarily need to reconstruct entire trajectories for long horizons and over limited samples which are the common limitations of data collected from HIS (Yacoby et al., 2020).

To identify the PSTs that are subject to be augmented, we introduce a three-step approach, *i.e.*, $(i)$ discrete representation mapping, followed by $(ii)$ determining support from discrete representations, where the support is used in step $(iii)$ to identify PSTs to be augmented.

**Step $(i)$ – Discrete Representation Mapping.** Trajectories collected from HIS can be complex, due to the unobservable underlying human mindset and high-dimensional state space (Mandel et al., 2014). Discrete representation mapping has been recognized as effectively providing abstractions from complex original data and helping capture homogeneous behaviors shared across trajectories (Yang et al., 2021). We assume that states $s_t \in \mathcal{S}$ can be mapped into $C$ clusters, where each $s_t$ is associated with a cluster from the set $\mathbf{K} = \{K_1, \dots, K_C\}$. After mapping, each state $s_{i,t}$ on trajectory $\tau^{(i)}$ is mapped to $K_{i,t} \in \mathbf{K}$.

**Step $(ii)$ – Determine Support from Discrete Representations.** We assume that each trajectory $\tau^{(i)}$ can be mapped to a corresponding temporal discrete sequence $K^{(i)} = [K_{i,1}, \dots, K_{i,T}] \subset \mathbb{Z}^T$, based on the state mapping, where $T$ is the horizon of the environment and $\mathbb{Z}$ is the set of integers. We also define $\mathcal{H} = \{..., K^{(i)}, ...\}_{i=1}^N$ which is the set of all temporal discrete sequences mapped from the set of original trajectories $\mathcal{D}$. We define $\delta_{\zeta,\zeta+W-1}^{(i)} = [K_{i,\zeta}, ..., K_{i,\zeta+W-1}]$ as a *temporal discrete sub-sequence (TDSS)* with length $W \in [1, T]$ of $K^{(i)}$, where $\zeta \in [1, T - W + 1]$, denoted as $\delta_{\zeta,\zeta+W-1}^{(i)} \sqsubseteq K^{(i)}$. Note that $C$ is generally greatly smaller than $T \times N$ as considered in discrete representation mapping in general (Hallac et al., 2017; Yang et al., 2021). Therefore, it is possible that a temporal discrete sub-sequence $\delta_{\zeta^i,\zeta^i+W-1}^{(i)}$ is "equal" to another temporal discrete sub-sequence $\delta_{\zeta^j,\zeta^j+W-1}^{(j)}$, such that $\delta_{\zeta^i,\zeta^i+W-1}^{(i)} = \delta_{\zeta^j,\zeta^j+W-1}^{(j)}$ if every $K_{i,\zeta^i} = K_{j,\zeta^j}$ given $K_{i,\zeta^i}, K_{j,\zeta^j} \in \mathbb{Z}$. Though $\zeta^i$ does not necessarily equals to $\zeta^j$, we omit the superscript for concise expression. Then, the *support (or frequency)* of any TDSS $\delta_{\zeta,\zeta+W-1}^{(i)}$ appears in $\mathcal{H}$ is the number of $K^{(i)}$ in $\mathcal{H}$ containing the TDSS, i.e.,

$$support_{\mathcal{H}}(\delta_{\zeta,\zeta+W-1}^{(i)}) = \sum_{j=1}^N \left[ \mathbb{1}(\delta_{\zeta,\zeta+W-1}^{(j)} \sqsubseteq K^{(j)}) \times \mathbb{1}(\delta_{\zeta,\zeta+W-1}^{(j)} = \delta_{\zeta,\zeta+W-1}^{(i)}) \right], \quad (1)$$

where $\mathbb{1}(\cdot)$ is the indicator function.

**Step $(iii)$ – Identify PSTs.** We denote $\varphi_{\zeta,\zeta+W-1}^{(i)} = [..., (\hat{s}_t^{(i)}, \hat{a}_t^{(i)}, \hat{r}_t^{(i)}, \hat{s}_t'^{(i)}), ...]_{t=\zeta}^{\zeta+W-1}$ as a sub-trajectory with length $W$ of $\tau^{(i)}$. Given the mapping from trajectory $\tau^{(i)}$ to temporal discrete sequence $K^{(i)}$ (introduced in the step above), we define that each sub-trajectory $\varphi_{\zeta,\zeta+W-1}^{(i)}$ can be

mapped to a corresponding TDSS $\delta^{(i)}_{\zeta,\zeta+W-1}$. Now we can identify the PSTs that will be used to train the generative model for reconstructing new sub-trajectories (*i.e.*, augmentation) in Section 2.2, following the definition below.

**Definition 2.1** (Potential Sub-Trajectory (PST)). Given historical trajectories $\mathcal{D}$ and a threshold $\xi$, a sub-trajectory $\varphi^{(i)}_{\zeta,\zeta+W-1}$ is considered as a potential sub-trajectory if the support of its mapped temporal discrete sub-sequence $\delta^{(i)}_{\zeta,\zeta+W-1}$ satisfies $support_{\mathcal{H}}(\delta^{(i)}_{\zeta,\zeta+W-1}) \geq \xi$.

An intuitive way to determine the threshold $\xi$ is ranking the TDSSs by their supports in descending order and picking the top $k$ ones. In this study, we iteratively select the top $k$ *distinct* TDSSs until the support of the set of $G$ selected TDSSs $\{\delta^g_{\zeta,\zeta+W-1}\}^1$, $g \in [1, G]$, *i.e.*, the number of $K^{(i)}$ in $\mathcal{H}$ containing any TDSS $\delta \in \{\delta^g_{\zeta,\zeta+W-1}\}$, is greater than or equal to $.99N$, or we early stop at $k = 5$.

Following the step above, a set of PSTs is determined for historical trajectories $\mathcal{D}$, from which we can obtain a set of $G$ *distinct* corresponding TDSSs $\{\delta^g_{\zeta,\zeta+W-1}\}$ mapped from the PSTs. Then we can obtain $G$ sets of PSTs, such that each set $\mathcal{T}^g = \{\varphi^{(i)}_{\zeta,\zeta+W-1}\}$, where all $\varphi^{(i)}_{\zeta,\zeta+W-1} \in \mathcal{T}^g$ satisfy that their corresponding $\delta^{(i)}_{\zeta,\zeta+W-1} = \delta^g_{\zeta,\zeta+W-1}$. Each set of PSTs may contain unique information captured from the original historical trajectories $\mathcal{D}$, as previous works have found that the PSTs in the same set, $\mathcal{T}^g$, are in general associated with similar temporal and cross-attributes correlations (Gao et al., 2021; 2022a).

## 2.2 AUGMENTING THE PSTS

In this section, we introduce how to adapt VAE to capture the MDP transitions, *i.e., VAE-MDP*, underlying each set of PSTs, $\mathcal{T}^g$, as well as reconstruct new PST samples that will be fused back with the original historical trajectories $\mathcal{D}$ for OPE methods to estimate the returns of evaluation (target) policies. The adaptation mainly consists of three parts: the latent prior, variational encoder, and generative decoder. Given a set of PSTs, $\mathcal{T}^g = \{\delta_{\zeta,\zeta+W-1}\}^2$, the formulation of VAE-MDP consists of three major components, *i.e.*, ($i$) the latent prior $p(z_\zeta) \sim \mathcal{N}(0, I)$ representing the distribution of the initial latent states (at the beginning of each PST in the set $\mathcal{T}^g$), where $I$ is the identity covariance matrix. ($ii$) the encoder $q_\omega(z_t|s_{t-1}, a_{t-1}, s_t)$ that encodes the MDP transitions into the latent space, and ($iii$) the decoders $p_\eta(z_t|z_{t-1}, a_{t-1})$, $p_\eta(s_t|z_t)$, $p_\eta(r_{t-1}|z_t)$ that reconstructs new PST samples. The detailed setup can be found in Appendix A.4, and the overall encoding and decoding processes are illustrated in Figure 3.

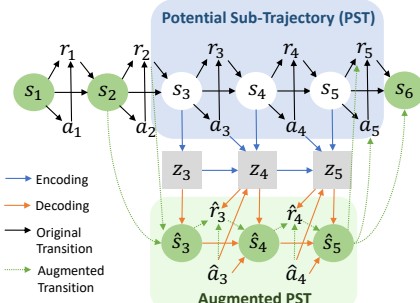

Figure 3: Illustration of VAE-MDP for Sub-trajectory augmentations. The PSTs are extracted by PSTs mining from original offline trajectories. Then VAE-MDP is employed to roll out new PSTs by reconstructing state-action space and inducing rewards. The augmented trajectory (colored in green) is formed by fusing back augmented PSTs to offline trajectories.

The training objective for VAE-MDP is to maximize the evidence lower bound (ELBO), which consists of the log-likelihood of reconstructing the states and rewards, and regularization of the approximated posterior, *i.e.*,

$$\mathcal{L} = -ELBO(\omega, \eta) = -\mathbb{E}_{q_\omega}\Big[\sum_{t=\zeta}^{\zeta+W-1}\log p_\eta(s_t|z_t) + \sum_{t=\zeta+1}^{\zeta+W-1}\log p_\eta(r_{t-1}|z_t) - KL\big(q_\omega(z_\zeta|s_\zeta)||p(z_\zeta)\big)$$
$$- \sum_{t=\zeta+1}^{\zeta+W-1}KL\big(q_\omega(z_t|z_{t-1}, a_{t-1}, s_t)||p_\eta(z_t|z_{t-1}, a_{t-1})\big)\Big].$$
$$(2)$$

The proof of Equation 2 are provided in Appendix A.5. Consequently, given a set of PSTs, $\mathcal{T}^g$, a VAE-MDP to the set can be trained to reconstruct a set of new PST samples, denoted as $\widehat{\mathcal{T}^g} =$

---

[1]From now we use superscript $^g$ to replace $^{(i)}$ for $\delta$'s, since there may exist multiple TDSSs that are equivalent.

[2]From now on we omit the superscripts of $\delta$ for conciseness.

$\{\hat{\varphi}^v_{\zeta,\zeta+W-1}\}$, $v \in [1, V]$, where $\hat{\varphi}^v_{\zeta,\zeta+W-1} = [..., (\hat{s}^v_t, \hat{a}^v_t, \hat{r}^v_t, \hat{s}'^v_t), ...]^{\zeta+W-1}_{t=\zeta}$ is a augmented PST and $V$ is the total number of augmented PST samples, generated from VAE-MDP, for the set $\mathcal{T}^g$.

## 2.3 Fuse Augmented PSTs back to Their Origins

With new augmented sub-trajectories rolled out by the VAE-MDP, we fuse them back to the original historical trajectories $\mathcal{D}$ for the OPE methods to leverage. This fusing process is designed to (*i*) provide enhanced coverage over the state-action space where the corresponding PSTs do not explicitly capture homogeneous behaviors, and still (*ii*) maintain the part of the covered state-action distribution associated with non-PSTs, since those may indicate object-specific information that is not shared across all trajectories, *e.g.*, the part of the surgical procedure specific to each patient, following from the surgery analogy above. Below we introduce how to fuse $\widehat{\mathcal{T}^g}$ with the original trajectories from $\mathcal{D}$. A graphical illustration of this step can be found in Figure 2.

Given a trajectory $\tau^{(i)} \in \mathcal{D}$, the $G$ sets of PSTs $\{\mathcal{T}^1, ..., \mathcal{T}^G\}$ mined from $\mathcal{D}$ following Section 2.1, and $G$ sets of augmented sub-trajectories $\{\widehat{\mathcal{T}^1}, ..., \widehat{\mathcal{T}^G}\}$ generated from $G$ corresponding VAE-MDPs following Section 2.2, an augmented trajectory $\hat{\tau}^{(i)}$ corresponding to $\tau^{(i)}$ can be obtained by $\hat{\tau}^{(i)} = \left[ \mathbb{1}(t \in [\zeta, \zeta + W - 1])(\hat{s}^v_t, \hat{a}^v_t, \hat{r}^v_t, \hat{s}'^v_t) \ \vee \ \mathbb{1}(t \notin [\zeta, \zeta + W - 1])(s^{(i)}_t, a^{(i)}_t, r^{(i)}_t, s'^{(i)}_t) \right]^T_{t=1}$; $(\hat{s}^v_t, \hat{a}^v_t, \hat{r}^v_t, \hat{s}'^v_t) \in \hat{\varphi}^v_{\zeta,\zeta+W-1}, \hat{\varphi}^v_{\zeta,\zeta+W-1} \in \widehat{\mathcal{T}^g}$. In this study, the $\hat{\varphi}^v_{\zeta,\zeta+W-1}$ is selected as the one whose state and action are the closest to the original state and action at step $\zeta + W - 1$. More details are provided in Appendix A.6.

## 3 Experiments

### 3.1 Setup

**Baselines.** We investigate a variety of augmentation methods as baselines, including (*i*) RL-oriented methods: **TDA** (Park et al., 2022) which originally incorporates with rewards learning by randomly extracting sub-trajectories from trajectories, we replace PST mining by TDA in OAT so that TDA can be used for OPE with augmentation; **permutation**, Gaussian **jittering**, and **scaling** have been broadly employed in RL (Laskin et al., 2020b; Liu et al., 2020; Raileanu et al., 2021); (*ii*) generative methods: **TimeGAN** (Yoon et al., 2019) and **VAE** (Barak et al., 2022); (*iii*) time-series methods: **SPAWNER** (Kamycki et al., 2019) and **DGW** (Iwana & Uchida, 2021b) that consider similarities across time series. We implement RL-augmentation methods strictly following original algorithms, and use open-sourced code provided by the authors for the generative and time-series augmentation methods. Since generative and time-series methods are not proposed towards trajectories, we treat trajectories as multivariate time series as their input.

**Ablations.** One ablation of our approach is to apply VAE-MDP to reconstruct *entire* trajectories as augmentations, *i.e.*, without PST mining (Section 2.1) and fusing back (Section 2.3). Moreover, TDA (Park et al., 2022) and VAE (Barak et al., 2022) can be considered as two ablations as well, since TDA isolates our PST mining from OAT and VAE augments entire trajectories following the vanilla VAE (Kingma & Welling, 2013), *i.e.*, without being adapted to the Markovian setting.

**OPE methods considered.** Outputs from all augmentation methods are fed into five OPE methods to compare the performance achieved with versus without augmentations. The OPE methods we consider include importance sampling (**IS**) (Precup, 2000), fitted Q-evaluation (**FQE**) (Le et al., 2019), distribution correction estimation (**DICE**) (Yang et al., 2020a), doubly robust (**DR**) (Thomas & Brunskill, 2016), and model-based (**MB**) (Zhang et al., 2020a). We use the open-sourced implementations provided by the Deep OPE (DOPE) benchmark (Fu et al., 2021).

**Standard validation metrics.** To validate OPE's performance (for both with and without augmentations), we use standard OPE metrics as introduced in the DOPE benchmark, which include absolute error, Spearman's rank correlation coefficient (Spearman, 1987), regret@1, and regret@5. Definitions of the metrics are described in Appendix B.3.

### 3.2 Environments

To evaluate our method, OAT, as well as the existing augmentation approaches for OPE, we use both simulated and real-world environments, spanning the domains of robotics, healthcare, and e-learning. The environments are human-involved which is generally challenging with highly limited

quantity of demonstrations containing underrepresented state space, due to homogeneous interventions when collecting the historical trajectories.

**Adroit.** Adriot (Rajeswaran et al., 2018) is a simulation environment with four synthetic real-world robotics tasks, where a simulated Shadow Hand robot is asked to hammer a nail (`hammer`), open a door (`door`), twirl a pen (`pen`), or pick up and move a ball (`relocate`). Each task contains three training datasets with different levels of human-involvements, including full demonstration data from human (`human`), induced data from a fine-tuned RL policy (`expert`), and mixing data with a 50-50 ratio of demonstration and induced data (`cloned`). We follow the experimental settings provided in Deep OPE benchmark, with 11 DAPG-based evaluation policies ranging from random to expert performance (Fu et al., 2021).

**Real-World Sepsis Treatment.** We investigate a challenging task in healthcare, sepsis treatments, which has raised broad attention in OPE (Namkoong et al., 2020; Nie et al., 2022). Specifically, the trajectories are taken from electronic health records containing 221,700 patient visits collected from a hospital over two years. The state space is constituted by 15 continuous sepsis-related clinical attributes that represent patients' health status, including heart rate, creatinine, etc. The cardinality of the action space is 4, *i.e.*, two binary treatment options over {`antibiotic_administration`, `oxygen_assistance`}. Given the four stages of sepsis defined by the clinicians (Delano & Ward, 2016), the rewards are set for each stage: infection ($\pm5$), inflammation ($\pm10$), organ failure ($\pm20$), and septic shock ($\pm50$). Negative rewards are given when a patient enters a worse stage, and positive rewards are given when the patient recovers to a better stage. The environment considers discrete time steps, with the horizon being 1160 steps. Five evaluation (target) policies are obtained by training Deep Q Networks (DQNs) (Mnih et al., 2015) respectively over different hyper-parameters. More details are provided in Appendix D.

**Real-World Intelligent Tutor.** Another important human-involved task for OPE is intelligent tutoring, where students interact with intelligent tutors, with the goal of improving students' engagements and learning outcomes. Such topics have been investigated in prior OPE works (Mandel et al., 2014; Nie et al., 2022). Specifically, we collect trajectories recorded from 1,307 students' interaction logs with an intelligent tutor, over seven semesters of an undergraduate course at an university. Since students' underlying learning states are unobservable (Mandel et al., 2014), we consult with domain experts who help defines the state space which is constituted by 142 attributes that could possibly capture students' learning status from their logs. The cardinality of the action space is 3, *i.e.,* on each problem, the tutor need to decide whether the student should `solve` the next problem by themselves, `study` a solution provided by the tutor, or `work together` with the tutor to solve on the problem. During the tutoring, each student is required to solve 12 problems, thus the horizon of the environment is considered as 12 discrete steps. Sparse rewards are obtained at the end of the tutoring, which are defined as students' normalized learning gains (Chi et al., 2011). We use the trajectories collected from six semesters as the training data, where the behavior policy follows an expert policy commonly used in e-learning (Zhou et al., 2019), and test on the upcoming semester. There are 4 evaluation policies, including three obtained by training DQNs over different hyper-parameters respectively, in addition to one expert policy. More details are provided in Appendix E.

## 3.3 RESULTS

**The need of PSTs mining.** To better understand the need of PSTs mining (Section 2.1) conceptually, we visualize the set of augmented trajectories produced by our method, against the original set of historical trajectories $\mathcal{D}$, over the `Maze2D-umaze` environment which is a toy navigation task requiring an agent to reach a fixed goal location (Fu et al., 2020). We uniformly down-sample a limited number (i.e., 250) of trajectories from the original dataset provided by D4RL (overall 12k trajectories), and use our method to augment this subset such that the total number of trajectories becomes ten times ($\times10$) larger. The visualization is shown in Figure 4. It can be observed

Figure 4: Visualization of trajectories in Maze2D-umaze. Left: the original 250 trajectories; Right: augmented data with ten times numbers of trajectories ($\times10$).

that there exist 3 sets of PSTs (as circled in the figure) that have significantly increased state space visitation after augmentation, benefiting from the PSTs mining methodology introduced in Section 2.1.

### 3.3.1 RESULTS OVER ADROIT

Figure 5 summarizes the averaged improvements across five OPE methods, over all four tasks (*i.e.*, `hammer`, `door`, `pen`, `relocate`) in Adroit `human`, quantified by the percentage increases over the four validation metrics achieved by the OPE methods evaluated over the augmented against the original datasets. Overall, our method significantly improves OPE methods in terms of all standard validation metrics, and achieves the best performance compared to all augmentation baselines. This illustrates the effectiveness and robustness of our proposed methods across environments and tasks. There is no clear winner among baselines, where VAE, TimeGAN, and scaling in general perform better in terms of MAE, DGW and scaling performs better in terms of rank correlation, permutation and jittering perform better in terms of regrest@5. More specifically, besides the fact that all methods can in general improve MAE, most baselines lead to negative effects in terms of the other three metrics. Detailed results are presented in Appendix C.1.

More importantly, it can be observed that the ablation baseline VAE-MDP is significantly outperformed by OAT across all metrics, which further justifies the importance of augmenting over the PSTs

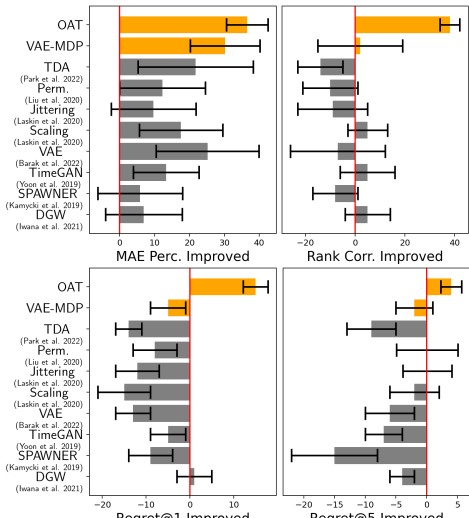

Figure 5: OPE improvement results averaging across **5 OPE methods** and **4 tasks** in Adroit `human` using each augmentation method. Top-left: Mean absolute error (MAE) percentage improved. Top-right: rank correlation improvements. Bottom-left & bottom-right: Regret@1 and @5 improvements, respectively.

instead of the entire horizon. It can be also observed that VAE-MDP in general outperforms the vanilla VAE without adaptation to the Markovian setting, illustrating the importance of the adaptation step introduced in Section 2.2. We also find that generative models achieve the best performance among the baselines over environments that have relatively shorter horizons (*e.g.*, `pen`), while their performance is diminished when horizons increased. That further indicates the advantage of PSTs mining that provides much shorter and representative sub-trajectories for generative learning.

### 3.3.2 RESULTS OVER REAL-WORLD HEALTHCARE AND E-LEARNING

Figure 6 presents the average MAE improvements across all OPE methods in e-learning (left), and improved rank correlation in healthcare (right). Complete results for empirical study and all validation metrics are provided in Appendix E. Regret@5 is not applicable to both environments, since the total number of evaluation policies are less than or equal to five.

Overall, our method can significantly improve OPE performance in terms of MAE, rank correlation, and regret@1 in both real-world environments. In both e-learning and healthcare, most augmentation baselines lead to neutral to negative

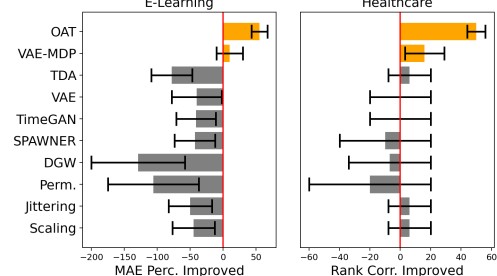

Figure 6: OPE improvement results averaging across 5 OPE methods in e-learning (left) and healthcare (right).

percentage improvements over the metrics considered, while OAT significantly improves OPE's performance over all baselines, with the ablation VAE-MDP attains the 2nd best performance. A reason for baselines perform worse in real-world environments than in simulations can be that real-world HIS are considered sophisticated, as the human mental states impact their behaviors implicitly. This further indicates the importance of extracting underlying information from historical trajectories $\mathcal{D}$, as did in OAT and VAE-MDP, as well as effectively enriching the coverage of state-action space to provide more comprehensive coverage for OPE methods to leverage, powered by the methodologies introduced in Section 2.

### 3.3.3 MORE DISCUSSIONS

We explore the following two major questions that are commonly involved in analyses over HIS.

**Will the level of human involvements affect trajectory augmentations for OPE?** As presented in Figure 7, we evaluate augmentation methods across the four tasks in Adroit environment with three different levels of human involvements (LoHI) sorted from the most to least, *i.e.,* human, cloned, and expert. The results show that our method achieves the best performance in terms of all validation metrics when humans are involved in data collection (i.e., human, cloned). The performance of our method is slightly attenuated (but still effective) when the LoHI decreased, while our ablation VAE-MDP leads MAE when the LoHI is 0% (i.e., expert). Though TDA is effective under the case when the LoHI is 0%, it still performs worse than OAT and consistently worse at other levels. Such a finding further confirms the effectiveness of PST mining. Moreover, most baselines are ineffective when the LoHI is below 50%. The reason is that the trajectories obtained from demonstrations often provide limited and/or biased coverage over the state-action space (Fu et al., 2021; Chang et al., 2021), thus any augmentation

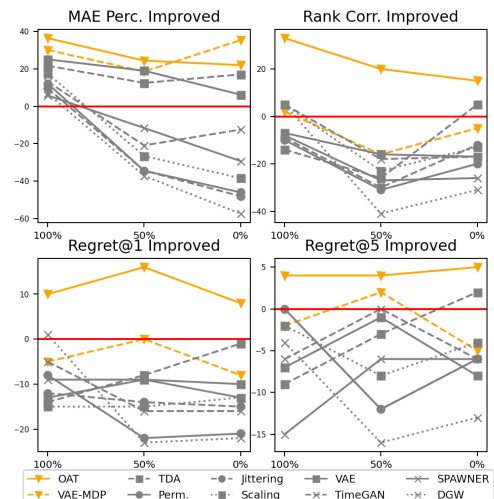

Figure 7: OPE improvement results with three human-involving levels, i.e., 100% (human), 50% (cloned), 0% (expert), averaging across 5 OPE methods and 4 tasks in Adroit.

methods that can potentially increase the coverage might be able to improve OPE's performance. In contrast, the historical trajectories induced from simulations using fine-tuned policies tend to result in better coverage over the state-action space in general (Fu et al., 2020), and the augmentation methods that do not consider the Markovian setting generate trajectories that could be less meaningful (*i.e.*, providing limited or even unrealistic information) to the OPE methods, making them less effective (*e.g.*, negative effects on OPE when LoHI is 0%). For example, in the realm of surgery, permutation can result in a trajectory with stitching happened before incision, which is unrealistic.

**Can trajectory augmentation facilitate OPE in terms of significance test?** OPE validation metrics generally focus on standard error metrics as proposed in (Fu et al., 2021), while domain experts emphasis statistical significance test for real-world HIS (Robertson & Kaptein, 2016; Zhou et al., 2022). For example, rank correlation summarizes the performance of relative rankings of a set of policies using averaged returns; in contrast, statistical significance tests can examine if the relationships being found are due to randomness. Moreover, they can be easier conveyed to and interpreted by domain experts (Guilford, 1950; Ju et al., 2019).

One key measurement for RL-induced policies is whether they significantly outperform the expert policy in HIS (Zhou et al., 2019; 2022). We conduct t-test over OPE estimations (with and without augmentations) obtained from bootstrapping as introduced in (Hao et al.,

Table 1: Statistical significance test at the level of $\rho < 0.05$ with bootstrapping on three RL-induced policy $\pi_1, \pi_2, \pi_3$ compared to expert policy $\pi_{expert}$ from real-world intelligent tutoring. The results that show significance are in bold.

| IS result | $\pi_1$ $t_p$ | $\pi_2$ $t_p$ | $\pi_3$ $t_p$ |
|---|---|---|---|
| No Aug. | **7.24**$_{.00}$ | **7.07**$_{.00}$ | **-4.48**$_{.00}$ |
| OAT | **3.10**$_{.00}$ | 1.33$_{.19}$ | 2.11$_{.06}$ |
| TDA | **10.14**$_{.00}$ | **5.82**$_{.00}$ | **-13.58**$_{.00}$ |
| Perm. | -1.78$_{.08}$ | -1.77$_{.08}$ | 1.77$_{.08}$ |
| Jittering | -1.89$_{.06}$ | -1.89$_{.06}$ | 1.90$_{.06}$ |
| Scaling | -1.33$_{.06}$ | -1.33$_{.06}$ | 1.33$_{.06}$ |
| VAE | -1.94$_{.06}$ | -1.92$_{.06}$ | 1.54$_{.13}$ |
| TimeGAN | 1.90$_{.06}$ | **-2.25**$_{.03}$ | **-2.25**$_{.03}$ |
| SPAWNER | -1.00$_{.32}$ | -1.00$_{.32}$ | 1.00$_{.32}$ |
| DGW | -1.43$_{.06}$ | -1.43$_{.16}$ | 1.43$_{.16}$ |
| Empirical result | **2.01**$_{.04}$ | 0.61$_{.54}$ | 0.20$_{.84}$ |

2021), and measure whether there is a significant difference between the mean value of OPE estimation for each RL-induced policy against the expert policy. Interestingly, the results show that IS performs the best among all 5 OPE methods we considered, in terms of all standard validation metrics in e-learning experiments, with and without augmentations using each augmentation method. This can be caused by the fact that the behavioral policies are intrinsically similar, where 3 out of the 4 policies (*i.e.*, $\pi_2, \pi_3, \pi_{expert}$) lead to similar returns (as shown in Appendix E.1) and the horizon (*i.e.*, $T$=12) is not much long, the unbiased nature of IS estimators could dominate it's high variance downside. Such characteristics of IS made it broadly used in short-horizon settings (Mandel et al.,

2014; Xie et al., 2019). The statistical significance results are summarized in Table 1. It can be observed that, without augmentation, IS estimates that all RL-induced policies performs significantly different from the expert policy. However, in empirical study, only $\pi_1$ performs significantly better than expert policy, while the other two, i.e., $\pi_2$ and $\pi_3$ not. And our proposed method is the only one that improves the IS estimation to be aligned with empirical results across all three policies, while the baselines improve estimation at most one policy. Therefore, the results indicate the effectiveness of our proposed method in terms of both standard OPE validation metrics and significance test.

## 4 RELATED WORKS

**OPE** A variety of contemporary OPE methods has been proposed, which can be mainly divided into three categories (Voloshin et al., 2021b): (i) Inverse propensity scoring (Precup, 2000; Doroudi et al., 2017), such as Importance Sampling (IS) (Doroudi et al., 2017). (ii) Direct methods that directly estimate the value functions of the evaluation policy (Nachum et al., 2019; Xie et al., 2019; Zhang et al., 2021; Yang et al., 2022; Gao et al., 2022c), including but not limited to model-based estimators (MB) (Paduraru, 2013; Zhang et al., 2021), value-based estimators (Munos et al., 2016; Le et al., 2019) such as Fitted Q Evaluation (FQE), and minimax estimators (Liu et al., 2018; Zhang et al., 2020b; Voloshin et al., 2021a) such as DualDICE (Yang et al., 2020a). (iii) Hybrid methods combine aspects of both inverse propensity scoring and direct methods (Jiang & Li, 2016; Thomas & Brunskill, 2016), such as DR (Jiang & Li, 2016). However, a major challenge of applying OPE to real-world is that many methods can perform unpleasant when human-collected data is highly limited as demonstrated in (Fu et al., 2020; Gao et al., 2023a; 2024). Therefore, augmentation can be an important way to facilitate OPE performance.

**Data Augmentation for RL** In RL, data augmentation has been recognized as effective to improve generalizability of agents over various tasks (Laskin et al., 2020b;a; Kostrikov et al., 2020; Liu et al., 2021; Raileanu et al., 2021; Joo et al., 2022; Goyal et al., 2022). For instance, automatic augmentation selection frameworks are proposed for actor-critic algorithms by regularizing the policy and value functions (Raileanu et al., 2021). However, most of the prior work only consider image input which may not capture temporal dependencies in trajectories. More importantly, the prior work is proposed towards RL policy optimization by learning from high-reward regions of state-action space, while OPE aims to generalize over evaluation policies that can be heterogeneous and lead to varied performance. To the best of our knowledge, no prior work has extensively investigated various prior augmentation methods in OPE, nor proposed augmentation towards offline trajectories to scaffold OPE in real-world domains. More comprehensive review of related works on OPE and data augmentations in general can be found in Appendix G.

## 5 CONCLUSION & LIMITATION

We have proposed OAT, which can capture the dynamics underlying human-involved environments from historical trajectories that provide limited coverage of the state-action space and induce effective augmented trajectories to facilitate OPE. This is achieved by mining potential sub-trajectories, as well as extending a generative modeling framework to capture dynamics under the potential sub-trajectories. We have validated OAT in both simulation and real-world environments, and the results have shown that OAT can generally improve OPE performance and outperform a variety of data augmentation methods. Latent-model-based models such as VAE have been commonly used for augmentation in offline RL, while they generally rarely come with theoretical error bounds provided (Hafner et al., 2020; Lee et al., 2020; Rybkin et al., 2021). Such a challenge also remains for many data augmentation methods (Zheng et al., 2023). However, once the trajectories are augmented, one can choose to use the downstream OPE methods which come with guarantees, such as DR and DICE. Moreover, prior works found that distribution shift could be a challenge for OPE (Wang et al., 2021; Fu et al., 2021). Though this is beyond the scope of this work, given we use existing OPE methods as backbones to process the augmented trajectories, a potential future work is coming up with new OPE methods that can resolve the distribution shift. We conduct extensive experiments to examine the proposed augmentation method, and the results demonstrating its effectiveness. OAT can be stand-alone to generate trajectories without any assumptions over target policies. And it can be utilized by built-on-top works such as policy optimization and representation learning.

## SOCIAL IMPACTS

All real-world educational and healthcare data employed in this paper were obtained anonymously through exempt IRB-approved protocols and were scored using established rubrics. No demographic data or class grades were collected. All data were shared within the research group under IRB, and were de-identified and automatically processed for labeling. This research seeks to remove societal harms that come from lower engagement and retention of students who need more personalized interventions and developing more robust medical interventions for patients.

## ACKNOWLEDGMENTS

This research was supported by the NSF Grants: Integrated Data-driven Technologies for Individualized Instruction in STEM Learning Environments (1726550), CAREER: Improving Adaptive Decision Making in Interactive Learning Environments (1651909), Generalizing Data-Driven Technologies to Improve Individualized STEM Instruction by Intelligent Tutors (2013502), CAREER: Foundations for Secure Control of Cyber-Physical Systems (1652544), and National AI Institute for Edge Computing Leveraging Next Generation Wireless Networks (2112562). This research was also sponsored in part by the AFOSR under award number FA9550-19-1-0169. We would also like to thank the anonymous reviewers for insightful comments that lead to improved paper presentations.

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

## LIST OF APPENDICES

## A    MORE DETAILS ON METHODOLOGY

### A.1    COMPLEXITY ANALYSIS

OAT consists of three main steps, including PSTs mining, PSTs augmentation, and fuse back; thus, the overall complexity depends on the specific techniques used in each step. In PSTs mining, the discrete representation mapping uses TICC, which takes $O(TNC)$ time to assign $TN$ states ($T$ is the horizon, $N$ is the number of trajectories) into $C$ clusters (Hallac et al., 2017). To identify PSTs, we use MG-FSM (Miliaraki et al., 2013) for sub-trajectory mining which has $O(T)$ time complexity. We found that providing complexity for PSTs augmentation could be challenging given that training VAE requires stochastic gradient descent algorithms with parameter tuning (e.g., for step size and network architecture). In fusing back, the time complexity is $O(N)$ if the same amount of augmented trajectories is added, and we are positive this can be shortened with parallel processing. Overall, as shown in Table 2, on average, OAT takes less training time of timeGAN and VAE with the same neural network architecture, respectively, which indicates OAT benefits more in augmenting on much shorter PSTs than the mining process.

For the cost of reaching effectiveness, the total number of TDSSs from which PSTs are selected is correlated to the length of the TDSSs which are capped at the horizon, i.e., the total number of TDSSs is capped at $C^T$. However, in Adroit-human, we noticed that the longest length of the PSTs found is 10 when $\xi$ is 10 (less than $N/2$), so $T$ seems like a generous upper bound. For example, in Adroit door-human, PSTs mining takes 224s over 6.7k data points.

Table 2: Average training time of OAT and generative-model-based methods in Adroit environment.

| Training time (s) | Pen | Door | Relocate | Hammer |
|---|---|---|---|---|
| OAT | 1662 | 688 | 734 | 1476 |
| VAE | 4247 | 2266 | 3184 | 3756 |
| TimeGAN | 1888 | 3027 | 4178 | 3716 |

### A.1.1    TRAINING TIME WITH VAE-MDP

We plot the training time of VAE-MDP vs length of PSTs across all 12 Adroit datasets in Figure 8, obtained from the same architecture of LSTMs and training settings as provided in Appendix B.2. From the Figure 8, we can observe that the training time of VAE-MDP is increased almost linearly with the length of PSTs. In experiments, the lengths of PSTs mined from most of the datasets (*i.e.*, 9 out of 12 datasets) are within 11, which only require less than 1 minute to train the VAE-MDP.

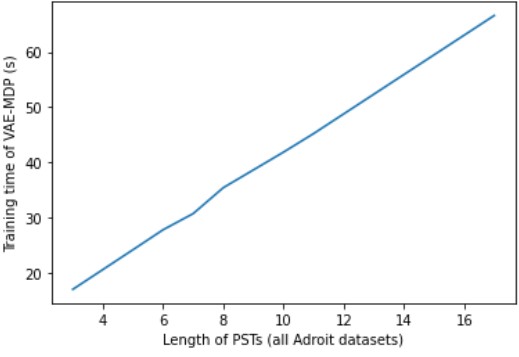

Figure 8: Plots of training time of VAE-MDP (in seconds) vs length of PSTs across all Adroit tasks.

## A.2 MORE ANALYSIS ON AUGMENTED TRAJECTORIES FROM OAT

### A.2.1 SCATTER PLOTS ON RETURNS

We present scatter plots plotting the true returns of each policy against the estimated returns. Each point on the plot represents one target policy.

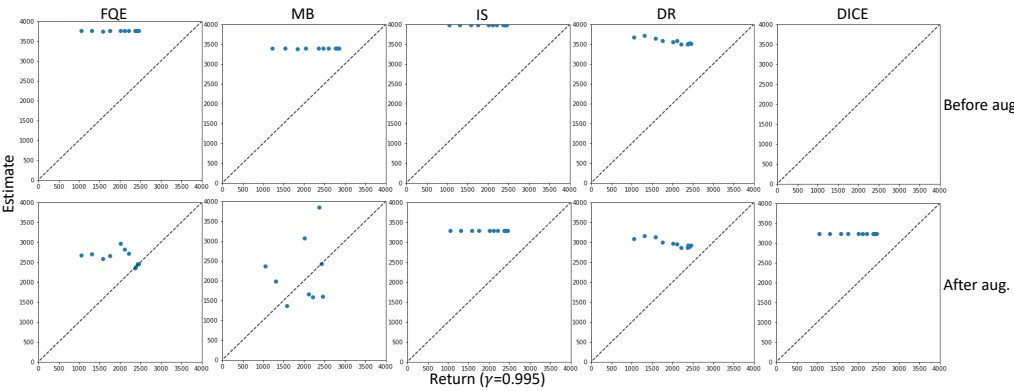

Figure 9: Scatter plots of estimate vs ground truth return before (top) and after (bottom) OAT augmentation in Adroit `pen-human` environment.

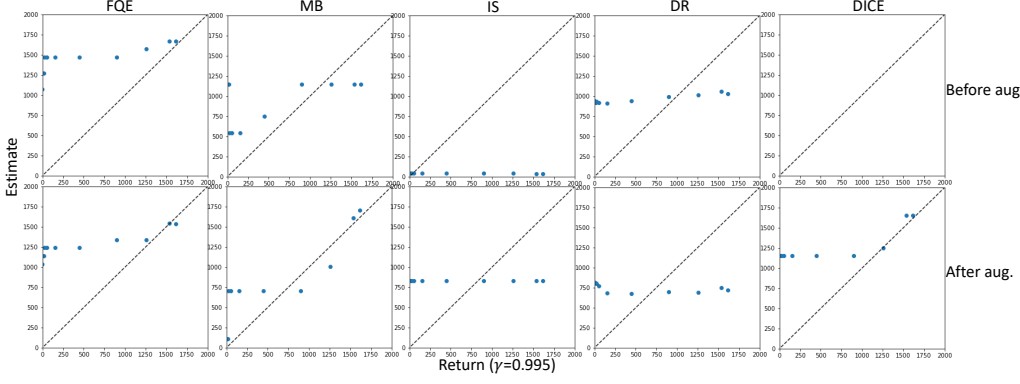

Figure 10: Scatter plots of estimate vs ground truth return before (top) and after (bottom) OAT augmentation in Adroit `relocate-human` environment.

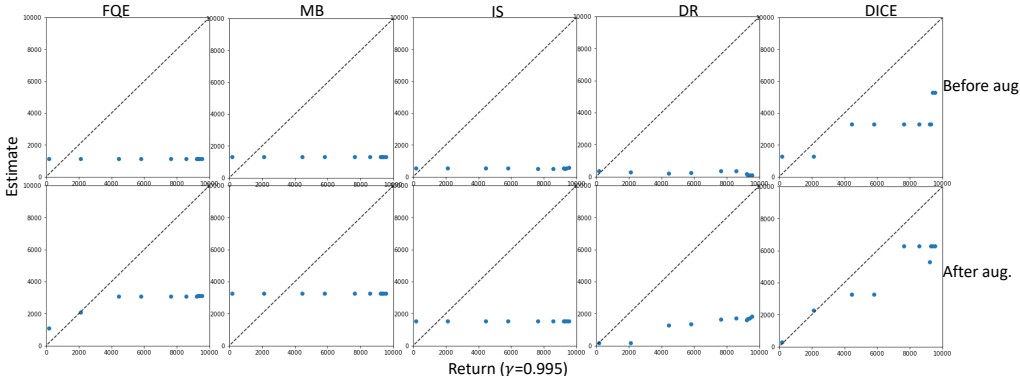

Figure 11: Scatter plots of estimate vs ground truth return before (top) and after (bottom) OAT augmentation in Adroit `hammer-human` environment.

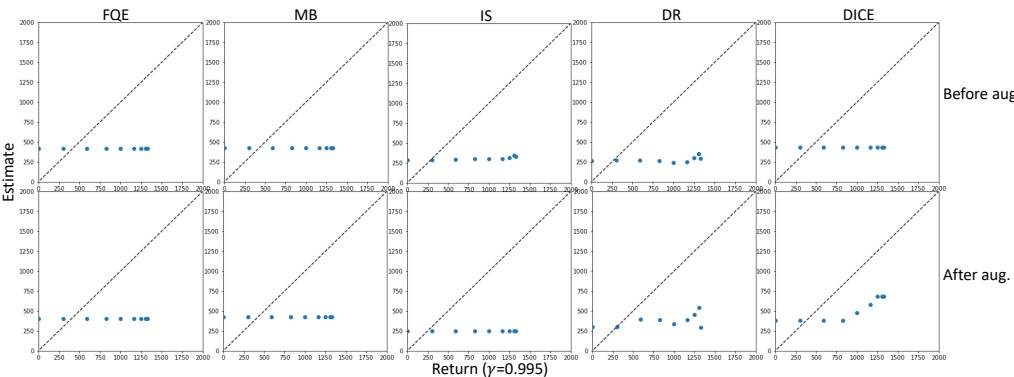

Figure 12: Scatter plots of estimate vs ground truth return before (top) and after (bottom) OAT augmentation in Adroit `door-human` environment.

### A.3 DETAILS OF DISCRETE REPRESENTATION MAPPING

In this work, we leverage Toeplitz inverse covariance-based clustering (TICC) (Hallac et al., 2017) to map states $s_t \in \mathcal{S}$ into $C$ clusters, where each $s_t$ is associated with a cluster from the set $\mathbf{K} = \{K_1, \ldots, K_C\}$. The states mapped to the same cluster can be considered sharing graphical connectivity structure of both temporal and cross-attributes information captured by TICC. There are variations of TICC targeting specific characteristics of data. Specifically, we used MT-TICC (Yang et al., 2021) which is proposed towards time-awareness and multi-trajectories. The hyperparameter $C$ can be determined by calculating silhouette score (Hallac et al., 2017).

#### A.3.1 TICC PROBLEM

Each cluster $c \in [1, C]$ is defined as a Markov random field (Rue & Held, 2005), or correlation network, captured by its Gaussian inverse covariance matrix $\Sigma_c^{-1} \in \mathbb{R}^{m \times m}$, where $m$ is the dimension of state space. We also define the set of clusters $\mathbf{K} = \{K_1, \ldots, K_C\} \subset \mathbb{R}$ as well as the set of inverse covariance matrices $\mathbf{\Sigma}^{-1} = \{\Sigma_1^{-1}, \ldots, \Sigma_C^{-1}\}$. Then the objective is set to be

$$\max_{\mathbf{\Sigma}^{-1}, \mathbf{K}} \sum_{c=1}^{C} \Big[ \sum_{s_t^{(i)} \in K_c} \big( \mathcal{L}(s_t^{(i)}; \Sigma_c^{-1}) - \epsilon \mathbb{1}\{s_{t-1}^{(i)} \notin K_c\}\big)\Big], \tag{3}$$

where the first term defines the log-likelihood of $s_t^{(i)}$ coming from $K_c$ as $\mathcal{L}(s_t^{(i)}; \Sigma_c^{-1}) = -\frac{1}{2}(s_t^{(i)} - \mu_c k)^T \Sigma_c^{-1}(s_t^{(i)} - \mu_c) + \frac{1}{2}\log\det\Sigma_c^{-1} - \frac{n}{2}\log(2\pi)$ with $\mu_c$ being the empirical mean of cluster $K_c$, the second term $\mathbb{1}\{s_{t-1}^{(i)} \notin K_c\}$ penalizes the adjacent events that are not assigned to the same cluster and $\epsilon$ is a constant balancing off the scale of the two terms. This optimization problem can be solved using the expectation-maximization family of algorithms by updating $\Sigma^{-1}$ and $\mathbf{K}$ alternatively Hallac et al. (2017).

### A.4 DETAILED FORMULATION OF THE VAE-MDP

**The latent prior** $p(z_\zeta) \sim \mathcal{N}(0, I)$ representing the distribution of the initial latent states (at the beginning of each PST in the set $\mathcal{T}^g$), where $I$ is the identity covariance matrix.

**The encoder** $q_\omega(z_t|s_{t-1}, a_{t-1}, s_t)$ is used to approximate the posterior distribution $p_\eta(z_t|s_{t-1}, a_{t-1}, s_t) = \frac{p_\eta(z_{t-1}, a_{t-1}, z_t, s_t)}{\int_{z_t \in \mathcal{Z}} p(z_{t-1}, a_{t-1}, z_t, s_t)dz_t}$, where $\mathcal{Z} \subset \mathbb{R}^m$ and $m$ is the dimension. Given that $q_\omega(z_{\zeta:\zeta+W-1}|s_{\zeta:\zeta+W-1}, a_{\zeta:\zeta+W-2}) = q_\omega(z_\zeta|s_\zeta)\prod_{t=\zeta+1}^{\zeta+W-1} q_\omega(z_t|z_{t-1}, a_{t-1}, s_t)$, both distributions $q_\omega(z_\zeta|s_\zeta)$ and $q_\omega(z_t|z_{t-1}, a_{t-1}, s_t)$ follow diagonal Gaussian, where mean and diagonal covariance are determined by multi-layer perceptrons (MLPs) and long short-term memory (LSTM), with neural network weights $\omega$. Thus, one can infer $z_\zeta^\omega \sim q_\omega(z_\zeta|s_\zeta)$, $z_t^\omega \sim q_\omega(z_t|h_t^\omega)$, with $h_t^\omega = f_\omega(h_{t-1}^\omega, z_{t-1}^\omega, a_{t-1}, s_t)$ where $f_\omega$ represents LSTM layer and $h_t^\omega$ represents LSTM recurrent hidden state.

**The decoder** $p_\eta(z_t, s_t, r_{t-1}|z_{t-1}, a_{t-1})$ is used to sample new trajectories. Given $p_\eta(z_{\zeta+1:\zeta+W-1}, s_{\zeta:\zeta+W-1}, r_{\zeta:\zeta+W-2}|z_\zeta, \beta) = \prod_{t=\zeta}^{\zeta+W-1} p_\eta(s_t|z_t)\prod_{t=\zeta+1}^{T} p_\eta(z_t|z_{t-1}, a_{t-1})p_\eta(r_{t-1}|z_t)$, where $a_t$'s are determined following the behavioral policy $\beta$, distributions $p_\eta(s_t|z_t)$ and $p_\eta(r_{t-1}|z_t)$ follow diagonal Gaussian with mean and covariance determined by MLPs and $p_\eta(z_t|z_{t-1}, a_{t-1})$ follows diagonal Gaussian with mean and covariance determined by LSTM.

Thus, the generative process can be formulated as, *i.e.*, at initialization, $z_\zeta^\eta \sim p(z_\zeta)$, $s_\zeta^\eta \sim p_\eta(s_\zeta|z_\zeta^\eta)$, $a_\zeta \sim \beta(a_\zeta|s_\zeta^\eta)$; followed by $z_t^\eta \sim p_\eta(\tilde{h}_t^\eta)$, $r_{t-1}^\eta \sim p_\eta(r_{t-1}|z_t^\eta)$, $s_t^\eta \sim p_\eta(s_t|z_t^\eta)$, $a_t \sim \beta(a_t|s_t^\eta)$, with $\tilde{h}_t^\eta = g_\eta[f_\eta(h_{t-1}^\eta, z_{t-1}^\eta, a_{t-1})]$ where $g_\eta$ represents an MLP.

## A.5 PROOF OF EQUATION 2

The derivation of the evidence lower bound (ELBO) for the joint log-likelihood distribution can be found below.

$$\log p_\eta(s_{\zeta:\zeta+W-1}, r_{\zeta:\zeta+W-2}) \tag{4}$$

$$= \log \int_{z_{\zeta+1:\zeta+W-1} \in \mathcal{Z}} p_\eta(s_{\zeta:\zeta+W-1}, z_{\zeta+1:\zeta+W-1}, r_{\zeta:\zeta+W-2}) dz \tag{5}$$

$$= \log \int_{z_{\zeta+1:\zeta+W-1} \in \mathcal{Z}} \frac{p_\eta(s_{\zeta:\zeta+W-1}, z_{\zeta+1:\zeta+W-1}, r_{\zeta:\zeta+W-2})}{q_\omega(z_{\zeta:\zeta+W-1}|s_{\zeta:\zeta+W-1}, a_{\zeta:\zeta+W-2})} q_\omega(z_{\zeta:\zeta+W-1}|s_{\zeta:\zeta+W-1}, a_{\zeta:\zeta+W-2}) dz \tag{6}$$

$$\overset{Jensen's\ inequality}{\geq} \mathbb{E}_{q_\omega}[\log p(z_\zeta) + \log p_\eta(s_{\zeta:\zeta+W-1}, z_{\zeta+1:\zeta+W-1}, r_{\zeta:\zeta+W-2}|z_\zeta) - \log q_\omega(z_{\zeta:\zeta+W-1}|s_{\zeta:\zeta+W-1}, a_{\zeta:\zeta+W-2})] \tag{7}$$

$$= \mathbb{E}_{q_\omega}\Big[ \log p(z_\zeta) + \log p_\eta(s_\zeta|z_\zeta) + \sum_{t=\zeta}^{\zeta+W-1} \log p_\eta(s_t, z_t, r_{t-1}|z_{t-1}, a_{t-1})$$
$$- \log q_\omega(z_\zeta|s_\zeta) - \sum_{t=\zeta+1}^{\zeta+W-1} \log q_\omega(z_t|z_{t-1}, a_{t-1}, s_t) \Big] \tag{8}$$

$$= \mathbb{E}_{q_\omega}\Big[ \log p(z_\zeta) - \log q_\omega(z_\zeta|s_\zeta) + \log p_\eta(s_\zeta|z_\zeta) + \sum_{t=\zeta+1}^{\zeta+W-1} \log \big(p_\eta(s_t|z_t)p_\eta(r_{t-1}|z_t)p_\eta(z_t|z_{t-1}, a_{t-1})\big)$$
$$- \sum_{t=\zeta+1}^{\zeta+W-1} \log q_\omega(z_t|z_{t-1}, a_{t-1}, s_t) \Big] \tag{9}$$

$$= \mathbb{E}_{q_\omega}\Big[ \sum_{t=\zeta}^{\zeta+W-1} \log p_\eta(s_t|z_t) + \sum_{t=\zeta+1}^{\zeta+W-1} \log p_\eta(r_{t-1}|z_t)$$
$$- KL\big(q_\omega(z_\zeta|s_\zeta)||p(z_\zeta)\big) - \sum_{t=\zeta+1}^{\zeta+W-1} KL\big(q_\omega(z_t|z_{t-1}, a_{t-1}, s_t)||p_\eta(z_t|z_{t-1}, a_{t-1})\big) \Big]. \tag{10}$$

## A.6 MORE DETAILS OF FUSING BACK STEP

We have ensured that the transition from the original trajectory to the beginning of the augmented PSTs are smoothed, by letting the generation of $\hat{s}_\zeta$, the initial state in the PST (e.g., $\hat{s}_3$ in Figure 3), to be conditioned on $s_{\zeta-1}$ which is the last state in the original trajectory (e.g., $\hat{s}_2$ in Figure 3), which equivalently set $s_{\zeta-1}$ as the underlying initial state for the generated PST. To smooth the end of PSTs, we select the augmented PSTs with the states and actions that have the least distance to the original states and actions, at step $\zeta + W - 1$. In experiments, we use the Euclidean distance as the measure.

## A.7 OAT WITH NON-TEMPORALLY-ALIGNED TRAJECTORIES

In the main context, for notation conciseness, we use horizon $T$ to denote the length of all trajectories for notational simplicity, since our work is the first one introducing sub-trajectory mining and augmentation to OPE. Note that OAT can work with non-temporally-aligned trajectories, *i.e.*, trajectories with different lengths and varied start and end times. In the PSTs mining step, the length $W$ is flexible and the TDSSs can be generated with a small length (e.g., 2) and expanded recursively until reaching a maximum length (e.g., $T$). Since our goal is to extract PSTs whose TDSSs are shared across trajectories with large supports, a threshold $\xi$ is used to bound the supports of PSTs so that the PSTs with the greatest supports can be always extracted. Moreover, in our experiment, the Adroit-human environments contain temporally non-aligned trajectories (e.g., length of trajectories from `door` and `relocate`, varied from 223 to 300, and 297 to 527, respectively). More environmental details are provided in Appendix C. And Appendix C.2 shows the the length of trajectories, and start and end time of PSTs found on each trajectory can be varied.

# B EXPERIMENTAL SETUP

## B.1 TRAINING RESOURCES

We implement the proposed method in Python. Training of our method and baselines are supported by four NVIDIA TITAN Xp 12GB, three NVIDIA Quadro RTX 6000 24GB, and four NVIDIA RTX A5000 24GB GPUs.

## B.2 IMPLEMENTATION DETAILS & HYPER-PARAMETERS

The cluster number for discrete representation mapping can be determined by silhouette score using training data following (Hallac et al., 2017), we perform search among $[10, 20]$ for $C$ in all datasets and the one with the highest silhouette score is selected. In our experiments, $C = 18, 10, 19, 16$ for {pen, door, relocate, hammer}-human, respectively; $C = 20, 10, 10, 10$ for {pen, door, relocate, hammer}-cloned, respectively; $C = 11, 10, 10, 10$ for {pen, door, relocate, hammer}-expert, respectively; $C = 14, 17$ for e-learning and healthcare, respectively.

The experimental results are obtained with selecting the PSTs using the threshold $\xi \in [2, N]$ at the top 1, *i.e.*, we use the PST with the highest support of its corresponding TDSS, for easier investigation of the PSTs mining and comparison to other augmentation baselines such as TDA, and present straightforward and general effects of our method. The percentage supports of the selected PSTs, *i.e.*, $support(\cdot)/N$, are all $\geq 82\%$ across all datasets and all experimental environments, especially can cover $100\%$ trajectories in all Adroit human tasks, which may further indicates the effectiveness of PSTs mining. We choose the neural network architectures as follows.

For the components involving LSTMs, which include $q_\omega(z_t|z_{t-1}, a_{t-1}, s_t)$ and $p_\eta(z_t|z_{t-1}, a_{t-1})$, their architecture include one LSTM layer with 64 nodes, followed by a dense layer with 64 nodes. All other components do not have LSTM layers involved, so they are constituted by a neural network with 2 dense layers, with 128 and 64 nodes respectively. The output layers that determine the mean and diagonal covariance of diagonal Gaussian distributions use linear and softplus activations, respectively. The ones that determine the mean of Bernoulli distributions (*e.g.*, for capturing early termination of episodes) are configured to use sigmoid activations. For training OAT and its ablation VAE-MDP, maximum number of iteration is set to 100 and minibatch size set to 4 (given the small numbers of trajectories, *i.e.*, 25 for each task) in Adroit, and 1,000 and 64 for real-world healthcare and e-learning, respectively. Adam optimizer is used to perform gradient descent. To determine the learning rate, we perform grid search among $\{1e-4, 3e-3, 3e-4, 5e-4, 7e-4\}$. Exponential decay is applied to the learning rate, which decays the learning rate by 0.997 every iteration. For OPE, the model-based methods are evaluated by directly interacting with each target policy for 50 episodes, and the mean of discounted total returns ($\gamma = 0.995$ for Adroit, $\gamma = 0.99$ for Healthcare, $\gamma = 0.9$ for e-learning) over all episodes is used as estimated performance for the policy.

## B.3 EVALUATION METRICS

*Absolute error* The absolute error is defined as the difference between the actual value and estimated value of a policy:

$$AE = |V^\pi - \hat{V}^\pi| \tag{11}$$

where $V^\pi$ represents the actual value of the policy $\pi$, and $\hat{V}^\pi$ represents the estimated value of $\pi$.

*Regret@1* Regret@1 is the (normalized) difference between the value of the actual best policy, and the actual value of the best policy chosen by estimated values. It can be defined as:

$$R1 = (\max_{i \in 1:P} V_i^\pi - \max_{j \in \text{best}(1:P)} V_j^\pi) / \max_{i \in 1:P} V_i^\pi \tag{12}$$

where $\text{best}(1:P)$ denotes the index of the best policy over the set of $P$ policies as measured by estimated values $\hat{V}^\pi$.

Table 3: Summary of Adroit.

| Task | State Dim. | Action Dim. | Early Term. | Continuous Ctrl. | Dataset Size |
|---|---|---|---|---|---|
| Pen-human | 45 | 24 | Yes | Yes | 5000 |
| Door-human | 39 | 28 | No | Yes | 6729 |
| Hammer-human | 46 | 26 | No | Yes | 11310 |
| Relocate-human | 39 | 30 | No | Yes | 9942 |
| Pen-cloned | 45 | 24 | Yes | Yes | $5*10^5$ |
| Door-cloned | 39 | 28 | No | Yes | $10^6$ |
| Hammer-cloned | 46 | 26 | No | Yes | $10^6$ |
| Relocate-cloned | 39 | 30 | No | Yes | $10^6$ |
| Pen-expert | 45 | 24 | Yes | Yes | $5*10^5$ |
| Door-expert | 39 | 28 | No | Yes | $10^6$ |
| Hammer-expert | 46 | 26 | No | Yes | $10^6$ |
| Relocate-expert | 39 | 30 | No | Yes | $10^6$ |

*Rank correlation* Rank correlation measures the Spearman's rank correlation coefficient between the ordinal rankings of the estimated values and actual values across policies:

$$\rho = \frac{Cov(\text{rank}(V_{1:P}^\pi), \text{rank}(\hat{V}_{1:P}^\pi))}{\sigma(\text{rank}(V_{1:P}^\pi))\sigma(\text{rank}(\hat{V}_{1:P}^\pi))} \tag{13}$$

where $\text{rank}(V_{1:P}^\pi)$ denotes the ordinal rankings of the actual values across policies, and $\text{rank}(\hat{V}_{1:P}^\pi)$ denotes the ordinal rankings of the estimated values across policies.

## C  ADROIT

As shown in Figure 13, Adriot (Rajeswaran et al., 2018) is a simulation environment with four synthetic real-world robotics tasks, where a 24-DoF simulated Shadow Hand robot is asked to hammer a nail (hammer), open a door (door), twirl a pen (pen), or pick up and move a ball (relocate). Each task contains three training datasets with different levels of human-involvements, including full demonstration data from human (human), induced data from a fine-tuned RL policy (expert), and mixing data with a 50-50 ratio of demonstration and induced data (cloned). Task properties are provided in Table 3.

### C.1  DETAILED RESULTS

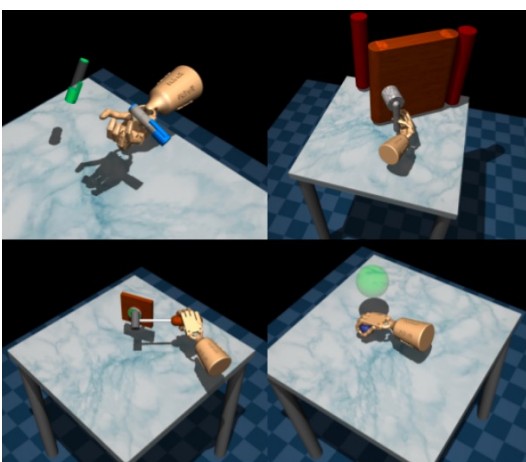

Figure 13: Four tasks in Adroit environment.

Table 4: Summary results averaging across five OPE methods without augmentation and with each augmentation method in Adroit `human`.

| | Pen | | | | Relocate | | | |
|---|---|---|---|---|---|---|---|---|
| | MAE | Rank Corr. | Regret@1 | Regret@5 | MAE | Rank Corr. | Regret@1 | Regret@5 |
| NoAug. | 3014 | -0.104 | 0.184 | 0.03 | 1956.4 | 0.204 | 0.434 | 0.298 |
| OAT | 1042.2 | 0.276 | 0.166 | 0.026 | 537 | 0.36 | 0.42 | 0.158 |
| VAE-MDP | 1527.8 | 0.226 | 0.204 | 0.02 | 430.8 | 0.142 | 0.53 | 0.256 |
| VAE | 1302.4 | -0.03 | 0.334 | 0.062 | 834.8 | 0.04 | 0.654 | 0.484 |
| TimeGAN | 1538.2 | 0.006 | 0.216 | 0.022 | 1209 | 0.408 | 0.604 | 0.34 |
| SPAWNER | 1817.8 | -0.192 | 0.218 | 0.144 | 1560.6 | 0.338 | 0.436 | 0.272 |
| DGW | 1578 | -0.028 | 0.292 | 0.054 | 1226.8 | 0.164 | 0.434 | 0.294 |
| Permutation | 1548.6 | -0.132 | 0.27 | 0.076 | 1628 | 0.338 | 0.51 | 0.152 |
| Jittering | 1632.8 | -0.096 | 0.202 | 0.076 | 1407.4 | 0.038 | 0.574 | 0.168 |
| Scaling | 1308.2 | -0.02 | 0.382 | 0.076 | 1462.8 | 0.244 | 0.72 | 0.212 |
| TDA | 1030.4 | -0.116 | 0.262 | 0.06 | 832.2 | 0.182 | 0.608 | 0.496 |

| | Hammer | | | | Door | | | |
|---|---|---|---|---|---|---|---|---|
| | MAE | Rank Corr. | Regret@1 | Regret@5 | MAE | Rank Corr. | Regret@1 | Regret@5 |
| NoAug. | 5266 | 0.344 | 0.34 | 0.058 | 603.8 | 0.14 | 0.274 | 0.004 |
| OAT | 3187.8 | 0.608 | 0.084 | 0.018 | 477.8 | 0.676 | 0.152 | 0.026 |
| VAE-MDP | 3418.6 | 0.02 | 0.454 | 0.126 | 497.2 | 0.336 | 0.224 | 0.052 |
| VAE | 3733.2 | -0.198 | 0.47 | 0.104 | 642.8 | 0.482 | 0.288 | 0.046 |
| TimeGAN | 4681 | 0.262 | 0.34 | 0.24 | 507 | 0.392 | 0.27 | 0.03 |
| SPAWNER | 4244.8 | -0.156 | 0.55 | 0.37 | 687.6 | 0.146 | 0.37 | 0.186 |
| DGW | 5238.8 | 0.242 | 0.296 | 0.144 | 578.2 | 0.342 | 0.134 | 0.052 |
| Permutation | 4103.2 | -0.202 | 0.534 | 0.076 | 583.4 | 0.264 | 0.236 | 0.088 |
| Jittering | 4256.4 | 0.004 | 0.452 | 0.102 | 700.2 | 0.268 | 0.342 | 0.056 |
| Scaling | 3832.4 | 0.166 | 0.404 | 0.102 | 580.8 | 0.222 | 0.326 | 0.078 |
| TDA | 3448 | -0.298 | 0.56 | 0.078 | 511.2 | 0.118 | 0.376 | 0.112 |

Table 5: MAE results of OPE without and with each augmentation method in Adroit `human` environment. Results are obtained by averaging over 3 random seeds used for training at a discount factor of 0.995, with standard deviations shown after ±.

| FQE | pen | door | relocate | hammer | MB | pen | door | relocate | hammer |
|---|---|---|---|---|---|---|---|---|---|
| NoAug. | 3872±140 | 389±60 | 593±113 | 6000±612 | | 1218±23 | 403±18 | 353±21 | 3778±78 |
| OAT | **1231±33** | 464±17 | 742±7 | **3500±42** | | **316±126** | **461±4** | **298±5** | 3372±28 |
| VAE-MDP | 843±44 | 498±10 | **419±6** | 3358±24 | | 541±310 | 493±3 | 423±1 | **2912±13** |
| TimeGAN | 919±141 | 459±92 | 962±96 | 4353±495 | | 1063±450 | 435±34 | 413±1 | 5811±97 |
| VAE | 614±9 | 531±7 | 520±2 | 3746±16 | | 615±14 | 530±3 | 421±1 | 3734±9 |
| SPAWNER | 1508±52 | 504±35 | 735±131 | 6529±172 | | 1107±0 | 473±4 | 698±4 | 3561±11 |
| DGW | 792±222 | **320±20** | 787±69 | 9578±431 | | 1278±175 | 430±29 | 810±6 | 3779±37 |
| Permutation | 924±140 | 520±41 | 542±10 | 4069±56 | | 998±0 | 483±4 | 754±4 | 3324±11 |
| Jittering | 1092±105 | 403±30 | 746±0 | 5526±323 | | 1064±9 | 539±11 | 512±60 | 3257±103 |
| Scaling | 875±52 | 360±20 | 624±179 | 4599±306 | | 961±337 | 478±5 | 524±84 | 3627±63 |
| TDA | 1185±29 | 469±6 | 805±10 | 3407±22 | | **471±248** | 470±4 | 810±4 | 3402±11 |

| IS | pen | door | relocate | hammer | DR | pen | door | relocate | hammer |
|---|---|---|---|---|---|---|---|---|---|
| NoAug. | 3926±128 | 870±173 | 3926±128 | 7352±1118 | | 2846±200 | 379±65 | 606±116 | 5768±751 |
| OAT | **1364±97** | **508±5** | **436±2** | 3779±32 | | 1015±186 | 494±6 | **450±27** | **3815±142** |
| VAE-MDP | **1315±383** | **499±6** | **437±4** | 3678±83 | | 2954±883 | 502±10 | **451±23** | **3811±262** |
| TimeGAN | 1752±212 | 591±12 | 1995±5 | 5683±12 | | 1352±282 | 494±70 | 667±122 | 4224±138 |
| VAE | 1896±87 | 513±5 | 930±7 | 3628±174 | | **784±155** | 515±7 | **545±16** | 3832±166 |
| SPAWNER | 2769±0 | 1007±9 | 2871±19 | 3567±10 | | 870±39 | 450±86 | 591±69 | 4008±444 |
| DGW | 2360±0 | 541±10 | 534±8 | 5289±11 | | 861±99 | 545±55 | 573±38 | 4270±92 |
| Permutation | 2433±11 | 520±13 | 3093±20 | 5332±11 | | **787±149** | **368±42** | 613±84 | 4467±245 |
| Jittering | 2350±0 | 1114±2 | 2111±28 | 5334±10 | | 1058±127 | 419±12 | 519±28 | 3841±274 |
| Scaling | **1284±40** | 523±8 | 2118±72 | 3710±16 | | 822±132 | 525±7 | 642±29 | 3892±239 |
| TDA | **1269±101** | 572±5 | 882±25 | **3418±134** | | 991±193 | 477±11 | 857±156 | **3613±261** |

| DICE | pen | door | relocate | hammer |
|---|---|---|---|---|
| NoAug. | 3208±22 | 978±10 | 4304±68 | 3432±6 |
| OAT | **1285±5** | **462±5** | 759±14 | **1473±12** |
| VAE-MDP | 1986±40 | 494±5 | **424±3** | 3334±9 |
| TimeGAN | 2605±15 | 556±6 | 2008±15 | 3334±9 |
| VAE | 2603±3 | 1125±11 | 1758±10 | 3726±18 |
| SPAWNER | 2835±11 | 1004±10 | 2908±49 | 3559±12 |
| DGW | 2599±0 | 1055±10 | 3430±63 | 3278±9 |
| Permutation | 2601±2 | 1026±11 | 3138±49 | 3324±10 |
| Jittering | 2600±1 | 1026±11 | 3149±51 | 3324±10 |
| Scaling | 2599±0 | 1018±11 | 3406±60 | 3334±10 |
| TDA | **1236±8** | 568±5 | 807±14 | 3400±11 |

Table 6: Rank correlation results of OPE without and with each augmentation method in Adroit `human` environment. Results are obtained by averaging over 3 random seeds used for training at a discount factor of 0.995, with standard deviations shown after ±.

| FQE | pen | door | relocate | hammer | MB | pen | door | relocate | hammer |
|-----|-----|------|----------|--------|-----|-----|------|----------|--------|
| NoAug. | 0.31±0.21 | 0.07±0.09 | 0.62±0.11 | 0.14±0.10 | | -0.12±0.33 | 0.13±0.13 | 0.16±0.10 | 0.29±0.23 |
| OAT | -0.02±0.65 | **0.55±0.60** | -0.05±0.55 | **0.34±0.92** | | **0.67±0.08** | **0.99±0.01** | 0.40±0.78 | **0.62±0.47** |
| VAE-MDP | -0.02±0.64 | **0.68±0.24** | -0.86±0.04 | -0.44±0.72 | | 0.12±0.34 | 0.14±0.58 | **0.71±0.16** | 0.29±0.00 |
| TimeGAN | -0.17±0.28 | 0.39±0.20 | **0.76±0.07** | **0.37±0.06** | | 0.52±0.48 | 0.18±0.13 | -0.02±0.19 | 0.16±0.23 |
| VAE | -0.52±0.38 | 0.39±0.20 | -0.83±0.11 | -0.61±0.46 | | 0.46±0.28 | 0.93±0.02 | 0.25±0.30 | 0.11±0.67 |
| SPAWNER | 0.12±0.20 | 0.44±0.17 | 0.62±0.11 | -0.11±0.31 | | 0.13±0.44 | 0.04±0.79 | 0.19±0.60 | -0.82±0.02 |
| DGW | -0.12±0.25 | 0.47±0.21 | 0.17±0.36 | **0.47±0.16** | | -0.02±0.15 | 0.29±0.19 | 0.35±0.31 | 0.20±0.19 |
| Permutation | -0.17±0.18 | 0.50±0.05 | 0.43±0.09 | **0.48±0.03** | | -0.18±0.56 | 0.02±0.74 | 0.23±0.07 | -0.85±0.10 |
| Jittering | -0.17±0.21 | 0.45±0.00 | -0.33±0.78 | -0.29±0.26 | | 0.05±0.04 | 0.31±0.01 | 0.17±0.42 | -0.22±0.20 |
| Scaling | **0.36±0.24** | 0.53±0.06 | 0.40±0.15 | **0.35±0.34** | | -0.35±0.38 | 0.21±0.65 | 0.24±0.33 | 0.35±0.15 |
| TDA | -0.26±0.68 | -0.41±0.35 | -0.27±0.89 | -0.12±0.52 | | 0.09±0.09 | 0.72±0.11 | **0.71±0.12** | -0.21±0.60 |

| IS | pen | door | relocate | hammer | DR | | door | relocate | hammer |
|-----|-----|------|----------|--------|-----|-----|------|----------|--------|
| NoAug. | 0.28±0.28 | 0.12±0.35 | 0.23±0.07 | 0.39±0.07 | | 0.36±0.29 | 0.01±0.18 | 0.65±0.19 | 0.04±0.25 |
| OAT | **0.57±0.31** | **0.81±0.06** | **0.82±0.07** | 0.75±0.14 | | 0.04±0.64 | 0.45±0.46 | 0.29±0.70 | **0.74±0.16** |
| VAE-MDP | 0.48±0.36 | 0.12±0.71 | 0.22±0.70 | -0.28±0.51 | | **0.34±0.68** | 0.50±0.11 | 0.72±0.18 | -0.35±0.79 |
| TimeGAN | -0.01±0.80 | 0.38±0.75 | - | -0.85±0.01 | | -0.20±0.64 | 0.53±0.10 | 0.65±0.08 | 0.40±0.03 |
| VAE | - | 0.65±0.00 | -0.31±0.00 | 0.31±0.86 | | -0.17±0.64 | 0.29±0.29 | -0.04±0.69 | 0.06±0.75 |
| SPAWNER | -0.83±0 | - | - | **0.81±0.16** | | -0.09±0.20 | 0.36±0.26 | 0.62±0.09 | 0.47±0.16 |
| DGW | - | -0.03±0.61 | **0.82±0.16** | 0.26±0.57 | | 0.02±0.06 | 0.63±0.15 | 0.09±0.67 | 0.27±0.36 |
| Permutation | -0.21±0.74 | 0.37±0.56 | -0.70±0.00 | - | | -0.12±0.37 | 0.57±0.14 | 0.38±0.31 | 0.27±0.36 |
| Jittering | -0.28±0.79 | 0.12±0.38 | **0.85±0.00** | 0.23±0.73 | | -0.08±0.51 | **0.64±0.18** | -0.04±0.42 | 0.18±0.84 |
| Scaling | - | 0.24±0.42 | **0.65±0.00** | 0.66±0.14 | | -0.15±0.67 | 0.26±0.45 | 0.09±0.67 | 0.09±0.67 |
| TDA | -0.15±0.53 | 0.37±0.28 | -0.67±0.00 | **0.81±0.13** | | -0.11±0.63 | 0.18±0.34 | -0.22±0.85 | -0.28±0.79 |

| DICE | pen | door | relocate | hammer |
|------|-----|------|----------|--------|
| NoAug. | -0.01±0.39 | **0.61±0.34** | -0.18±0.45 | **0.94±0.01** |
| OAT | **0.12±0.70** | **0.58±0.29** | **0.35±0.88** | 0.52±0.67 |
| VAE-MDP | **0.21±0.59** | 0.24±0.53 | 0.02±0.68 | 0.38±0.73 |
| TimeGAN | -0.11±0.19 | 0.48±0.09 | 0.27±0.84 | 0.38±0.73 |
| VAE | 0.08±0.45 | 0.15±0.74 | 0.17±0.76 | -0.24±0.16 |
| SPAWNER | -0.29±0.59 | -0.11±0.71 | 0.26±0.83 | -0.32±0.58 |
| DGW | -0.02±0.54 | 0.35±0.57 | 0.24±0.87 | -0.55±0.28 |
| Permutation | 0.02±0.64 | -0.14±0.59 | 0.28±0.88 | -0.21±0.75 |
| Jittering | 0.00±0.58 | -0.18±0.60 | 0.27±0.84 | -0.50±0.45 |
| Scaling | 0.04±0.67 | -0.13±0.58 | 0.25±0.87 | -0.61±0.30 |
| TDA | -0.15±0.64 | -0.27±0.62 | **0.32±0.91** | -0.21±0.84 |

Table 7: Regret@1 results of OPE without and with each augmentation method in Adroit `human` environment. Results are obtained by averaging over 3 random seeds used for training at a discount factor of 0.995, with standard deviations shown after $\pm$.

| FQE | pen | door | relocate | hammer | MB | pen | door | relocate | hammer |
|---|---|---|---|---|---|---|---|---|---|
| NoAug. | 0.07±0.05 | 0.05±0.08 | **0.17±0.14** | 0.46±0.23 | | 0.15±0.15 | 0.44±0.42 | 0.73±0.36 | 0.15±0.17 |
| OAT | 0.20±0.26 | 0.26±0.36 | 0.99±0.01 | 0.34±0.48 | | 0.12±0.17 | **0.00±0.00** | 0.35±0.46 | **0.00±0.00** |
| VAE-MDP | 0.38±0.21 | **0.01±0.01** | 1.00±0.01 | 0.81±0.30 | | 0.13±0.11 | 0.39±0.45 | 0.41±0.28 | 0.42±0.16 |
| TimeGAN | 0.19±0.13 | 0.23±0.13 | 0.39±0.24 | 0.15±0.17 | | 0.22±0.25 | 0.51±0.36 | 1.00±0.00 | 0.19±0.24 |
| VAE | 0.57±0.00 | 0.05±0.05 | 1.00±0.01 | 0.73±0.26 | | 0.09±0.06 | **0.00±0.01** | **0.03±0.02** | 0.34±0.48 |
| SPAWNER | **0.03±0.00** | **0.01±0.00** | 0.27±0.32 | 0.94±0.11 | | 0.12±0.12 | 0.35±0.48 | 0.41±0.43 | 1.02±0.00 |
| DGW | 0.23±0.17 | 0.14±0.08 | 0.32±0.42 | **0.02±0.01** | | 0.33±0.12 | 0.12±0.09 | 0.67±0.47 | 0.14±0.18 |
| Permutation | 0.37±0.16 | 0.15±0.16 | 0.87±0.11 | 0.05±0.03 | | 0.19±0.13 | 0.37±0.47 | 0.09±0.09 | 1.02±0.00 |
| Jittering | 0.11±0.06 | 0.12±0.00 | 0.67±0.47 | 0.73±0.26 | | **0.08±0.07** | 0.63±0.10 | 0.48±0.41 | 0.45±0.31 |
| Scaling | 0.13±0.16 | 0.17±0.11 | 0.81±0.12 | 0.25±0.22 | | 0.50±0.10 | 0.68±0.48 | 0.94±0.04 | 0.14±0.08 |
| TDA | 0.35±0.25 | 0.79±0.29 | 0.68±0.45 | 0.81±0.30 | | 0.26±0.23 | 0.12±0.10 | 0.31±0.30 | 0.72±0.37 |

| IS | pen | door | relocate | hammer | DR | pen | door | relocate | hammer |
|---|---|---|---|---|---|---|---|---|---|
| NoAug. | 0.17±0.15 | 0.45±0.40 | 0.63±0.41 | 0.19±0.30 | | **0.09±0.00** | 0.05±0.09 | 0.17±0.15 | 0.46±0.23 |
| OAT | 0.12±0.17 | **0.06±0.05** | **0.02±0.02** | **0.00±0.00** | | 0.20±0.19 | **0.00±0.01** | 0.37±0.45 | **0.05±0.05** |
| VAE-MDP | **0.05±0.03** | 0.37±0.47 | 0.37±0.45 | 0.30±0.34 | | 0.16±0.22 | **0.00±0.01** | 0.05±0.00 | 0.71±0.43 |
| TimeGAN | 0.38±0.27 | 0.13±0.17 | 1.00±0.00 | 1.00±0.02 | | 0.16±0.21 | 0.13±0.17 | 0.26±0.33 | 0.33±0.32 |
| VAE | 0.44±0.19 | 0.68±0.48 | 1.00±0.00 | 0.19±0.25 | | 0.36±0.15 | 0.21±0.25 | 0.74±0.37 | 0.37±0.46 |
| SPAWNER | 0.57±0.00 | 1.03±0.00 | 1.00±0.00 | 0.34±0.45 | | 0.12±0.08 | **0.01±0.01** | **0.00±0.00** | **0.02±0.25** |
| DGW | 0.44±0.19 | 0.15±0.16 | **0.00±0.00** | 0.59±0.28 | | 0.25±0.24 | 0.03±0.02 | 0.68±0.44 | **0.03±0.00** |
| Permutation | 0.44±0.19 | 0.05±0.06 | 0.91±0.13 | 0.74±0.39 | | 0.10±0.06 | 0.08±0.12 | 0.31±0.19 | 0.50±0.40 |
| Jittering | 0.44±0.19 | 0.34±0.48 | 0.68±0.45 | 0.36±0.46 | | 0.10±0.05 | 0.09±0.11 | 0.54±0.31 | 0.35±0.47 |
| Scaling | 0.57±0.00 | 0.06±0.05 | 0.67±0.47 | 0.26±0.18 | | 0.39±0.19 | 0.19±0.26 | 0.68±0.44 | 0.68±0.44 |
| TDA | 0.10±0.13 | 0.05±0.06 | 1.00±0.00 | **0.00±0.01** | | 0.39±0.19 | 0.35±0.46 | 0.68±0.44 | 0.60±0.43 |

| DICE | pen | door | relocate | hammer |
|---|---|---|---|---|
| NoAug. | 0.44±0.19 | **0.38±0.46** | 0.47±0.40 | 0.44±0.01 |
| OAT | **0.19±0.14** | 0.44±0.09 | **0.37±0.45** | **0.03±0.05** |
| VAE-MDP | 0.30±0.22 | 0.35±0.15 | 0.82±0.26 | **0.03±0.05** |
| TimeGAN | **0.13±0.12** | 0.35±0.15 | **0.37±0.45** | **0.03±0.05** |
| VAE | 0.21±0.15 | 0.50±0.35 | 0.50±0.39 | 0.72±0.37 |
| SPAWNER | 0.25±0.11 | 0.45±0.32 | 0.50±0.39 | 0.43±0.39 |
| DGW | 0.21±0.11 | 0.23±0.23 | 0.50±0.39 | 0.70±0.42 |
| Permutation | 0.25±0.11 | 0.53±0.22 | **0.37±0.45** | 0.36±0.44 |
| Jittering | 0.28±0.15 | 0.53±0.22 | 0.50±0.39 | 0.37±0.43 |
| Scaling | 0.32±0.20 | 0.53±0.22 | 0.50±0.39 | 0.69±0.41 |
| TDA | 0.21±0.15 | 0.57±0.16 | **0.37±0.45** | 0.67±0.47 |

Table 8: Regret@5 results of OPE without and with each augmentation method in Adroit `human` environment. Results are obtained by averaging over 3 random seeds used for training at a discount factor of 0.995, with standard deviations shown after ±.

| FQE | pen | door | relocate | hammer | MB | pen | door | relocate | hammer |
|---|---|---|---|---|---|---|---|---|---|
| NoAug. | **0.01±0.02** | **0.00±0.01** | 0.75±0.22 | 0.13±0.09 | | 0.03±0.04 | 0.02±0.03 | 0.03±0.02 | 0.01±0.01 |
| OAT | **0.01±0.02** | 0.13±0.18 | 0.15±0.21 | 0.07±0.09 | | **0.00±0.00** | **0.00±0.00** | 0.02±0.02 | **0.00±0.00** |
| VAE-MDP | 0.06±0.08 | **0.00±0.00** | 0.87±0.10 | 0.13±0.09 | | **0.01±0.01** | 0.08±0.12 | **0.00±0.00** | 0.17±0.05 |
| TimeGAN | 0.03±0.04 | **0.00±0.01** | 0.47±0.35 | 0.13±0.09 | | **0.00±0.00** | 0.01±0.01 | 0.02±0.02 | 0.02±0.01 |
| VAE | 0.06±0.08 | **0.00±0.00** | 0.75±0.22 | 0.13±0.09 | | **0.00±0.00** | **0.00±0.00** | **0.00±0.00** | 0.07±0.09 |
| SPAWNER | 0.02±0.01 | **0.01±0.00** | **0.00±0.00** | **0.01±0.01** | | 0.02±0.01 | 0.19±0.26 | 0.07±0.10 | 0.17±0.05 |
| DGW | **0.01±0.02** | **0.00±0.00** | 0.68±0.32 | 0.10±0.08 | | 0.03±0.04 | 0.01±0.01 | 0.15±0.10 | 0.01±0.01 |
| Permutation | **0.00±0.00** | **0.00±0.00** | **0.02±0.02** | **0.01±0.01** | | 0.10±0.06 | 0.19±0.26 | **0.00±0.00** | 0.23±0.12 |
| Jittering | **0.00±0.00** | **0.00±0.00** | 0.45±0.37 | 0.02±0.01 | | 0.08±0.07 | 0.01±0.01 | 0.07±0.10 | 0.05±0.04 |
| Scaling | **0.00±0.00** | **0.01±0.01** | 0.03±0.02 | **0.00±0.01** | | 0.08±0.07 | 0.08±0.12 | 0.07±0.10 | 0.01±0.01 |
| TDA | 0.07±0.08 | 0.27±0.23 | 0.56±0.41 | **0.01±0.01** | | 0.10±0.06 | 0.00±0.00 | 0.07±0.10 | 0.15±0.17 |

| IS | pen | door | relocate | hammer | DR | pen | door | relocate | hammer |
|---|---|---|---|---|---|---|---|---|---|
| NoAug. | **0.00±0.00** | **0.00±0.00** | 0.07±0.10 | 0.01±0.02 | | 0.10±0.08 | **0.00±0.01** | 0.32±0.46 | 0.14±0.18 |
| OAT | **0.00±0.00** | **0.00±0.00** | **0.00±0.00** | **0.00±0.00** | | 0.06±0.08 | **0.00±0.01** | 0.32±0.46 | **0.01±0.01** |
| VAE-MDP | **0.01±0.01** | 0.09±0.11 | 0.30±0.43 | 0.14±0.18 | | **0.01±0.01** | **0.00±0.00** | 0.02±0.02 | 0.16±0.16 |
| TimeGAN | 0.01±0.01 | 0.08±0.12 | 0.97±0.00 | 1.02±0.00 | | 0.06±0.08 | 0.00±0.01 | 0.09±0.09 | **0.00±0.00** |
| VAE | 0.19±0.27 | 0.00±0.00 | 0.95±0.03 | 0.13±0.18 | | 0.05±0.06 | 0.08±0.12 | 0.40±0.41 | 0.14±0.18 |
| SPAWNER | 0.57±0.00 | 0.56±0.00 | 0.97±0.00 | 1.02±0.00 | | 0.04±0.04 | 0.01±0.01 | **0.00±0.00** | 0.60±0.43 |
| DGW | 0.19±0.27 | 0.15±0.16 | **0.00±0.00** | 0.46±0.40 | | **0.01±0.01** | 0.01±0.01 | 0.34±0.45 | **0.00±0.01** |
| Permutation | 0.19±0.27 | **0.04±0.06** | 0.44±0.00 | **0.00±0.00** | | 0.03±0.00 | **0.00±0.00** | **0.00±0.00** | **0.00±0.01** |
| Jittering | 0.19±0.27 | **0.00±0.00** | **0.00±0.00** | 0.15±0.17 | | 0.05±0.06 | 0.00±0.01 | 0.02±0.02 | 0.13±0.18 |
| Scaling | 0.18±0.00 | 0.05±0.06 | 0.32±0.46 | 0.01±0.01 | | 0.06±0.08 | 0.08±0.12 | 0.34±0.45 | 0.34±0.45 |
| TDA | **0.01±0.02** | **0.00±0.01** | 0.97±0.00 | 0.00±0.00 | | 0.06±0.08 | **0.00±0.00** | 0.56±0.41 | 0.16±0.16 |

| DICE | pen | door | relocate | hammer |
|---|---|---|---|---|
| NoAug. | **0.01±0.01** | 0.00±0.01 | 0.32±0.42 | **0.00±0.00** |
| OAT | 0.06±0.08 | **0.00±0.00** | 0.30±0.43 | **0.01±0.02** |
| VAE-MDP | **0.01±0.01** | 0.09±0.11 | **0.09±0.09** | 0.03±0.05 |
| TimeGAN | **0.01±0.02** | 0.06±0.00 | 0.15±0.21 | 0.03±0.05 |
| VAE | **0.01±0.02** | 0.15±0.16 | 0.32±0.42 | 0.05±0.03 |
| SPAWNER | 0.07±0.08 | 0.16±0.12 | 0.32±0.42 | 0.05±0.04 |
| DGW | 0.03±0.04 | 0.09±0.11 | 0.30±0.43 | 0.15±0.17 |
| Permutation | 0.06±0.08 | 0.21±0.15 | 0.30±0.43 | 0.14±0.18 |
| Jittering | 0.06±0.08 | 0.27±0.22 | 0.30±0.43 | 0.16±0.16 |
| Scaling | 0.06±0.08 | 0.17±0.11 | 0.30±0.43 | 0.15±0.17 |
| TDA | 0.06±0.08 | 0.29±0.21 | 0.32±0.46 | 0.07±0.05 |

Table 9: MAE results of OPE without and with each augmentation method in resampled Adroit `cloned` environment. The data are randomly sampled from original training data as the same data points as the corresponding task in `human` environment. Results are obtained by averaging over 3 random seeds used for training at a discount factor of 0.995, with standard deviations shown after $\pm$.

| FQE | pen | door | relocate | hammer | MB | pen | door | relocate | hammer |
|---|---|---|---|---|---|---|---|---|---|
| NoAug. | 715±11 | 359±16 | 371±5 | 3714±34 | | 622±102 | 1009±92 | 357±6 | 4119±7 |
| OAT | 663±24 | **336±7** | 438±3 | 3780±20 | | 607±2 | 537±3 | **339±1** | 3743±18 |
| VAE-MDP | **323±9** | 539±6 | 384±4 | **3624±16** | | 813±674 | 538±3 | 386±1 | **3611±4** |
| TimeGAN | 1237±25 | 572±7 | 539±4 | 5268±24 | | 1065±118 | 573±4 | 541±1 | 5268±15 |
| VAE | 412±8 | 561±6 | 384±2 | 3823±14 | | **409±19** | 561±3 | 385±1 | 3814±10 |
| SPAWNER | 1173±0 | **502±5** | 443±2 | 3988±6 | | 1173±0 | 502±3 | 446±1 | 3988±6 |
| DGW | 1176±0 | 550±5 | 864±8 | 4127±6 | | 1176±0 | 550±3 | 871±4 | 4127±6 |
| Permutation | 1176±0 | 537±5 | 822±11 | 4128±6 | | 1176±0 | 536±3 | 832±4 | 4128±6 |
| Jittering | 1176±0 | 537±5 | 820±11 | 4128±6 | | 1176±0 | 536±3 | 833±4 | 4128±6 |
| Scaling | 1176±0 | 517±6 | 696±5 | 4123±6 | | 1176±0 | 517±3 | 705±4 | 4123±6 |
| TDA | 610±4 | 526±6 | 438±3 | 3746±16 | | 599±0 | 527±3 | 441±1 | 3715±14 |

| IS | pen | door | relocate | hammer | DR | pen | door | relocate | hammer |
|---|---|---|---|---|---|---|---|---|---|
| NoAug. | 636±21 | 1072±24 | 458±13 | 8162±91 | | 731±115 | 458±34 | 475±22 | 6719±118 |
| OAT | **259±9** | 563±3 | **369±2** | 3905±9 | | **422±9** | **390±41** | **371±8** | 3885±166 |
| VAE-MDP | 300±17 | **507±4** | 406±11 | 3686±47 | | 493±76 | 542±20 | 415±40 | **3689±89** |
| TimeGAN | 978±192 | 576±4 | 695±27 | 5325±85 | | 771±191 | 606±18 | 696±26 | 5514±133 |
| VAE | 300±21 | 517±2 | 433±15 | **3722±15** | | **386±19** | 540±22 | 437±20 | 3764±80 |
| SPAWNER | 1173±0 | 611±3 | 506±1 | 3989±6 | | 1174±0 | 527±14 | 509±14 | 3990±5 |
| DGW | 1176±0 | 567±3 | 1149±4 | 4129±6 | | 1176±0 | 597±23 | 1180±35 | 4129±5 |
| Permutation | 1176±0 | 557±3 | 1107±4 | 4129±6 | | 1176±0 | 592±25 | 1110±39 | 4129±5 |
| Jittering | 1176±0 | 557±3 | 1107±4 | 4129±6 | | 1176±0 | 588±24 | 1115±89 | 4130±5 |
| Scaling | 1176±0 | 543±3 | 945±4 | 4124±6 | | 1176±0 | 580±31 | 968±35 | 4125±5 |
| TDA | 604±6 | **501±5** | 445±10 | **3725±11** | | 261±12 | 539±34 | 453±9 | 3765±142 |

| DICE | pen | door | relocate | hammer |
|---|---|---|---|---|
| NoAug. | 1218±41 | 1138±7 | 1841±15 | 3752±8 |
| OAT | 1166±38 | **536±3** | **441±1** | 3750±11 |
| VAE-MDP | 778±5 | **538±4** | 1606±8 | **3614±15** |
| TimeGAN | 1276±42 | 573±4 | 1925±10 | 5252±18 |
| VAE | 1020±14 | 561±5 | 1602±10 | 3813±20 |
| SPAWNER | 1173±0 | 1067±7 | 1856±4 | 3988±6 |
| DGW | 1176±0 | 1140±3 | 869±8 | 4127±6 |
| Permutation | 1176±0 | 1140±7 | 832±5 | 4128±6 |
| Jittering | 1176±0 | 1140±7 | 831±8 | 4128±6 |
| Scaling | 1176±0 | 1098±7 | 704±8 | 4123±6 |
| TDA | **601±0** | 1119±9 | 1835±6 | **3731±15** |

Table 10: Rank correlation results of OPE without and with each augmentation method in resampled Adroit `cloned` environment. The data are randomly sampled from original training data as the same data points as the corresponding task in `human` environment. Results are obtained by averaging over 3 random seeds used for training at a discount factor of 0.995, with standard deviations shown after $\pm$.

| FQE | pen | door | relocate | hammer | MB | pen | door | relocate | hammer |
|---|---|---|---|---|---|---|---|---|---|
| NoAug. | **0.51±0.25** | 0.55 ± 0.27 | -0.28±0.17 | 0.50 ± 0.09 | | **0.29±0.19** | 0.63±0.14 | 0.45±0.30 | 0.22±0.12 |
| OAT | -0.26±0.32 | **0.95±0.02** | **0.48±0.51** | **0.35±0.89** | | **0.31±0.54** | **0.77±0.05** | 0.52±0.15 | 0.38±0.52 |
| VAE-MDP | -0.18±0.44 | 0.26±0.52 | -0.32±0.65 | 0.28±0.78 | | 0.10±0.44 | 0.19±0.18 | **0.88±0.11** | 0.05±0.61 |
| TimeGAN | -0.39±0.43 | 0.54±0.28 | -0.47±0.41 | -0.01±0.71 | | -0.12±0.63 | 0.58±0.34 | 0.46±0.71 | **0.68±0.30** |
| VAE | -0.69±0.19 | 0.38±0.48 | -0.40±0.62 | 0.16±0.50 | | **0.37±0.47** | 0.56±0.29 | 0.36±0.73 | -0.23±0.23 |
| SPAWNER | -0.19±0.68 | 0.18±0.59 | -0.32±0.73 | 0.02±0.79 | | 0.05±0.70 | 0.62±0.06 | 0.79±0.19 | 0.40±0.82 |
| DGW | -0.46±0.49 | -0.07±0.67 | -0.75±0.11 | -0.32±0.82 | | -0.30±0.20 | 0.47±0.14 | -0.09±0.58 | 0.09±0.25 |
| Permutation | -0.36±0.69 | -0.28±0.24 | 0.12±0.59 | -0.29±0.79 | | 0.15±0.25 | 0.60±0.13 | 0.43±0.75 | 0.51±0.32 |
| Jittering | -0.32±0.80 | 0.54±0.49 | -0.17±0.53 | -0.47±0.49 | | 0.25±0.60 | 0.67±0.10 | -0.19±0.46 | 0.45±0.44 |
| Scaling | -0.19±0.80 | **0.63±0.22** | -0.25±0.64 | -0.26±0.83 | | 0.03±0.62 | 0.45±0.30 | 0.67±0.29 | 0.66±0.18 |
| TDA | -0.26±0.77 | 0.37±0.26 | -0.22±0.48 | -0.31±0.87 | | 0.00±0.50 | 0.35±0.33 | 0.61±0.27 | -0.45±0.62 |

| IS | pen | door | relocate | hammer | DR | pen | door | relocate | hammer |
|---|---|---|---|---|---|---|---|---|---|
| NoAug. | 0.00±0.00 | -0.32±0.59 | 0.25±0.54 | 0.79±0.04 | | **0.42±0.24** | **0.46±0.72** | -0.22±0.70 | 0.29±0.49 |
| OAT | **0.78±0.00** | **0.96±0.01** | **0.99±0.00** | 0.77±0.15 | | -0.38±0.50 | 0.23±0.53 | **0.33±0.78** | **0.62±0.37** |
| VAE-MDP | 0.15±0.28 | 0.55±0.06 | 0.96±0.04 | -0.02±0.74 | | -0.15±0.60 | -0.57±0.06 | -0.02±0.19 | -0.23±0.82 |
| TimeGAN | 0.89±0.00 | -0.78±0.27 | 0.88±0.08 | 0.50±0.29 | | -0.43±0.36 | -0.42±0.14 | -0.10±0.62 | -0.18±0.78 |
| VAE | 0.61±0.11 | -0.23±0.66 | 0.88±0.10 | **0.98±0.01** | | -0.46±0.34 | -0.10±0.62 | -0.02±0.70 | -0.20±0.76 |
| SPAWNER | -0.89±0.00 | -0.10±0.00 | -0.29±0.25 | 0.37±0.39 | | -0.23±0.66 | 0.02±0.64 | -0.06±0.64 | -0.16±0.78 |
| DGW | - | -0.45±0.00 | -0.67±0.00 | - | | -0.05±0.32 | -0.36±0.31 | -0.09±0.63 | -0.20±0.77 |
| Permutation | - | -0.50±0.00 | -0.67±0.00 | - | | -0.18±0.36 | -0.43±0.21 | -0.20±0.71 | -0.22±0.75 |
| Jittering | - | - | -0.42±0.02 | - | | -0.31±0.26 | -0.33±0.39 | 0.03±0.74 | 0.06±0.78 |
| Scaling | - | -0.67±0.00 | -0.24±0.44 | - | | 0.07±0.55 | -0.31±0.38 | -0.25±0.71 | -0.20±0.73 |
| TDA | 0.37±0.42 | 0.41±0.22 | -0.48±0.35 | 0.79±0.13 | | -0.22±0.37 | -0.32±0.41 | -0.05±0.70 | -0.25±0.79 |

| DICE | pen | door | relocate | hammer |
|---|---|---|---|---|
| NoAug. | 0.03±0.56 | **0.39±0.51** | 0.05±0.52 | **0.31±0.76** |
| OAT | **0.24±0.71** | **0.41±0.57** | 0.36±0.89 | **0.44±0.62** |
| VAE-MDP | -0.61±0.38 | 0.26±0.65 | **0.82±0.17** | -0.28±0.79 |
| TimeGAN | -0.36±0.36 | 0.06±0.70 | 0.77±0.24 | -0.43±0.62 |
| VAE | -0.18±0.45 | 0.18±0.79 | 0.30±0.61 | -0.11±0.64 |
| SPAWNER | -0.30±0.33 | **0.47±0.66** | -0.18±0.78 | -0.19±0.73 |
| DGW | **0.16±0.10** | **0.38±0.58** | 0.22±0.85 | -0.36±0.45 |
| Permutation | 0.24±0.43 | -0.48±0.11 | **0.96±0.02** | -0.20±0.75 |
| Jittering | -0.39±0.61 | **0.37±0.43** | -0.30±0.82 | -0.14±0.77 |
| Scaling | **0.14±0.60** | **0.40±0.55** | 0.14±0.81 | -0.16±0.70 |
| TDA | -0.02±0.56 | -0.11±0.61 | 0.30±0.83 | -0.31±0.78 |

Table 11: Regret@1 results of OPE without and with each augmentation method in resampled Adroit `cloned` environment. The data are randomly sampled from original training data as the same data points as the corresponding task in `human` environment. Results are obtained by averaging over 3 random seeds used for training at a discount factor of 0.995, with standard deviations shown after ±.

| FQE | pen | door | relocate | hammer | MB | pen | door | relocate | hammer |
|---|---|---|---|---|---|---|---|---|---|
| NoAug. | 0.10±0.13 | 0.07±0.04 | 1.00±0.01 | **0.03±0.01** | | **0.06±0.06** | 0.19±0.14 | 0.34±0.45 | 0.21±0.18 |
| OAT | 0.47±0.00 | **0.01±0.01** | **0.33±0.47** | 0.34±0.48 | | **0.10±0.13** | 0.04±0.03 | 0.48±0.37 | 0.13±0.18 |
| VAE-MDP | 0.30±0.22 | 0.13±0.17 | 0.67±0.47 | 0.26±0.37 | | 0.20±0.26 | 0.79±0.19 | **0.02±0.02** | 0.39±0.31 |
| TimeGAN | 0.41±0.23 | 0.39±0.45 | 1.00±0.00 | 0.40±0.44 | | 0.31±0.22 | 0.42±0.44 | 0.35±0.46 | 0.08±0.08 |
| VAE | 0.57±0.00 | 0.46±0.41 | 0.66±0.47 | 0.47±0.39 | | 0.12±0.06 | 0.08±0.12 | 0.33±0.47 | 0.94±0.11 |
| SPAWNER | 0.20±0.26 | 0.44±0.32 | 0.67±0.47 | 0.37±0.46 | | 0.13±0.16 | 0.21±0.25 | 0.26±0.33 | 0.26±0.37 |
| DGW | **0.05±0.03** | 0.43±0.43 | 0.99±0.02 | 0.68±0.48 | | 0.57±0.00 | 0.52±0.41 | 0.90±0.12 | 0.01±0.02 |
| Permutation | 0.38±0.27 | 0.72±0.27 | 0.74±0.36 | 0.60±0.43 | | 0.57±0.00 | **0.00±0.00** | 0.40±0.42 | 0.34±0.48 |
| Jittering | 0.35±0.25 | 0.09±0.11 | 0.67±0.47 | 0.47±0.39 | | 0.14±0.11 | 0.02±0.03 | 0.81±0.12 | **0.01±0.02** |
| Scaling | 0.35±0.25 | 0.09±0.11 | 0.67±0.47 | 0.60±0.43 | | 0.24±0.24 | 0.19±0.26 | 0.59±0.39 | 0.01±0.01 |
| TDA | 0.35±0.25 | 0.19±0.26 | 0.67±0.47 | 0.68±0.48 | | 0.17±0.21 | 0.47±0.42 | 0.40±0.42 | 0.81±0.30 |

| IS | pen | door | relocate | hammer | DR | pen | door | relocate | hammer |
|---|---|---|---|---|---|---|---|---|---|
| NoAug. | 0.57±0.00 | 0.47±0.42 | 0.68±0.45 | **0.34±0.48** | | 0.22±0.25 | **0.26±0.36** | 0.68±0.45 | 0.26±0.37 |
| OAT | **0.19±0.27** | **0.05±0.06** | **0.00±0.00** | **0.00±0.01** | | 0.28±0.20 | **0.26±0.36** | **0.37±0.45** | **0.00±0.01** |
| VAE-MDP | **0.19±0.13** | **0.07±0.04** | **0.00±0.00** | 0.35±0.47 | | **0.17±0.10** | 0.93±0.11 | 0.82±0.26 | 0.37±0.46 |
| TimeGAN | 0.57±0.00 | 1.02±0.00 | 0.02±0.02 | 0.85±0.09 | | 0.50±0.05 | 0.93±0.11 | 0.82±0.26 | 0.37±0.46 |
| VAE | 0.35±0.24 | 0.68±0.48 | **0.00±0.00** | **0.34±0.46** | | 0.44±0.19 | 0.62±0.41 | 0.67±0.47 | 0.37±0.46 |
| SPAWNER | 0.57±0.00 | 0.81±0.31 | 1.00±0.00 | 0.37±0.43 | | 0.35±0.24 | 0.60±0.43 | 0.68±0.45 | 0.37±0.46 |
| DGW | 0.57±0.00 | 1.03±0.00 | 1.00±0.00 | 1.02±0.00 | | 0.35±0.24 | 0.93±0.11 | 0.68±0.45 | 0.37±0.46 |
| Permutation | 0.57±0.00 | 1.03±0.00 | 1.00±0.00 | 1.02±0.00 | | 0.35±0.25 | 0.93±0.11 | 0.68±0.44 | 0.37±0.46 |
| Jittering | 0.57±0.00 | 1.03±0.00 | 1.00±0.00 | 1.02±0.00 | | 0.54±0.05 | 0.62±0.41 | 0.73±0.36 | 0.37±0.46 |
| Scaling | 0.57±0.00 | 1.03±0.00 | 0.91±0.13 | 1.02±0.00 | | **0.17±0.21** | 0.62±0.41 | 0.81±0.26 | 0.37±0.46 |
| TDA | **0.17±0.21** | **0.08±0.06** | 1.00±0.00 | 0.07±0.09 | | 0.25±0.24 | 0.62±0.41 | 0.68±0.45 | 0.37±0.46 |

| DICE | pen | door | relocate | hammer |
|---|---|---|---|---|
| NoAug. | 0.26±0.18 | 0.34±0.48 | 0.73±0.36 | 0.34±0.48 |
| OAT | 0.28±0.21 | 0.35±0.48 | 0.33±0.47 | **0.03±0.05** |
| VAE-MDP | 0.36±0.18 | 0.36±0.30 | **0.03±0.02** | 0.67±0.47 |
| TimeGAN | 0.44±0.12 | 0.60±0.25 | 0.11±0.08 | 0.68±0.48 |
| VAE | 0.31±0.24 | 0.50±0.35 | 0.35±0.46 | 0.65±0.46 |
| SPAWNER | 0.26±0.23 | 0.34±0.48 | 0.65±0.43 | 0.40±0.44 |
| DGW | 0.29±0.23 | 0.36±0.47 | 0.41±0.43 | 0.53±0.35 |
| Permutation | 0.50±0.10 | 0.94±0.12 | **0.02±0.02** | 0.40±0.44 |
| Jittering | 0.38±0.27 | **0.02±0.03** | 0.68±0.45 | 0.47±0.42 |
| Scaling | 0.50±0.10 | 0.34±0.48 | 0.73±0.36 | 0.40±0.44 |
| TDA | **0.08±0.06** | 0.60±0.25 | 0.37±0.45 | 0.68±0.48 |

Table 12: Regret@5 results of OPE without and with each augmentation method in resampled Adroit `cloned` environment. The data are randomly sampled from original training data as the same data points as the corresponding task in `human` environment. Results are obtained by averaging over 3 random seeds used for training at a discount factor of 0.995, with standard deviations shown after ±.

| FQE | pen | door | relocate | hammer | MB | pen | door | relocate | hammer |
|---|---|---|---|---|---|---|---|---|---|
| NoAug. | **0.00±0.00** | **0.00±0.00** | **0.00±0.00** | **0.00±0.01** | | **0.00±0.00** | **0.00±0.01** | 0.02±0.02 | 0.01±0.01 |
| OAT | **0.01±0.02** | **0.00±0.00** | **0.02±0.02** | 0.03±0.05 | | 0.01±0.02 | **0.00±0.00** | 0.02±0.02 | 0.01±0.01 |
| VAE-MDP | 0.07±0.08 | 0.08±0.12 | 0.34±0.45 | 0.03±0.05 | | **0.00±0.00** | 0.02±0.03 | **0.00±0.00** | 0.07±0.09 |
| TimeGAN | 0.06±0.08 | **0.00±0.01** | 0.36±0.43 | 0.07±0.09 | | 0.01±0.02 | **0.00±0.00** | **0.00±0.00** | **0.00±0.00** |
| VAE | 0.08±0.07 | **0.00±0.01** | 0.45±0.37 | 0.08±0.09 | | 0.01±0.02 | **0.00±0.00** | 0.02±0.02 | 0.01±0.01 |
| SPAWNER | 0.07±0.08 | 0.13±0.17 | 0.67±0.47 | 0.08±0.09 | | 0.05±0.06 | **0.00±0.00** | 0.07±0.10 | 0.03±0.05 |
| DGW | 0.31±0.24 | 0.23±0.24 | 0.87±0.10 | 0.16±0.16 | | 0.19±0.27 | **0.00±0.00** | 0.35±0.45 | **0.00±0.00** |
| Permutation | 0.38±0.27 | 0.09±0.11 | 0.15±0.21 | 0.16±0.16 | | 0.10±0.08 | **0.00±0.00** | 0.24±0.34 | **0.00±0.00** |
| Jittering | 0.07±0.08 | 0.08±0.12 | 0.17±0.20 | 0.16±0.16 | | 0.01±0.01 | **0.00±0.00** | 0.64±0.31 | 0.01±0.01 |
| Scaling | 0.35±0.25 | 0.02±0.03 | 0.39±0.30 | 0.16±0.16 | | 0.07±0.08 | 0.08±0.12 | **0.00±0.00** | **0.00±0.00** |
| TDA | 0.07±0.08 | 0.04±0.06 | 0.17±0.20 | 0.10±0.08 | | 0.01±0.02 | **0.00±0.00** | 0.09±0.09 | 0.14±0.08 |

| IS | pen | door | relocate | hammer | DR | pen | door | relocate | hammer |
|---|---|---|---|---|---|---|---|---|---|
| NoAug. | 0.18±0.00 | 0.25±0.00 | 0.63±0.44 | 0.13±0.18 | | **0.00±0.00** | **0.13±0.18** | 0.56±0.41 | **0.00±0.00** |
| OAT | **0.00±0.00** | **0.00±0.00** | **0.00±0.00** | **0.00±0.00** | | 0.05±0.06 | **0.13±0.18** | 0.24±0.34 | **0.00±0.00** |
| VAE-MDP | 0.01±0.01 | **0.00±0.01** | **0.00±0.00** | 0.14±0.18 | | 0.09±0.06 | 0.25±0.18 | **0.02±0.02** | 0.16±0.16 |
| TimeGAN | 0.02±0.02 | 0.16±0.12 | **0.00±0.00** | 0.01±0.01 | | 0.10±0.08 | 0.25±0.18 | 0.40±0.41 | 0.14±0.18 |
| VAE | 0.10±0.08 | 0.27±0.23 | **0.00±0.00** | 0.00±0.00 | | 0.09±0.06 | 0.19±0.26 | 0.32±0.46 | 0.14±0.18 |
| SPAWNER | 0.18±0.00 | 0.56±0.00 | 0.02±0.02 | 0.00±0.01 | | 0.06±0.08 | **0.13±0.18** | 0.26±0.33 | 0.14±0.18 |
| DGW | 0.18±0.00 | 0.56±0.00 | 0.97±0.00 | 0.39±0.00 | | **0.00±0.00** | 0.15±0.16 | 0.34±0.45 | 0.14±0.18 |
| Permutation | 0.18±0.00 | 0.56±0.00 | 0.97±0.00 | 0.39±0.00 | | 0.05±0.06 | 0.15±0.16 | 0.56±0.41 | 0.14±0.18 |
| Jittering | 0.18±0.00 | 0.56±0.00 | 0.97±0.00 | 0.39±0.00 | | 0.01±0.01 | **0.13±0.17** | 0.24±0.34 | 0.13±0.18 |
| Scaling | 0.18±0.00 | 0.56±0.00 | 0.11±0.08 | 0.39±0.00 | | 0.05±0.06 | **0.13±0.17** | 0.56±0.41 | 0.14±0.18 |
| TDA | 0.01±0.01 | 0.04±0.06 | 0.65±0.46 | **0.00±0.00** | | 0.05±0.06 | **0.13±0.17** | 0.34±0.45 | 0.16±0.16 |

| DICE | pen | door | relocate | hammer |
|---|---|---|---|---|
| NoAug. | 0.05±0.06 | 0.02±0.03 | **0.00±0.00** | 0.03±0.05 |
| OAT | **0.01±0.01** | 0.35±0.48 | 0.30±0.43 | **0.01±0.02** |
| VAE-MDP | 0.06±0.04 | 0.13±0.17 | **0.00±0.00** | 0.07±0.05 |
| TimeGAN | 0.11±0.07 | 0.19±0.26 | **0.00±0.00** | 0.08±0.09 |
| VAE | 0.03±0.04 | 0.13±0.18 | 0.15±0.21 | 0.07±0.05 |
| SPAWNER | **0.02±0.02** | 0.13±0.18 | 0.56±0.41 | 0.07±0.05 |
| DGW | **0.01±0.01** | **0.00±0.00** | 0.30±0.43 | 0.07±0.04 |
| Permutation | **0.00±0.00** | 0.13±0.18 | **0.00±0.00** | 0.07±0.05 |
| Jittering | 0.04±0.04 | **0.00±0.00** | 0.60±0.43 | 0.05±0.04 |
| Scaling | 0.03±0.04 | **0.00±0.00** | 0.32±0.46 | 0.07±0.05 |
| TDA | 0.06±0.04 | 0.23±0.24 | 0.30±0.43 | 0.07±0.05 |

Table 13: MAE results of OPE without and with each augmentation method in resampled Adroit `expert` environment. The data are randomly sampled from original training data as the same data points as the corresponding task in `human` environment. Results are obtained by averaging over 3 random seeds used for training at a discount factor of 0.995, with standard deviations shown after $\pm$.

| FQE | pen | door | relocate | hammer | MB | pen | door | relocate | hammer |
|---|---|---|---|---|---|---|---|---|---|
| NoAug. | 1101±47 | 1751±52 | 1729±557 | 2822±756 | | 2363±135 | **708±88** | 621±28 | 13110±535 |
| OAT | 1254±21 | 1213±45 | **419±30** | 3599±159 | | **1212±2** | 1233±3 | **444±2** | **3383±193** |
| VAE-MDP | **467±5** | **793±36** | 468±5 | **1983±45** | | 1217±1 | 746±10 | 474±1 | 9890±105 |
| TimeGAN | 2012±24 | 1008±37 | 994±11 | 6317±194 | | 3108±2 | 1027±1 | 1007±2 | 19197±1121 |
| VAE | 1250±12 | 903±14 | 1605±8 | 2039±43 | | 3109±1 | 912±6 | 1610±6 | 10031±210 |
| SPAWNER | 1247±11 | 1264±47 | 778±10 | 4738±157 | | 3108±1 | 2738±6 | 778±10 | 21218±915 |
| DGW | 1245±11 | 1343±29 | 762±8 | 7264±117 | | 3109±1 | 2885±6 | 762±8 | 36422±33 |
| Permutation | 1249±12 | 1270±28 | 763±8 | 6350±104 | | 3109±1 | 2725±6 | 763±8 | 31965±28 |
| Jittering | 1247±13 | 1259±43 | 1772±19 | 6267±158 | | 3108±2 | 2724±6 | 763±8 | 29471±76 |
| Scaling | 1247±11 | 1500±49 | 1495±15 | 5729±144 | | 2589±367 | 1478±13 | 762±8 | 27187±209 |
| TDA | 1251±12 | 1058±45 | 477±4 | 3420±186 | | **1217±2** | 1043±13 | 481±1 | 12350±1943 |
| **IS** | pen | door | relocate | hammer | DR | pen | door | relocate | hammer |
| NoAug. | 1881±23 | 1005±22 | 1863±43 | 3659±159 | | 1632±783 | 970±86 | 1837±52 | 3262±856 |
| OAT | 939±31 | 1089±3 | **1136±5** | 3354±64 | | **597±74** | 1044±51 | 2078±86 | 2989±614 |
| VAE-MDP | **676±24** | **871±23** | 1858±42 | **2225±110** | | 1260±299 | 886±71 | 1837±52 | **2494±901** |
| TimeGAN | 833±43 | 1054±44 | 1858±42 | 5929±83 | | 1299±452 | 972±7 | 1837±52 | 5029±1100 |
| VAE | 1579±38 | 1480±41 | 1741±46 | 2481±209 | | 2149±477 | **654±63** | 1734±77 | **2179±158** |
| SPAWNER | 1933±14 | 2400±6 | 1526±19 | 4128±6 | | 2384±309 | 1089±51 | 1529±19 | 3840±606 |
| DGW | 2793±9 | 2571±6 | 1496±16 | 6935±6 | | 3109±347 | 1172±53 | **1499±16** | 6556±643 |
| Permutation | 2675±12 | 2386±6 | 1497±16 | 5820±6 | | 2997±344 | 1081±61 | **1500±17** | 5380±780 |
| Jittering | 2675±14 | 2386±6 | 1499±17 | 5821±6 | | 3038±349 | 1081±50 | **1503±17** | 5465±701 |
| Scaling | 2683±12 | 2858±6 | 1498±16 | 5320±6 | | 3046±392 | 1288±75 | **1501±16** | 4909±707 |
| TDA | 733±27 | 985±18 | 2052±113 | 3542±138 | | 1325±398 | 963±89 | 2015±161 | 3141±902 |
| **DICE** | pen | door | relocate | hammer | | | | | |
| NoAug. | 3122±106 | 1250±21 | 2369±19 | 4171±47 | | | | | |
| OAT | **1224±73** | 1148±10 | **360±27** | 3884±46 | | | | | |
| VAE-MDP | **1146±53** | 812±6 | 473±3 | **1996±44** | | | | | |
| TimeGAN | 1943±94 | 570±10 | 694±5 | 6637±68 | | | | | |
| VAE | 3014±140 | **428±1** | 387±2 | 2021±38 | | | | | |
| SPAWNER | 3067±143 | 1294±7 | 1524±19 | 4985±30 | | | | | |
| DGW | 3103±37 | 1363±7 | 1492±16 | 7411±26 | | | | | |
| Permutation | 3108±63 | 1284±1 | 1495±16 | 6510±21 | | | | | |
| Jittering | 3104±57 | 1288±7 | 1495±16 | 6510±30 | | | | | |
| Scaling | 3097±52 | 1531±8 | 1494±16 | 5970±30 | | | | | |
| TDA | **1193±54** | 1087±7 | 479±2 | 3697±29 | | | | | |

Table 14: Rank correlation results of OPE without and with each augmentation method in resampled Adroit `expert` environment. The data are randomly sampled from original training data as the same data points as the corresponding task in `human` environment. Results are obtained by averaging over 3 random seeds used for training at a discount factor of 0.995, with standard deviations shown after ±.

| FQE | pen | door | relocate | hammer | MB | pen | door | relocate | hammer |
|---|---|---|---|---|---|---|---|---|---|
| NoAug. | **0.19±0.22** | 0.87±0.07 | -0.38±0.12 | 0.29±0.34 | | 0.24±0.30 | 0.74±0.15 | **0.86±0.05** | 0.06±0.35 |
| OAT | -0.22±0.33 | **0.97±0.03** | **0.37±0.87** | **0.36±0.90** | | 0.26±0.69 | **0.81±0.12** | **0.89±0.02** | **0.23±0.84** |
| VAE-MDP | -0.18±0.53 | 0.33±0.47 | -0.10±0.51 | -0.31±0.90 | | 0.29±0.71 | **0.86±0.00** | 0.30±0.73 | -0.27±0.41 |
| TimeGAN | -0.78±0.17 | **0.94±0.02** | -0.10±0.51 | -0.20±0.85 | | 0.07±0.75 | 0.36±0.75 | 0.30±0.73 | -0.25±0.56 |
| VAE | -0.50±0.35 | 0.63±0.25 | -0.47±0.35 | -0.44±0.73 | | 0.09±0.25 | 0.54±0.61 | -0.15±0.72 | -0.79±0.14 |
| SPAWNER | -0.82±0.10 | **0.96±0.04** | -0.22±0.50 | -0.31±0.91 | | -0.40±0.53 | 0.29±0.60 | -0.08±0.61 | -0.00±0.68 |
| DGW | -0.88±0.00 | 0.21±0.14 | -0.96±0.03 | -0.32±0.91 | | 0.06±0.53 | 0.19±0.62 | 0.27±0.33 | -0.01±0.72 |
| Permutation | -0.79±0.13 | 0.80±0.19 | -0.25±0.78 | -0.31±0.92 | | -0.29±0.70 | 0.13±0.75 | 0.14±0.74 | -0.65±0.32 |
| Jittering | -0.82±0.12 | **0.92±0.09** | -0.21±0.60 | -0.31±0.92 | | **0.58±0.09** | 0.23±0.83 | **0.93±0.05** | -0.40±0.66 |
| Scaling | -0.82±0.11 | 0.86±0.16 | -0.09±0.55 | -0.31±0.91 | | 0.22±0.79 | 0.77±0.29 | **0.92±0.07** | -0.37±0.55 |
| TDA | -0.68±0.12 | 0.80±0.25 | -0.06±0.80 | -0.31±0.92 | | **0.46±0.52** | 0.55±0.33 | 0.72±0.24 | 0.00±0.53 |

| IS | pen | door | relocate | hammer | DR | pen | door | relocate | hammer |
|---|---|---|---|---|---|---|---|---|---|
| NoAug. | -0.02±0.48 | 0.08±0.38 | **0.96±0.04** | 0.34±0.50 | | -0.85±0.03 | 0.32±0.18 | -0.28±0.75 | 0.33±0.85 |
| OAT | -0.59±0.11 | 0.43±0.35 | 0.84±0.08 | 0.97±0.02 | | -0.27±0.26 | 0.37±0.11 | **0.21±0.64** | **0.34±0.88** |
| VAE-MDP | -0.25±0.32 | 0.28±0.02 | **0.96±0.00** | **1.00±0.00** | | -0.83±0.08 | 0.41±0.31 | -0.28±0.75 | 0.15±0.63 |
| TimeGAN | **0.89±0.03** | **0.87±0.00** | **0.96±0.00** | 0.61±0.14 | | -0.82±0.05 | **0.42±0.10** | -0.28±0.75 | 0.15±0.80 |
| VAE | - | 0.19±0.04 | 0.75±0.00 | 0.49±0.45 | | -0.89±0.02 | 0.35±0.46 | -0.26±0.38 | 0.08±0.75 |
| SPAWNER | -0.89±0.00 | - | -0.04±0.64 | 0.10±0.14 | | -0.89±0.02 | 0.08±0.28 | 0.10±0.69 | 0.23±0.85 |
| DGW | -0.32±0.37 | - | -0.59±0.26 | - | | -0.88±0.03 | -0.16±0.28 | -0.55±0.18 | 0.22±0.84 |
| Permutation | -0.23±0.45 | - | 0.23±0.59 | - | | -0.90±0.02 | 0.19±0.16 | -0.11±0.27 | 0.18±0.83 |
| Jittering | 0.11±0.59 | - | 0.50±0.07 | - | | -0.90±0.01 | 0.05±0.20 | -0.75±0.16 | 0.22±0.84 |
| Scaling | -0.46±0.21 | - | 0.27±0.62 | 0.00±0.00 | | -0.91±0.02 | 0.32±0.07 | -0.52±0.16 | 0.19±0.84 |
| TDA | 0.31±0.00 | 0.59±0.35 | 0.90±0.11 | 0.80±0.14 | | -0.87±0.00 | 0.31±0.33 | 0.02±0.72 | 0.24±0.82 |

| DICE | pen | door | relocate | hammer |
|---|---|---|---|---|
| NoAug. | 0.23±0.39 | -0.33±0.89 | 0.11±0.62 | -0.06±0.73 |
| OAT | **0.89±0.01** | 0.18±0.81 | -0.22±0.01 | **-0.05±0.72** |
| VAE-MDP | **0.87±0.07** | -0.23±0.82 | -0.17±0.84 | -0.20±0.85 |
| TimeGAN | **0.85±0.05** | -0.07±0.61 | -0.17±0.84 | -0.24±0.86 |
| VAE | 0.65±0.19 | -0.16±0.58 | 0.25±0.67 | -0.11±0.59 |
| SPAWNER | 0.55±0.50 | 0.11±0.74 | -0.16±0.67 | -0.10±0.78 |
| DGW | **0.86±0.06** | 0.16±0.77 | 0.18±0.84 | -0.07±0.75 |
| Permutation | **0.93±0.00** | 0.13±0.73 | **0.81±0.21** | -0.21±0.84 |
| Jittering | 0.72±0.19 | **0.28±0.85** | 0.21±0.83 | -0.12±0.79 |
| Scaling | 0.81±0.14 | **0.25±0.81** | 0.15±0.80 | -0.11±0.79 |
| TDA | 0.76±0.08 | 0.17±0.80 | 0.23±0.57 | -0.26±0.89 |

Table 15: Regret@1 results of OPE without and with each augmentation method in resampled Adroit `expert` environment. The data are randomly sampled from original training data as the same data points as the corresponding task in `human` environment. Results are obtained by averaging over 3 random seeds used for training at a discount factor of 0.995, with standard deviations shown after ±.

| FQE | pen | door | relocate | hammer | MB | pen | door | relocate | hammer |
|---|---|---|---|---|---|---|---|---|---|
| NoAug. | **0.17±0.14** | 0.05±0.06 | 0.91±0.13 | **0.05±0.04** | | 0.22±0.25 | **0.07±0.04** | 0.15±0.10 | 0.34±0.18 |
| OAT | **0.19±0.27** | **0.01±0.01** | **0.33±0.47** | 0.34±0.48 | | 0.22±0.19 | **0.04±0.03** | **0.00±0.00** | 0.33±0.46 |
| VAE-MDP | 0.26±0.23 | 0.35±0.48 | 0.68±0.45 | 0.68±0.48 | | 0.17±0.21 | **0.01±0.01** | 0.41±0.43 | 0.91±0.09 |
| TimeGAN | 0.54±0.05 | 0.02±0.03 | 0.68±0.45 | 0.60±0.43 | | 0.24±0.24 | 0.34±0.48 | 0.41±0.43 | 0.73±0.26 |
| VAE | 0.38±0.27 | 0.30±0.33 | 1.00±0.01 | 0.60±0.43 | | 0.29±0.00 | 0.26±0.36 | 0.66±0.47 | 0.94±0.11 |
| SPAWNER | 0.46±0.09 | 0.01±0.01 | 0.66±0.47 | 0.68±0.48 | | 0.41±0.17 | 0.60±0.43 | 0.57±0.42 | **0.20±0.16** |
| DGW | 0.46±0.09 | 0.19±0.26 | 1.00±0.01 | 0.68±0.48 | | 0.20±0.19 | 0.34±0.48 | 0.41±0.43 | 0.34±0.48 |
| Permutation | 0.37±0.16 | 0.05±0.06 | 0.67±0.47 | 0.68±0.48 | | 0.36±0.23 | 0.67±0.48 | 0.57±0.42 | 0.94±0.11 |
| Jittering | 0.46±0.09 | **0.01±0.01** | 0.68±0.45 | 0.68±0.48 | | **0.07±0.08** | 0.34±0.48 | 0.16±0.08 | 0.67±0.46 |
| Scaling | 0.50±0.10 | **0.01±0.01** | 0.68±0.45 | 0.68±0.48 | | 0.16±0.22 | **0.00±0.01** | **0.02±0.02** | 0.74±0.39 |
| TDA | 0.37±0.16 | **0.01±0.01** | 0.68±0.45 | 0.68±0.48 | | 0.18±0.14 | 0.34±0.48 | 0.41±0.43 | 0.52±0.41 |

| IS | pen | door | relocate | hammer | DR | pen | door | relocate | hammer |
|---|---|---|---|---|---|---|---|---|---|
| NoAug. | 0.14±0.03 | 0.94±0.12 | **0.00±0.00** | 0.05±0.04 | | **0.37±0.16** | **0.00±0.00** | 0.68±0.44 | 0.34±0.48 |
| OAT | 0.46±0.09 | **0.00±0.01** | **0.02±0.02** | **0.00±0.00** | | 0.43±0.05 | **0.00±0.00** | **0.07±0.10** | 0.34±0.48 |
| VAE-MDP | 0.19±0.13 | 0.39±0.45 | 0.67±0.47 | **0.00±0.00** | | **0.37±0.16** | 0.00±0.01 | 0.68±0.44 | **0.20±0.15** |
| TimeGAN | **0.01±0.00** | 0.69±0.48 | 0.67±0.47 | 0.05±0.04 | | 0.46±0.09 | 0.02±0.03 | 0.68±0.44 | 0.38±0.45 |
| VAE | 0.57±0.00 | 0.38±0.46 | 0.82±0.26 | 0.08±0.09 | | 0.46±0.09 | 0.34±0.48 | 0.37±0.45 | 0.40±0.44 |
| SPAWNER | 0.44±0.19 | 1.03±0.00 | 0.40±0.41 | 0.20±0.15 | | 0.46±0.09 | 0.34±0.48 | 0.37±0.45 | 0.34±0.48 |
| DGW | 0.29±0.23 | 1.03±0.00 | 0.81±0.26 | 1.02±0.00 | | **0.37±0.16** | 0.67±0.48 | 1.00±0.01 | 0.35±0.47 |
| Permutation | 0.14±0.15 | 1.03±0.00 | 0.24±0.34 | 1.02±0.00 | | **0.37±0.16** | 0.67±0.48 | 1.00±0.01 | 0.41±0.44 |
| Jittering | 0.17±0.21 | 1.03±0.00 | 0.17±0.20 | 1.02±0.00 | | **0.37±0.16** | 0.34±0.48 | 1.00±0.01 | 0.40±0.44 |
| Scaling | 0.36±0.18 | 1.03±0.00 | 0.33±0.47 | 0.86±0.23 | | 0.46±0.09 | **0.00±0.00** | 1.00±0.01 | 0.41±0.44 |
| TDA | 0.57±0.00 | **0.01±0.01** | **0.02±0.02** | **0.01±0.01** | | **0.37±0.16** | **0.00±0.00** | 0.37±0.45 | 0.35±0.47 |

| DICE | pen | door | relocate | hammer |
|---|---|---|---|---|
| NoAug. | 0.20±0.26 | 0.69±0.48 | **0.30±0.43** | 0.67±0.47 |
| OAT | 0.02±0.01 | **0.34±0.48** | 1.00±0.00 | 0.67±0.47 |
| VAE-MDP | 0.01±0.01 | 0.68±0.48 | 0.64±0.45 | 0.66±0.46 |
| TimeGAN | **0.00±0.00** | 0.42±0.44 | 0.64±0.45 | 0.67±0.47 |
| VAE | 0.02±0.01 | 0.76±0.36 | 0.58±0.42 | **0.40±0.41** |
| SPAWNER | 0.06±0.05 | 0.42±0.44 | 0.66±0.46 | 0.67±0.47 |
| DGW | 0.02±0.01 | **0.34±0.48** | 0.58±0.42 | 0.67±0.47 |
| Permutation | 0.01±0.01 | **0.34±0.48** | **0.31±0.30** | 0.67±0.47 |
| Jittering | 0.11±0.13 | **0.34±0.48** | 0.58±0.42 | 0.65±0.46 |
| Scaling | 0.02±0.01 | **0.34±0.48** | 0.58±0.42 | 0.67±0.47 |
| TDA | 0.03±0.00 | **0.34±0.48** | 0.68±0.45 | 0.67±0.47 |

Table 16: Regret@5 results of OPE without and with each augmentation method in resampled Adroit `expert` environment. The data are randomly sampled from original training data as the same data points as the corresponding task in `human` environment. Results are obtained by averaging over 3 random seeds used for training at a discount factor of 0.995, with standard deviations shown after ±.

| FQE | pen | door | relocate | hammer | MB | pen | door | relocate | hammer |
|---|---|---|---|---|---|---|---|---|---|
| NoAug. | **0.02±0.01** | **0.00±0.01** | 0.24±0.34 | **0.00±0.01** | | **0.01±0.01** | **0.00±0.01** | **0.00±0.00** | 0.07±0.09 |
| OAT | 0.05±0.06 | **0.00±0.00** | 0.24±0.34 | **0.03±0.05** | | 0.06±0.08 | **0.00±0.00** | **0.00±0.00** | 0.03±0.05 |
| VAE-MDP | 0.10±0.06 | **0.00±0.01** | **0.17±0.20** | 0.13±0.09 | | 0.06±0.08 | **0.00±0.00** | 0.02±0.02 | 0.03±0.00 |
| TimeGAN | 0.08±0.06 | **0.00±0.00** | **0.17±0.20** | 0.10±0.08 | | 0.07±0.08 | **0.00±0.00** | 0.02±0.02 | 0.13±0.09 |
| VAE | **0.01±0.02** | **0.00±0.01** | 0.45±0.37 | 0.13±0.09 | | 0.05±0.06 | **0.00±0.01** | 0.45±0.37 | 0.29±0.14 |
| SPAWNER | 0.12±0.06 | **0.00±0.00** | 0.26±0.33 | 0.13±0.09 | | 0.09±0.07 | 0.02±0.03 | 0.09±0.09 | 0.07±0.09 |
| DGW | 0.14±0.03 | **0.00±0.01** | 0.95±0.03 | 0.16±0.16 | | 0.06±0.08 | **0.00±0.00** | 0.07±0.10 | 0.13±0.18 |
| Permutation | 0.14±0.03 | **0.00±0.01** | 0.45±0.37 | 0.13±0.09 | | 0.08±0.06 | 0.19±0.26 | 0.24±0.34 | 0.20±0.16 |
| Jittering | 0.08±0.06 | **0.00±0.01** | **0.17±0.20** | 0.13±0.09 | | **0.00±0.00** | 0.19±0.26 | **0.00±0.00** | 0.07±0.05 |
| Scaling | 0.08±0.06 | **0.00±0.01** | **0.17±0.20** | 0.13±0.09 | | 0.06±0.08 | **0.00±0.00** | **0.00±0.00** | 0.08±0.08 |
| TDA | 0.09±0.04 | **0.00±0.01** | 0.24±0.34 | 0.13±0.09 | | **0.00±0.00** | **0.00±0.00** | **0.00±0.00** | **0.01±0.02** |

| IS | pen | door | relocate | hammer | DR | pen | door | relocate | hammer |
|---|---|---|---|---|---|---|---|---|---|
| NoAug. | 0.02±0.02 | 0.23±0.24 | **0.00±0.00** | 0.03±0.05 | | 0.14±0.03 | **0.00±0.00** | 0.56±0.41 | 0.13±0.18 |
| OAT | 0.07±0.03 | **0.00±0.01** | **0.00±0.00** | **0.00±0.00** | | **0.05±0.06** | **0.00±0.00** | **0.07±0.10** | 0.13±0.18 |
| VAE-MDP | 0.04±0.04 | 0.23±0.24 | 0.65±0.46 | **0.00±0.00** | | 0.14±0.03 | **0.00±0.00** | 0.56±0.41 | **0.07±0.05** |
| TimeGAN | **0.00±0.00** | 0.37±0.26 | 0.65±0.46 | **0.00±0.00** | | 0.11±0.07 | **0.00±0.00** | 0.56±0.41 | 0.13±0.18 |
| VAE | 0.18±0.00 | 0.21±0.25 | 0.79±0.25 | 0.01±0.01 | | 0.14±0.03 | **0.00±0.00** | 0.27±0.32 | 0.13±0.18 |
| SPAWNER | 0.06±0.08 | 0.56±0.00 | 0.15±0.10 | 0.39±0.00 | | 0.14±0.03 | **0.00±0.00** | 0.24±0.34 | 0.13±0.18 |
| DGW | 0.06±0.08 | 0.56±0.00 | 0.71±0.21 | 0.39±0.00 | | 0.14±0.03 | **0.00±0.01** | 0.58±0.39 | 0.13±0.18 |
| Permutation | 0.05±0.03 | 0.56±0.00 | 0.15±0.21 | 0.39±0.00 | | 0.14±0.03 | **0.00±0.00** | **0.03±0.02** | 0.13±0.18 |
| Jittering | 0.04±0.04 | 0.56±0.00 | 0.02±0.02 | 0.39±0.00 | | 0.14±0.03 | **0.00±0.00** | 0.70±0.34 | 0.13±0.18 |
| Scaling | 0.13±0.07 | 0.56±0.00 | 0.15±0.21 | 0.27±0.17 | | 0.14±0.03 | **0.00±0.00** | 0.41±0.40 | 0.13±0.18 |
| TDA | 0.13±0.07 | **0.00±0.00** | **0.00±0.00** | **0.00±0.00** | | 0.14±0.03 | **0.00±0.00** | 0.30±0.43 | 0.13±0.18 |

| DICE | pen | door | relocate | hammer |
|---|---|---|---|---|
| NoAug. | **0.00±0.00** | 0.37±0.26 | 0.15±0.21 | **0.04±0.04** |
| OAT | **0.00±0.00** | **0.19±0.26** | 0.30±0.43 | **0.04±0.04** |
| VAE-MDP | **0.00±0.00** | **0.19±0.26** | 0.05±0.00 | 0.14±0.18 |
| TimeGAN | **0.00±0.00** | 0.21±0.25 | 0.54±0.39 | 0.07±0.05 |
| VAE | **0.00±0.00** | **0.17±0.16** | 0.17±0.20 | 0.08±0.08 |
| SPAWNER | 0.01±0.01 | **0.19±0.26** | 0.56±0.41 | 0.05±0.04 |
| DGW | **0.00±0.00** | **0.19±0.26** | 0.32±0.46 | 0.05±0.04 |
| Permutation | **0.00±0.00** | **0.19±0.26** | **0.02±0.02** | 0.07±0.05 |
| Jittering | **0.00±0.00** | **0.19±0.26** | 0.32±0.46 | 0.05±0.04 |
| Scaling | **0.00±0.00** | **0.19±0.26** | 0.34±0.45 | **0.04±0.04** |
| TDA | **0.00±0.00** | **0.19±0.26** | **0.02±0.02** | 0.14±0.18 |

Table 17: PST information in Adroit `pen-human`.

| trajectory index | trajectory length | PTS start time | PST end time |
|---|---|---|---|
| 0 | 200 | 1 | 7 |
| 1 | 200 | 1 | 7 |
| 2 | 200 | 1 | 7 |
| 3 | 200 | 0 | 6 |
| 4 | 200 | 1 | 7 |
| 5 | 200 | 2 | 8 |
| 6 | 200 | 2 | 8 |
| 7 | 200 | 1 | 7 |
| 8 | 200 | 1 | 7 |
| 9 | 200 | 1 | 7 |
| 10 | 200 | 1 | 7 |
| 11 | 200 | 1 | 7 |
| 12 | 200 | 1 | 7 |
| 13 | 200 | 0 | 6 |
| 14 | 200 | 0 | 6 |
| 15 | 200 | 1 | 7 |
| 16 | 200 | 1 | 7 |
| 17 | 200 | 1 | 7 |
| 18 | 200 | 2 | 8 |
| 19 | 200 | 1 | 7 |
| 20 | 200 | 1 | 7 |
| 21 | 200 | 0 | 6 |
| 22 | 200 | 1 | 7 |
| 23 | 200 | 1 | 7 |
| 24 | 200 | 1 | 7 |

Table 18: PST information in Adroit `door-human`.

| trajectory index | trajectory length | PTS start time | PST end time |
|---|---|---|---|
| 0 | 300 | 204 | 217 |
| 1 | 300 | 238 | 251 |
| 2 | 300 | 202 | 215 |
| 3 | 300 | 210 | 223 |
| 4 | 300 | 189 | 202 |
| 5 | 293 | 189 | 202 |
| 6 | 284 | 183 | 196 |
| 7 | 265 | 161 | 174 |
| 8 | 265 | 153 | 166 |
| 9 | 291 | 167 | 180 |
| 10 | 247 | 151 | 164 |
| 11 | 236 | 136 | 149 |
| 12 | 271 | 165 | 178 |
| 13 | 242 | 144 | 157 |
| 14 | 277 | 157 | 170 |
| 15 | 257 | 158 | 171 |
| 16 | 260 | 166 | 179 |
| 17 | 223 | 134 | 147 |
| 18 | 257 | 164 | 177 |
| 19 | 277 | 146 | 159 |
| 20 | 288 | 189 | 202 |
| 21 | 289 | 187 | 200 |
| 22 | 256 | 156 | 169 |
| 23 | 226 | 146 | 159 |
| 24 | 225 | 135 | 148 |

## C.2 START AND END TIME OF PSTS.

We present that the length of trajectories, and start and end time of found PSTs on each trajectory in Adroit `human` can be varied, as shown in Tables 17, 18, 19, & 20. It can be observed that PST can start at different time steps across trajectories, which resolves the non-temporally-aligned cases.

Table 19: PST information in Adroit `relocate-human`.

| trajectory index | trajectory length | PTS start time | PST end time |
| --- | --- | --- | --- |
| 0 | 352 | 19 | 26 |
| 1 | 437 | 24 | 31 |
| 2 | 360 | 52 | 59 |
| 3 | 402 | 30 | 37 |
| 4 | 346 | 33 | 40 |
| 5 | 527 | 59 | 66 |
| 6 | 412 | 28 | 35 |
| 7 | 416 | 40 | 47 |
| 8 | 477 | 37 | 44 |
| 9 | 382 | 32 | 39 |
| 10 | 385 | 24 | 31 |
| 11 | 523 | 27 | 34 |
| 12 | 451 | 33 | 40 |
| 13 | 382 | 33 | 40 |
| 14 | 357 | 31 | 38 |
| 15 | 375 | 39 | 46 |
| 16 | 439 | 36 | 43 |
| 17 | 378 | 25 | 32 |
| 18 | 377 | 31 | 38 |
| 19 | 336 | 42 | 49 |
| 20 | 512 | 36 | 43 |
| 21 | 333 | 32 | 39 |
| 22 | 297 | 44 | 51 |
| 23 | 358 | 36 | 43 |
| 24 | 328 | 31 | 38 |

Table 20: PST information in Adroit `hammer-human`.

| trajectory index | trajectory length | PTS start time | PST end time |
| --- | --- | --- | --- |
| 0 | 537 | 153 | 161 |
| 1 | 399 | 100 | 108 |
| 2 | 472 | 116 | 124 |
| 3 | 624 | 115 | 123 |
| 4 | 469 | 153 | 161 |
| 5 | 418 | 137 | 145 |
| 6 | 456 | 110 | 118 |
| 7 | 404 | 224 | 232 |
| 8 | 364 | 123 | 131 |
| 9 | 529 | 120 | 128 |
| 10 | 455 | 131 | 139 |
| 11 | 371 | 108 | 116 |
| 12 | 496 | 180 | 188 |
| 13 | 433 | 140 | 148 |
| 14 | 484 | 129 | 137 |
| 15 | 398 | 102 | 110 |
| 16 | 418 | 112 | 120 |
| 17 | 621 | 226 | 234 |
| 18 | 467 | 121 | 129 |
| 19 | 390 | 104 | 112 |
| 20 | 348 | 105 | 113 |
| 21 | 464 | 86 | 94 |
| 22 | 425 | 89 | 97 |
| 23 | 506 | 93 | 101 |
| 24 | 362 | 82 | 90 |

## D    REAL-WORLD SEPSIS TREATMENT

Sepsis, which is defined as life-threatening organ dysfunction in response to infection, is the leading cause of mortality and the most expensive condition associated with in-hospital stay (Liu et al., 2014; Gao et al., 2022b). In particular, septic shock, which is the most advanced complication of sepsis due to severe abnormalities of circulation and/or cellular metabolism (Bone et al., 1992), reaches a mortality rate as high as 50% (Martin et al., 2003). It is critical to find an effective policy that can be followed to prevent septic shock and recover from sepsis.

### D.1    TASK DETAILS

**Labels.** The hospital provided the EHRs over two years, including 221,700 visits with 35 static variables such as gender, age, and past medical condition, and 43 temporal variables including vital signs, lab analytes, and treatments. Our study population is patients with a suspected infection which was identified by the administration of any type of antibiotic, antiviral, antibacterial, antiparasitic, or antifungal, or a positive test result of PCR (Point of Care Rapid). On the basis of the Third International Consensus Definitions for Sepsis and Septic Shock (Singer et al., 2016), our medical experts identified septic shock as any of the following conditions are met:

- Persistent hypertension as shown through two consecutive readings ($\leq$ 30 minutes apart). Systolic Blood Pressure (SBP) < 90 mmHg Mean Arterial Pressure (MAP) < 65 mmHg Decrease in SBP $\geq$ 40 mmHg with an 8-hour period

- Any vasopressor administration.

From the EHRs, 3,499 septic shock positive and 81,398 negative visits were identified based on the intersection of the expert sepsis diagnostic rules and International Codes for Disease 9th division (ICD-9); the 36,122 visits with mismatched labels between the expert rule and the ICD-9 were excluded in our study. 2,205 shock visits were obtained by excluding the visits admitted with septic shock and the long-stay visits and then we did the stratified random sampling from non-shock visits, keeping the same distribution of age, gender, ethnicity, and length of hospital stay. The final data constituted 4,410 visits with an equal ratio of shock and non-shock visits.

**States.** To approximate patient observations, 15 sepsis-related attributes were selected based on the sepsis diagnostic rules. In our data, the average missing rate across the 15 sepsis-related attributes was 78.6%. We avoided deleting sparse attributes or resampling with a regular time interval because the attributes suggested by medical experts are critical to decision making for sepsis treatment, and the temporal missing patterns of EHRs also provide the information of patient observations. The missing values were imputed using Temporal Belief Memory (Kim & Chi, 2018) combined with missing indicators (Lipton et al., 2016).

**Actions.** For actions, we considered two medical treatments: antibiotic administration and oxygen assistance. Note that the two treatments can be applied simultaneously, which results in a total of four actions. Generally, the treatments are mixed in discrete and continuous action spaces according to their granularity. For example, a decision of whether a certain drug is administrated is discrete, while the dosage of drug is continuous. Continuous action space has been mainly handled by policy-based RL models such as actor-critic models (Lillicrap et al., 2015), and it is generally only available for online RL. Since we cannot search continuous action spaces while online interacting with actual patients, we focus on discrete actions. Moreover, in this work, the RL agent aims to let the physicians know when and which treatment should be given to a patient, rather than suggests an optimal amount of drugs or duration of oxygen control that requires more complex consideration.

**Rewards.** Two leading clinicians, both with over 20-year experience on the subject of sepsis, guided to define the reward function based on the severity of septic stages. The rewards were defined as follows: infection [-5], inflammation [-10], organ failures [-20], and septic shock [-50]. Whenever a patient was recovered from any stage of them, the positive reward for the stage was gained back.

The data was divided into 80% (the earlier 80% according to the time of the first event recorded in patients' visits) for training and (the later) 20% for test, following the common practice while splitting up time-series for training and testing (Campos et al., 2014).

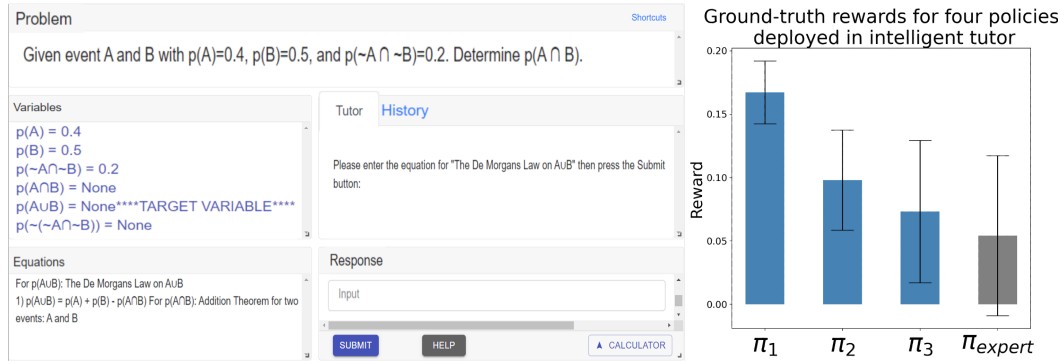

Figure 14: Our intelligent tutor GUI (left) and empirical results with three RL-induced policies and one expert policy (right).

**Policies** We assume that the clinical care team is well-trained with sufficient medical knowledge and follows standard protocols in sepsis treatments, thus we consider the behavioral policy, parameterized through behavior cloning (Azizsoltani & Jin, 2019), that generates the trajectories above as an expert policy. We estimate the behavior policy with behavior cloning as in (Fu et al., 2021; Hanna et al., 2019). The evaluation policies were trained using off-policy DQN algorithm with different hyper-parameter settings, where DQN was trained using default setting (learning rate $1e-3$, $\gamma = 0.99$), learning rate $1e-4$, learning rate $1e-5$, a different random seed, $\gamma = 0.9$, respectively.

**Evaluate Performance of Target Policies** Since the RL agent cannot directly interact with patients, it only depends on offline data for both policy induction and evaluation. In similar fashion to prior studies (Komorowski et al., 2018; Azizsoltani & Jin, 2019; Raghu et al., 2017), the induced policies were evaluated using the septic shock rate. And the OPE validation metric, rank correlation, can be calculated by comparing the ranking by OPE estimations versus the rankings of septic shock rate of target policies. The assumption behind that is (Raghu et al., 2017): when a septic shock prevention policy is indeed effective, the more the real treatments in a patient trajectory agree with the induced policy, the lower the chance the patient would get into septic shock; vice versa, the less the real treatments in a patient trajectory agree with the induced policy (more dissimilar), the higher the chance the patient would get into septic shock. Specifically, we follow the recent design by (Ju et al., 2021): We measured agreement rate with the agent policy, which is the number of actions of a target policy agreed with the agent policy among the total number of actions in a trajectory. Then we sort the trajectories by their similarity rate in ascending order and calculate the septic shock rate for the top $10\%$ of trajectories with the highest similarity rate. If the agent policies are indeed effective, the more the actually executed treatments agree with the agent policy, the less likely the patient is going to have septic shock.

# E    REAL-WORLD INTELLIGENT TUTORING

## E.1    TASK DETAILS

Our data contains a total of 1,307 students' interaction logs with a web-based ITS collected over seven semesters' classroom studies. The ITS is used in an undergraduate STEM course at a college, which has been extensively used by over $2,000$ students with $\sim$800k recorded interaction logs through eight academic years. The ITS is designed to teach entry-level undergraduate students with ten major probability principles, including complement theorem, Bayes' rule, etc. The GUI of the ITS is provided in Figure 14.

**States.** During tutoring, there are many factors that might determine or indicate students' learning state, but many of them are not well understood by educators. Thus, to be conservative, we extract varieties of attributes that might determine or indicate student learning observations from student-system interaction logs. In sum, 142 attributes with both discrete and continuous values are extracted, which can be categorized into the following five groups:

(i) **Autonomy (10 features)**: the amount of work done by the student, such as the number of times the student restarted a problem;

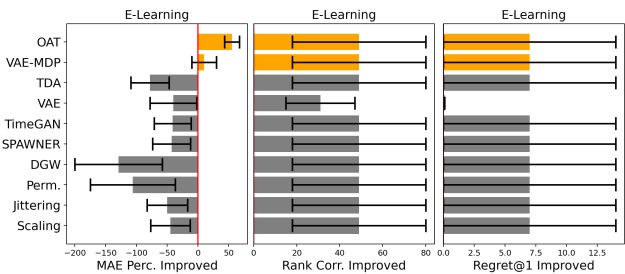

Figure 15: OPE improvement results averaging across OPE methods in e-learning.

(ii) **Temporal Situation (29 features)**: the time-related information about the work process, such as average time per step;

(iii) **Problem-Solving (35 features)**: information about the current problem-solving context, such as problem difficulty;

(iv) **Performance (57 features)**: information about the student's performance during problem-solving, such as percentage of correct entries;

(v) **Hints (11 features)**: information about the student's hint usage, such as the total number of hints requested.

**Actions.** For each problem, the ITS agent will decide whether the student should *solve* the next problem, *study* a solution provided by the tutor or *work together* with the tutor to solve on the problem. For each problem, the agent makes two levels of granularity: problem first and then step. For problem level, it first decides whether the next problem should be a worked example (WE), problem solving (PS), or a collaborative problem solving worked example (CPS). In WEs, students observe how the tutor solves a problem; in PSs, students solve the problem themselves; in CPSs, the students and the tutor co-construct the solution. If a CPS is selected, the tutor will then make step-level decisions on whether to elicit the next step from the student or to tell the solution step to the student directly.

**Rewards.** There was no immediate reward but the empirical evaluation matrix (i.e., delayed reward), which was the students' Normalized Learning Gain (NLG). NLG measured students' learning gain irrespective of their incoming competence. NLG is defined as: $NLG = \frac{score_{posttest} - score_{pretest}}{\sqrt{1 - score_{pretest}}}$, where 1 denotes the maximum score for both pre- and post-test that were taken before and after usage of the ITS, respectively.

**Policies.** The study were conducted across seven semesters, where the first six semesters' data were collected over expert policy and the seventh semester's data were collected over four different policies (three policies were RL-induced policies and one was the expert policy). The expert policy randomly picked actions. The three RL-induced policies were trained using off-policy DQN algorithm with different learning rates $lr = \{1e - 3, 1e - 4, 1e - 5\}$.

**Evaluate Performance of Target Policies.** Target policies are randomly assigned to 140 students who take the Probability course in one semester. During the studies, all students used the same tutor, followed the same general procedure, studied the same training materials, and worked through the same training problems. All students went through the same four phases: 1) reading textbook, 2) pre-test, 3) working on the ITS, and 4) post-test. During reading textbook, students read a general description of each principle, reviewed examples, and solved some training problems to get familiar with the ITS. Then the students took a pre-test which contained a total of 14 single- and multiple-principle problems. Students were not given feedback on their answers, nor were they allowed to go back to earlier questions (so as the post-test). Next, students worked on the ITS, where they received the 12 problems on ITS in the same order. After that, students took the 20-problem post-test, where 14 of the problems were isomorphic to the pre-test and the remainders were non-isomorphic multiple-principle problems. Tests were auto-graded following the same grading criteria. Test scores were normalized to the range of [0, 1].

|  | State Dim. | Action Dim. | Early Term. | Continuous Ctrl. | Dataset | Dataset Size |
|---|---|---|---|---|---|---|
| Halfcheetah | 17 | 6 | No | Yes | random | 999,000 |
|  |  |  |  |  | medium-replay | 201,798 |
|  |  |  |  |  | medium | 999,000 |
|  |  |  |  |  | medium-expert | 1,998,000 |
|  |  |  |  |  | expert | 999,000 |
| Hopper | 11 | 3 | Yes | Yes | random | 999,999 |
|  |  |  |  |  | medium-replay | 401,598 |
|  |  |  |  |  | medium | 999,998 |
|  |  |  |  |  | medium-expert | 1,998,966 |
|  |  |  |  |  | expert | 999,061 |
| Walker2d | 17 | 6 | Yes | Yes | random | 999,999 |
|  |  |  |  |  | medium-replay | 301,698 |
|  |  |  |  |  | medium | 999,322 |
|  |  |  |  |  | medium-expert | 1,998,318 |
|  |  |  |  |  | expert | 999,000 |
| Ant | 27 | 8 | Yes | Yes | random | 999,427 |
|  |  |  |  |  | medium-replay | 301,698 |
|  |  |  |  |  | medium | 999,175 |
|  |  |  |  |  | medium-expert | 1,998,158 |
|  |  |  |  |  | expert | 999,036 |

Table 21: Summary of the Gym-Mujoco environments and datasets used to train OAT and baselines.

## F ADDITIONAL EXPERIMENTS AND RESULTS: GYM-MUJOCO

### F.1 DETAILS OF GYM-MUJOCO

A total of 4 environments are provided by Gym-Mujoco, and we follow the guidelines from DOPE benchmark to validate our work and baselines (Fu et al., 2020). Moreover, each environment is provided with 5 training datasets collected using different behavioral policies, resulting in a total of 20 sets of tasks. DOPE also provides 11 target policies for each environment, whose performance are to be evaluated by the OPE methods. Though the Gym-Mujoco environments may not fit our major interests, *i.e., human-involved tasks*, we provide additional validation of our work on them, given they are popular testbeds and may be interested by readers. Table 21 shows summary of the Gym-Mujoco environments and datasets.

### F.2 RESULTS ON GYM-MUJOCO AND DISCUSSIONS

The Table 22 presents summary results on Gym-Mujoco environments using OAT and augmentation baselines. The results are obtained by calculating the improved percentage (for MAE), or distance (for Rank Correlation, Regret@1, and Regret@5) after augmentation compared to the original OPE

| | Halfcheetah | | | | Hopper | | | |
| --- | --- | --- | --- | --- | --- | --- | --- | --- |
| | MAE Perc. | Rank Corr. | Regret@1 | Regret@5 | MAE Perc. | Rank Corr. | Regret@1 | Regret@5 |
| OAT | **25.11** | **0.10** | **0.08** | **0.01** | **5.60** | **0.17** | **0.05** | **0.05** |
| VAE-MDP | 19.01 | 0.06 | 0.01 | **0.01** | 1.80 | 0.16 | -0.08 | 0.04 |
| TDA | 15.86 | 0.04 | -0.01 | 0.00 | 0.62 | 0.04 | -0.11 | 0.04 |
| Permutation | 10.14 | 0.03 | 0.00 | 0.00 | -3.72 | 0.01 | -0.25 | 0.00 |
| Jittering | 12.32 | 0.00 | -0.02 | **0.01** | -3.34 | 0.00 | -0.33 | 0.01 |
| Scaling | 12.08 | 0.01 | -0.02 | 0.00 | -4.13 | 0.02 | -0.27 | 0.00 |
| VAE | 18.72 | 0.00 | -0.02 | -0.01 | -3.21 | 0.14 | -0.08 | 0.04 |
| TimeGAN | 17.56 | -0.02 | -0.08 | 0.00 | -1.19 | 0.15 | -0.08 | 0.04 |
| SPAWNER | 8.06 | -0.11 | -0.15 | -0.03 | -10.21 | -0.02 | -0.16 | -0.12 |
| DGW | 7.22 | -0.12 | -0.11 | -0.07 | -11.38 | -0.07 | -0.18 | -0.15 |
| | Walker2d | | | | Ant | | | |
| | MAE Perc. | Rank Corr. | Regret@1 | Regret@5 | MAE Perc. | Rank Corr. | Regret@1 | Regret@5 |
| OAT | **3.39** | **0.15** | **0.06** | **0.03** | **1.50** | **0.27** | **0.20** | **0.01** |
| VAE-MDP | 3.37 | 0.01 | 0.00 | 0.02 | 1.26 | 0.15 | 0.06 | **0.01** |
| TDA | 0.00 | 0.04 | -0.04 | **0.03** | 0.42 | 0.04 | -0.03 | 0.00 |
| Permutation | -5.39 | 0.03 | -0.05 | 0.02 | -0.13 | 0.05 | 0.00 | **0.01** |
| Jittering | -5.62 | -0.01 | -0.08 | 0.00 | -0.11 | 0.02 | -0.04 | -0.02 |
| Scaling | -6.37 | 0.04 | -0.06 | 0.01 | -0.17 | 0.00 | -0.02 | 0.00 |
| VAE | -1.12 | -0.07 | -0.11 | 0.00 | -0.22 | 0.10 | -0.10 | -0.02 |
| TimeGAN | -3.45 | -0.03 | -0.12 | 0.01 | -0.30 | 0.08 | 0.01 | **0.01** |
| SPAWNER | -14.91 | -0.13 | -0.14 | -0.02 | -1.36 | -0.19 | -0.15 | -0.04 |
| DGW | -12.67 | -0.10 | -0.15 | 0.00 | -2.24 | -0.20 | -0.22 | -0.02 |

Table 22: Summary results of **averaged improvements** on OPE by OAT and baselines in the Gym-Mujoco environments and datasets.

results without augmentation. OAT can outperforms the baselines in terms of all four evaluation metrics (as bold in Table 22).

# G MORE RELATED WORKS

**OPE** In real-world, deploying and evaluating RL policies online are high stakes in such domains, as a poor policy can be fatal to humans. It's thus crucial to propose effective OPE methods. OPE is used to evaluate the performance of a *target* policy given historical data drawn from (alternative) *behavior* policies. A variety of contemporary OPE methods has been proposed, which can be mainly divided into three categories (Voloshin et al., 2021b): (i) Inverse propensity scoring (Precup, 2000; Doroudi et al., 2017), such as Importance Sampling (IS) (Doroudi et al., 2017), to reweigh the rewards in historical data using the importance ratio between $\beta$ and $\pi$. (ii) Direct methods directly estimate the value functions of the evaluation policy (Nachum et al., 2019; Uehara et al., 2020; Xie et al., 2019; Zhang et al., 2021; Yang et al., 2022), including but not limited to model-based estimators (MB) (Paduraru, 2013; Zhang et al., 2021) that train dynamics and reward models on transitions from the offline data; value-based estimators (Munos et al., 2016; Le et al., 2019) such as Fitted Q Evaluation (FQE) which is a policy evaluation counterpart to batch Q learning; minimax estimators (Liu et al., 2018; Zhang et al., 2020b; Voloshin et al., 2021a) such as DualDICE that estimates the discounted stationary distribution ratios (Yang et al., 2020a). (iii) Hybrid methods combine aspects of both inverse propensity scoring and direct methods (Jiang & Li, 2016; Thomas & Brunskill, 2016). For example, DR (Jiang & Li, 2016) leverages a direct method to decrease the variance of the unbiased estimates produced by IS. However, a major challenge of applying OPE to real world is many methods can perform unpleasant when human-collected data is highly limited as in (Fu et al., 2020; Gao et al., 2023a), augmentation can be an important way to facilitate OPE performance.

**Data Augmentation** Data augmentation has been widely investigated in various domains, including computer vision, time series, and RL. In computer vision, images are the major target and augmentation have improved downstream models' performance (LeCun et al., 1998; Deng et al., 2009; Cubuk et al., 2019; Xie et al., 2020). However, many image-targeted methods, such as crop and rotate images, will discard important information in trajectories. In time series, a variety of data augmentation has been proposed to capture temporal and multivariate dependencies (Le Guennec et al., 2016; Kamycki et al., 2019; Yoon et al., 2019; Iwana & Uchida, 2021a). For instance, SPAWNER (Kamycki et al., 2019) and DGW (Iwana & Uchida, 2021b) augment time series by capturing group-level similarities to facilitate supervised learning. Generative models such as GAN

and VAE have achieved state-of-the-art performance in time series augmentation for both supervised and unsupervised learning (Antoniou et al., 2017; Donahue et al., 2018; Yoon et al., 2019; Barak et al., 2022). However, those approaches for images and time-series do not consider the Markovian nature in OPE training data, and may not be directly applicable to MDP trajectory augmentation.

