# OpenReview forum: "On Trajectory Augmentations for Off-Policy Evaluation"
_ICLR.cc/2024/Conference — ICLR 2024 poster_

### Official Review · Reviewer_pod6 · 2023-10-29

**Soundness:** 2 fair
**Presentation:** 2 fair
**Contribution:** 2 fair
**Rating:** 6
**Confidence:** 4

**Summary:**

The paper is proposed for the Off-policy evaluation tasks in the RL domain. The proposed method OAT intends to solve the scarce and underrepresented offline trajectory challenges by data augmentation. Specifically, a sub-trajectory mining method and fuse process is designed to find potential exploration space and integrate into existing state-action trajectories seamlessly.

**Strengths:**

1. The paper first proposes to augment the HIS trajectory data for offline RL Off-policy evaluation tasks, which is important for the real-world policy evaluation challenges.

2. The process of detecting the potential sub-sequence is explicitly explained and demonstrated, easy to follow and understand, such as from the discrete representations to determine the support value, and eventually identify the PSTs.

3. Authors applied the VAE-MDP process to generate the potential sub-trajectory, and empirical study shows the overall framework OAT is achieving promising results.

**Weaknesses:**

1. The main paper omitted some significant parts and placed them in the appendix, however should be explained in the main context, such as how the latent latent prior is constructed. And how the C different clusters are initially divided when identifying the PST is not clear.

2. Even though the experiment could show that the OPE method performance is improved, the paper is suggested to make a fair analysis of how reliable the reward is from the algorithm augmentation, which, however, is significant for the result value of off-policy methods.

**Questions:**

1. Could the authors discuss if it's only applicable for the behavior data to be augmented? Or can we also augment the target policies?
2. When identifying the PST, at the first step, How are these C different clusters divided? And how is the number of c determined?
3. As shown in Fig3, It is suggested to introduce how the latent prior is constructed, since it is the key step in constructing the new augmented PST.
4. What is the complexity of the training VAE-MDP? Is there any analysis conducted to show the relations between training time and TDSS length/Potential trajectory scale?
5. In Fig 4, it is obvious that the trajectories coverages are different on the left hand and right hand corner on the top of the maze, however if these two corners are not the potential area selected to augment, should they maintain the original distribution to the maximum extent? Apart from this illustration, It is suggested that the paper provides quantitative evaluation of the difference between augmented trajectory and ground truth trajectory.

**Details Of Ethics Concerns:**

Healthcare and e-learning data is used in the paper, it is suggested to make sure it follows the privacy policies.

---

> ### Author Response · Authors · 2023-11-17
> **Authors Responses (1/2)**
>
> Thank you for your time and efforts on evaluating our work. Please find our point-by-point response below.
>
> Q1. The main paper omitted some significant parts and placed them in the appendix, however should be explained in the main context, such as how the latent latent prior is constructed. And how the C different clusters are initially divided when identifying the PST is not clear.
>
> A1. Due to the space limitations, we leave the technical details straightforwardly adapting VAE into MDP in Appendix. But **we moved the details of constructing latent prior to Section 2.2 (highlighted in blue)**. The number of clusters is determined by a fully data-driven procedure following the original TICC paper (i.e., C is determined with the highest silhouette score in clustering historical trajectories) [1].
>
> Q2. Even though the experiment could show that the OPE method performance is improved, the paper is suggested to make a fair analysis of how reliable the reward is from the algorithm augmentation, which, however, is significant for the result value of off-policy methods.
>
> A2. We provided Tables 4-16 in Appendix C.1, presenting the original results, in terms of four different metrics (i.e., mean absolute errors (MAE), rank correlation, regret@1, and regret@5), on each OPE method without and with augmentation in Adroit. Given MAE directly measures the distance between estimated and true returns, we can observe that OAT leads to more accurate return estimates for different OPE methods on target policies. Especially for IS that purely relies on the importance ratio over behavior and target actions and rewards from historical trajectories, OAT effectively facilitates it on different tasks, indicating OAT providing more reliable rewards. Moreover, **we added scatter plots plotting the true returns of each target policy against the estimated returns, in Appendix A.2.1.** We can observe that the estimated rewards of each OPE method were generally improved by OAT, with less distance to true rewards.
>
> Q3. Could the authors discuss if it's only applicable for the behavior data to be augmented? Or can we also augment the target policies?
>
> A3. Our works follow the general OPE problem setup [2], where **only a fixed set of offline data collected from behavioral policy is available**, and the goal is to use such data to estimate the return of target policies (that are different from behavioral policies), without any online deployment of the target policies. Consequently, we would not be able to augment the trajectories collected under target policies, as they would not be available.
>
> Q4. When identifying the PST, at the first step, How are these C different clusters divided? And how is the number of c determined?
>
> A4. The C different clusters are divided by solving the TICC problem [1], by capturing graphical connectivity structure of both temporal and cross-attributes information. **We added those details to Appendix A.3.1 (highlighted in blue)**. Following [1], C is determined by the one with the highest silhouette score clustering historical trajectories within range [10,20].
>
> Q5. As shown in Fig3, It is suggested to introduce how the latent prior is constructed, since it is the key step in constructing the new augmented PST.
>
> A5. Thanks for the suggestion. **We moved the details of constructing latent prior to Section 2.2 (highlighted in blue).**
>
> Q6. What is the complexity of the training VAE-MDP? Is there any analysis conducted to show the relations between training time and TDSS length/Potential trajectory scale?
>
> A6. Given that training VAE-MDP requires stochastic gradient descent algorithms with parameter tuning (e.g., for step size and network architecture), we found that analyzing the complexity theoretically would be challenging, as there lacks a standard framework to facilitate that. Instead we provide additional empirical results here on training time -- specifically, **we plotted relations between training time and length of PSTs/corresponding TDSSs, in Appendix A.1.1.** The results show that the training time of VAE-MDP is increased almost linearly with the length of PSTs.

---

> > ### Author Response · Authors · 2023-11-17
> > **Authors Responses (2/2)**
> >
> > Q7. In Fig 4, it is obvious that the trajectories coverages are different on the left hand and right hand corner on the top of the maze, however if these two corners are not the potential area selected to augment, should they maintain the original distribution to the maximum extent? Apart from this illustration, It is suggested that the paper provides quantitative evaluation of the difference between augmented trajectory and ground truth trajectory.
> >
> > A7. Figure 4 is used as a toy example to illustrate the motivation behind PSTs mining and augmentation. To be more precise, the augmented sub-trajectories are not necessarily temporally-aligned, i.e., the starting time and ending time of found PSTs on each trajectory can be varied as described in Appendix C.3. As a result, a small number of sub-trajectories may cover some parts of the left-top and right-top corners (which makes it particularly obvious when augmenting 10 times numbers of trajectories in Figure 4). Nevertheless, one can still observe that augmented PSTs are effectively improving state-action coverage.
> >
> > Since our experimental environments (e.g., Adroit) are more complex than the Maze2D-umaze environment (state space is roughly $\mathbb{R}^{20}$), it would be very challenging to quantify the distance between augmented and original trajectories. Please note that we had conducted extensive experiments over 34 datasets over multiple application domains, encompassing varied levels of state-action coverage, human involvements, horizons, density of rewards, dimensions of states and actions, etc. Moreover, we used 4 different evaluation metrics to thoroughly examine the performance of 5 OPE methods based on the augmented trajectories.  From the results, one can observe that OAT effectively enhances all types of OPE methods, as in Appendix C.1, illustrating the importance of augmenting trajectories for OPE to better estimate target policies. Moreover, we added scatter plots plotting the true returns of each policy against the estimated returns, in Appendix A.2.1. We can observe that the estimated rewards of each OPE method were generally improved by OAT.
> >
> >
> > We hope these answers provide some explanations to address your concerns and showcase that our work is solving a significant challenge in a satisfying manner. We are happy to answer any followup questions or hear any comments from you.
> >
> >
> >
> > References
> >
> > [1] Hallac, David, et al. "Toeplitz inverse covariance-based clustering of multivariate time series data." KDD 2017.
> >
> > [2] Fu, Justin, et al. "Benchmarks for Deep Off-Policy Evaluation" ICLR 2021.

---

> > > ### Author Response · Authors · 2023-11-19
> > > **Mid-point check-in**
> > >
> > > As we are stepping into the 2nd half of the discussion period, should the reviewer have any follow-ups, we will try out best to address them in time. If satisfied, we would greatly appreciate the reviewer to update the reviews/acknowledge our responses. We sincerely thank the reviewer again for the efforts devoted to the review process, allowing the work to be thoroughly evaluated and discussed.

---

> > > > ### Comment · Reviewer_pod6 · 2023-11-20
> > > > **Thanks for the rebuttal**
> > > >
> > > > I want to thank the authors for addressing my concerns and I have decided to update my review and raise my score.

---

> > > > > ### Author Response · Authors · 2023-11-20
> > > > > **Thank you**
> > > > >
> > > > > Thank you for acknowledging our responses! We sincerely appreciate you spend this amount of time and efforts to help us improve the presentation of the manuscript.

---

> ### Comment · Area_Chair_EGyB · 2023-11-20
>
> Dear pod6,
>
> The author reviewer discussion period is ending soon this Wed. Are you satisfied with the author's clarification w.r.t. your questions or there are still outstanding items that you would like to have more discussions?
>
> Thanks again for your service to the community.
>
> Best,
> AC

---

### Official Review · Reviewer_3xn9 · 2023-10-30

**Soundness:** 2 fair
**Presentation:** 1 poor
**Contribution:** 2 fair
**Rating:** 6
**Confidence:** 4

**Summary:**

The paper proposes to use a specialized form of data-augmentation, specifically to improve the accuracy of off-policy evaluation algorithms. It bases its algorithm on looking at promising sub-trajectories and augmenting the dataset with new samples to improve coverage of the state-action space.

**Strengths:**

1. The paper tackles an important problem of OPE, which receives relatively far less attention than control-based papers.
2. Adapting data augmentation techniques for the RL setting, and particularly the OPE setting, is a very interesting direction.
3. The paper applies the proposed idea to more real-life datasets, which is great since lot of OPE work just evaluates ideas on mujoco.

**Weaknesses:**

1. It feels like the paper was somewhat rushed. There are some confusing parts/writing errors such as: what is "human-involved RLs" (third line in intro), "while human may behave" (5th line last para of intro), Figure 2 I believe should be referencing Sections 2.1/2.2/2.3 (not 3.X), "intrinsic nature that human may follow" (2nd line Section 2.1).
2. The notation in Section 2.1 is very difficult to parse. I suspect there is an easier way to explain this. Also Figure 3 is very confusing, and is not explained in the text (there are some references in the appendix, but I think there should be much more explanation of it given the complexity of the diagram).
3. I think drawing connections to how humans behave (such as in Figure 2 caption) is too strong. It is appropriate for intuition purposes but making general statements on how humans behave seems a bit strong.
4. Biggest concern. I dont think the paper gives a good intuition for why this works. While coverage is important, it seems like that is insufficient. Another challenge is distribution shift. Prior work [1] has discussed that under extreme distribution shift, OPE methods can diverge. However, this work does not discuss this. Moreover, it seems possible to me that this data augmentation technique could introduce samples that worsen the distribution shift, which could worsen accuracy of OPE methods.

[1] Instabilities of Offline RL with Pre-Trained Neural Representation. Wang et al. 2021.

**Questions:**

1. Could one generate better samples by somehow tying in the evaluation policy behavior into the data generation process? It seems like if the goal is to evaluate some policy $\pi_e$, we should account for what $\pi_e$ does/samples actions?
2. How is step 1 in the algorithm actually done? That is, under what basis are states grouped into clusters. Some  representation learning work does this based on behavior similarity [1] etc. How the states are grouped is important for all the remaining steps.
3. The "grounding" process of Eqn 2 is unclear to me. How do you avoid the model from hallucinating and generating samples that cannot occur in the MDP since interaction with the MDP does not happen in the OPE setting?

[1] MICo: Improved representations via sampling-based state similarity for Markov decision processes. Castro et al. 2021.

---

> ### Author Response · Authors · 2023-11-17
> **Authors Response**
>
> Thank you for your time and efforts on evaluating our work. Please find our point-by-point response below.
>
> Q1. There are some confusing parts/writing errors such as: what is "human-involved RLs" (third line in intro), "while human may behave" (5th line last para of intro), Figure 2 I believe should be referencing Sections 2.1/2.2/2.3 (not 3.X), "intrinsic nature that human may follow" (2nd line Section 2.1).
>
> A1. Thanks for pointing out the typos. We have fixed the references on Figure 2 in the manuscript.
> For the mentioned “confusing parts’’, please see our comments point-by-point below:
> * Human-involved RLs generally refer to the context of RL with human involvements, including human-involved decision-making processes, human-guided RL algorithms, etc., which are commonly discussed in related works [1-5].
> * The phrase, “while human may behave”, is discussed under the context that human-involved systems have limited state-action coverage, which can be caused by homogeneous behavior policies (as discussed in 1st paragraph of Introduction); on the other hand, human can behave diversely under different policies [6-7]. Following different policies can result in varied state-action coverage [8-10], which is a fundamental problem in RL.
> * The phrase, “intrinsic nature that human may follow”, is under the context discussing an intrinsic nature/characteristic in human-involved systems, that human could follow homogeneous behavioral policies or specific guidelines when they perform their professions.
>
> We have added more references in our manuscript, for readers' potential interests.
>
> Q2. The notation in Section 2.1 is very difficult to parse. I suspect there is an easier way to explain this. Also Figure 3 is very confusing, and is not explained in the text (there are some references in the appendix, but I think there should be much more explanation of it given the complexity of the diagram).
>
> A2. We understood that the notations/equations may be a bit intense in section 2.1, as we are the first work that introduces to OPE with sub-trajectory augmentation, which is a relatively new framework. As a result, we chose to present our methodology as detailed as possible so readers with different backgrounds can get the idea thoroughly (in case they would like to build on top in the future). We have double checked that there do not exist any sub-/super-scripts or variables that are redundant. **We further noticed that the other reviewer pod6 specifically pointed out that this part is easy to follow, as well as WbKh who pointed out our figures are informative -- we would greatly appreciate it if the reviewer can point out the specific part of methodology that is redundant/hard to follow.**
>
>
> Q3. I think drawing connections to how humans behave (such as in Figure 2 caption) is too strong. It is appropriate for intuition purposes but making general statements on how humans behave seems a bit strong.
>
> A3. When the trajectories are collected from humans, the states and/or actions are highly  related to human behaviors. Similar sub-trajectories may exhibit similar human behaviors. Such findings are reported in related works [21-23].
>
> Q4. I don't think the paper gives a good intuition for why this works.
> (Q4-1) While coverage is important, it seems like that is insufficient.
>
> A4-1. We respectfully disagree with the comment. Many existing works have justified that improving state-action coverage is an important and still open-ended problem [4, 8-10]. **Our work is the first to carry it onto the OPE domain, and attempts to solve it through offline trajectory augmentation.**
>
> (Q4-2) Another challenge is distribution shift. Prior work [1] has discussed that under extreme distribution shift, OPE methods can diverge. However, this work does not discuss this.
>
> A4-2. We agree distribution shift is a challenge for OPE. Although the mentioned work by Wang et al. only considers a very specific type of OPE method under a specific setting (i.e., FQE with linear approximation using pre-trained features from neural networks), DOPE benchmark [10] also found that OPE methods in general have such issues. *This is beyond the scope of this work, as we use existing OPE methods as backbones to process the augmented trajectories.* Hope in the future, the communities will come up with new OPE methods that can resolve the distribution shift in OPE.

---

> > ### Author Response · Authors · 2023-11-17
> > **Authors Response**
> >
> > (Q4-3) Moreover, it seems possible to me that this data augmentation technique could introduce samples that worsen the distribution shift, which could worsen accuracy of OPE methods.
> >
> > A4-3. As discussed in Limitations, latent-model-based models such as VAE have been commonly used for augmentation in offline RL, while they generally rarely come with theoretical error bounds provided, such as Dreamer [12, 13], PlaNet [14], SLAC [15], LatCo [9], Solar [16], etc. Such a challenge also remains for many data augmentation methods [17]. However, once the trajectories are augmented, one can choose to use the downstream OPE methods which come with guarantees, such as DR and DICE. Moreover, please note that our experiments include two real-world environments (e-learning and healthcare) where our method significantly outperforms existing works. And we conducted extensive experiments over 34 datasets with varied characteristics including application domains, state-action coverage, human involvements, horizons, density of rewards, dimensions of states and actions, etc. The experimental results show our method can generally enhance different OPE methods.
> >
> >
> > Q5. Could one generate better samples by somehow tying in the evaluation policy behavior into the data generation process? It seems like if the goal is to evaluate some policy \pi_e, we should account for what \pi_e does/samples actions?
> >
> > A5. In this work, we chose to use behavior data only for augmentation, so that the proposed method can be **stand-alone** to generate trajectories without any assumptions over target policies. And it can be easily utilized by broader built-on-top works such as policy optimization and representation learning. We agree that augmenting target policies is a possible future direction.
> >
> > Q6. How is step 1 in the algorithm actually done? That is, under what basis are states grouped into clusters. Some representation learning work does this based on behavior similarity [1] etc. How the states are grouped is important for all the remaining steps.
> >
> > A6. We appreciate the reviewer pointing us to the representation learning paper. As provided in Appendix A.3, we leverage TICC problem [18] to cluster states, which captures graphical connectivity structure of both temporal and cross-attributes information and has shown . Specifically, in experiments, we utilize its variant, MT-TICC [19], which further considers time-awareness and multi-trajectories based on TICC. **We also provided detailed formulations of the TICC problem in Appendix A.3.1 (highlighted in blue).**
> >
> > Q7. The "grounding" process of Eqn 2 is unclear to me. How do you avoid the model from hallucinating and generating samples that cannot occur in the MDP since interaction with the MDP does not happen in the OPE setting?
> >
> > A7. This is a good question that points out a potential future avenue following our work. We took a similar approach as existing works who also adapted VAEs to learn environmental dynamics [9, 12-16, 20], which cannot resolve such a limitation, but it is definitely worth looking into.
> >
> > We hope these answers provide some explanations to address your concerns and showcase that our work is solving a significant challenge in a satisfying manner. We are happy to answer any followup questions or hear any comments from you.

---

> > > ### Author Response · Authors · 2023-11-17
> > > **Authors Response**
> > >
> > > References
> > >
> > > [1] Wu, Jingda, et al. "Prioritized experience-based reinforcement learning with human guidance for autonomous driving." IEEE Transactions on Neural Networks and Learning Systems (2022).
> > >
> > > [2] Wu, Jingda, et al. "Toward human-in-the-loop AI: Enhancing deep reinforcement learning via real-time human guidance for autonomous driving." Engineering 21 (2023).
> > >
> > > [3] Zhu, Zhuangdi, et al. "Transfer learning in deep reinforcement learning: A survey." IEEE Transactions on Pattern Analysis and Machine Intelligence (2023).
> > >
> > > [4] Gao, Qitong, et al. "Off-Policy Evaluation for Human Feedback." NeurIPS 2023.
> > >
> > > [5] Peng, Zhenghao, et al. "Learning from Active Human Involvement through Proxy Value Propagation." NeurIPS 2023.
> > >
> > > [6] Yang, Xi, et al. "Student Subtyping via EM-Inverse Reinforcement Learning." International Educational Data Mining Society (2020).
> > >
> > > [7] Wang, Lu, et al. "Hierarchical imitation learning via subgoal representation learning for dynamic treatment recommendation." Proceedings of the Fifteenth ACM International Conference on Web Search and Data Mining. 2022.
> > >
> > > [8] Lillicrap, Timothy P., et al. "Continuous control with deep reinforcement learning." ICLR 2016.
> > >
> > > [9] Rybkin, Oleh, et al. "Model-based reinforcement learning via latent-space collocation." ICML 2021.
> > >
> > > [10] Fu, Justin, et al. "Benchmarks for Deep Off-Policy Evaluation." ICLR 2020.
> > >
> > > [11] Fu, Justin, et al. "D4rl: Datasets for deep data-driven reinforcement learning." arXiv preprint arXiv:2004.07219 (2020).
> > >
> > > [12] Hafner, Danijar, et al. "Dream to Control: Learning Behaviors by Latent Imagination." ICLR.
> > >
> > > [13] Hafner, Danijar, et al. "Mastering Atari with Discrete World Models." ICLR.
> > >
> > > [14] Hafner, Danijar, et al. "Learning latent dynamics for planning from pixels." ICML 2019.
> > >
> > > [15] Lee, Alex X., et al. "Stochastic latent actor-critic: Deep reinforcement learning with a latent variable model." NeurIPS 2020.
> > >
> > > [16] Zhang, Marvin, et al. "Solar: Deep structured representations for model-based reinforcement learning." ICML 2019.
> > >
> > > [17] Zheng, Chenyu, et al. "Toward Understanding Generative Data Augmentation." NeurIPS 2023.
> > >
> > > [18] Hallac, David, et al. "Toeplitz inverse covariance-based clustering of multivariate time series data." KDD 2017.
> > >
> > > [19] Yang, Xi. "Multi-series Time-aware Sequence Partitioning for Disease Progression Modeling." IJCAI 2021.
> > >
> > > [20] Gao, Qitong, et al. "Variational Latent Branching Model for Off-Policy Evaluation." ICLR 2022.
> > >
> > >
> > > [21] Gao, Ge, et al. "A reinforcement learning-informed pattern mining framework for multivariate time series classification." IJCAI 2022.
> > >
> > >
> > > [22] Renso, Chiara, et al. "How you move reveals who you are: understanding human behavior by analyzing trajectory data." Knowledge and information systems 37 (2013): 331-362.
> > >
> > >
> > > [23] Higgs, Bryan, and Montasir Abbas. "Segmentation and clustering of car-following behavior: Recognition of driving patterns." IEEE Transactions on Intelligent Transportation Systems 16.1 (2014): 81-90.

---

> > > > ### Comment · Reviewer_3xn9 · 2023-11-18
> > > >
> > > > Thanks to authors for responding. I think offline RL + data augmentation is an important direction that the community should be looking at. I am a bit concerned that the method: 1) does not account for the possibility of worsening distribution shift and 2) may generate samples/transitions that do not comply with the true dynamics of the environment. While their algorithm worked in their empirical setting, I think it may fail in other settings (I am not sure where).
> > > >
> > > > But as the authors point out, they believe these directions are for future work. I am sympathetic to this because I think initial work in data augmentation + offline RL is important, and perhaps others can build on this work to tackle the above problems.
> > > >
> > > > That said, I will raise my score. However, I would insist that the authors: 1) clear all the typos (review the paper again for anything I may have missed) and 2) point the above two points as future work explicitly and explain their thoughts on how their algorithm relates to the two challenges. It will provide a basis to other researchers to know which parts of the algorithm can be improved.

---

> > > > > ### Author Response · Authors · 2023-11-18
> > > > > **Thank you**
> > > > >
> > > > > Thank you for acknowledging our responses, and the positive comments that we investigated an important problem and proposed impactful method. **We have cleared all the typos, and discussed the two potential future works in Conclusion section (highlighted in blue).**

---

### Official Review · Reviewer_WbKh · 2023-11-01

**Soundness:** 3 good
**Presentation:** 2 fair
**Contribution:** 2 fair
**Rating:** 6
**Confidence:** 4

**Summary:**

The authors propose a novel approach to augment offline datasets for off-policy evaluation. This is achieved by introducing a three-step process (i) select relevant sub-trajectories in the dataset, (ii) use a VAE-based architecture to generate new trajectories, and (iii) add these trajectories back to the dataset.

Empirically, the authors show that the proposed method outperforms other data-augmentation methods on a diverse set of problems.

**Strengths:**

I am not an expert in the field of data-augmentation for RL, but I enjoyed the thought process behind the development of the framework: (i) select a criterion for what makes a specific region of the state-action space interesting for data augmentation, and (ii) use temporal generative models to sample new sub-trajectories.

Figures are generally informative and overall the paper is well written.

**Weaknesses:**

In my opinion, the main weaknesses of this work lie in (i) the unsupported justifications of the results, (ii) the lack of ablations to validate the proposed innovations, and (iii) the relatively narrow-scoped experiments.

**Questions:**

(i) The unsupported justifications of the results
In more than one occasion, the authors (rightly) discuss very specific reasons that could confirm/justify the observed results. However, I feel like in most cases in this work, the justifications are not supported by data. I feel this is better explained through an example. The authors say: "In contrast, the historical trajectories induced from simulations tend to result in better coverage over the state-action space in general, and the augmentation methods that do not consider the Markovian setting may generate trajectories that could be less meaningful to the OPE methods, making them less effective." What do the authors mean by less meaningful? And do they believe that this (i.e., non-Markovian augmentation are worse in scenarios with better state coverage) can be confirmed more generally?

In my opinion, the authors use phrases like "we conjecture", "a possible reason" without strictly backing up the claims with evidence (which would in turn greatly improve the quality of the paper)

(ii) The lack of ablations to validate the proposed innovations:
The authors compare against a wide set of benchmarks, although, as I'm not an expert in OPE, it is unclear to me whether these are explicitly tailored for the OPE problem or not. Moreover, since the proposed framework is a composed of multiple smaller contributions (discretization, VAE architecture, etc.), the authors should make sure to isolate each of these contributions individually and support the claims with evidence and experiments.

(iii) The relatively narrow-scoped experiments:
Did the authors consider using this approach within an RL context? How would this perform?

---

> ### Author Response · Authors · 2023-11-17
> **Authors Responses**
>
> We sincerely appreciate your time and efforts on evaluating our work. Please find our point-by-point response below.
>
> Q1. The unsupported justifications of the results.
>
> A1. **We added related citations and detailed evidence to justify our findings and statements in the revised manuscript (highlighted in blue).**
>
> Q2. The lack of ablations to validate the proposed innovations:
> (Q2-1)The authors compare against a wide set of benchmarks, although, as I'm not an expert in OPE, it is unclear to me whether these are explicitly tailored for the OPE problem or not.
>
> A2-1. To the best of our knowledge, no prior work systematically investigates trajectory augmentation for facilitating OPE, our work is the first to investigate it and provide a possible solution (i.e., OAT). Thus, we examine a broad range of possible data augmentation methods, which were original proposed towards *data that may sharing some characteristics to offline trajectories*, or *sharing some intuitions with OAT*, including: 4 RL-oriented (TDA, Permutation, jittering, scaling), 2 generative models (TimeGAN, VAE), 2 time series-oriented methods by capturing similar behaviors (SPAWNER, DGW). Moreover, we investigated 5 popular OPE methods that are broadly used in prior works [1-11], spanning 3 major categories of OPE as defined by Yue’s group [11].
>
> (Q2-2) Moreover, since the proposed framework is a composed of multiple smaller contributions (discretization, VAE architecture, etc.), the authors should make sure to isolate each of these contributions individually and support the claims with evidence and experiments.
>
> A2-2. Given the key idea is to mine potential sub-trajectories and considering Markovian nature on trajectories, we compared OAT to several ablations:
> * OAT w.o. Discretization & support determination (TDA)
> * OAT w.o. PSTs mining (VAE-MDP)
> * OAT w.o. MDP (VAE)
>
> Note that both discretization and support determination are two common successive techniques to extract shared high-level representations (e.g., PSTs) from high-dimensional complex data [14-16]. Thus, we ablated the proposed PSTs mining from two perspectives: ablating the support determination to identify PSTs from TDSSs by randomly selecting sub-trajectories to augment (which aligns with the original idea of TDA [17]); ablating the concept of PSTs by applying VAE-MDP on entire trajectories (VAE-MDP). The third ablation is applying VAE on entire trajectories, without adaptation to Markovian setting.
>
> Q3. The relatively narrow-scoped experiments. Did the authors consider using this approach within an RL context? How would this perform?
>
> A3. In this work, we specifically focus on the OPE problem [1-10] as the improvements resulting from the model we designed can be isolated, as opposed to policy optimization where the policy improvement step will also impact the performance. We followed the guidelines and standardized procedures introduced in a recent benchmark, DOPE [1], from Levine's group, which provided for each D4RL environment the target policies to be evaluated as well as an off-policy training dataset. The effectiveness and robustness of OAT was extensively validated over **34 datasets with varied characteristics** including applications, state-action coverage, human involvements, horizons, density of rewards, dimensions of states and actions, etc. The experiments contained 2 real-world applications, education and healthcare. Moreover, OAT can be **stand-alone** to generate trajectories without any assumptions over target policies. And it can be easily utilized by built-on-top works such as policy optimization and representation learning. **We added such discussions in the conclusion section -- highlighted in blue.**
>
> We hope these answers provide some explanations to address your concerns and showcase that our work is solving a significant challenge in a satisfying manner. We are happy to answer any followup questions or hear any comments from you.

---

> > ### Author Response · Authors · 2023-11-17
> > **Authors Responses**
> >
> > References
> >
> >
> > [1] Fu, Justin et al. "Benchmarks for Deep Off-Policy Evaluation." ICLR 2021.
> >
> >
> > [2] Zhang, Ruiyi et al. "GenDICE: Generalized Offline Estimation of Stationary Values." ICLR 2020.
> >
> >
> > [3] Zhang, Shangtong et al. "Gradientdice: Rethinking generalized offline estimation of stationary values." ICLR 2020.
> >
> >
> > [4] Zhang, Michael R. et al. "Autoregressive Dynamics Models for Offline Policy Evaluation and Optimization." ICLR 2021.
> >
> >
> > [5] Tang, Ziyang et al. "Doubly Robust Bias Reduction in Infinite Horizon Off-Policy Estimation." ICLR 2020.
> >
> >
> > [6] Yang, Mengjiao et al. "Off-policy evaluation via the regularized lagrangian." ICML 2020.
> >
> >
> > [7] Nachum, Ofir et al. "Dualdice: Behavior-agnostic estimation of discounted stationary distribution corrections." NeurIPS 2019.
> >
> >
> > [8] Wen, Junfeng et al. "Batch Stationary Distribution Estimation." ICML 2020.
> >
> >
> > [9] Dai, Bo et al. "Coindice: Off-policy confidence interval estimation." NeurIPS 2020.
> >
> >
> > [10] Kostrikov, Ilya, and Ofir Nachum. "Statistical bootstrapping for uncertainty estimation in off-policy evaluation." arXiv preprint arXiv:2007.13609 (2020).
> >
> >
> > [11] Voloshin, Cameron et al. "Empirical Study of Off-Policy Policy Evaluation for Reinforcement Learning." NeurIPS 2021.
> >
> >
> > [12] Hafner, Danijar et al. "Dream to Control: Learning Behaviors by Latent Imagination." ICLR 2020
> >
> >
> > [13] Lee, A. X. et al. Stochastic latent actor-critic: Deep reinforcement learning with a latent variable model. NeurIPS 2020.
> >
> >
> > [14] Gao, Ge, et al. "A reinforcement learning-informed pattern mining framework for multivariate time series classification." IJCAI 2022.
> >
> >
> > [15] Schäfer, Patrick, and Ulf Leser. "Fast and accurate time series classification with weasel." CIKM 2017.
> >
> >
> > [16] Chang, Joong Hyuk, and Won Suk Lee. "Efficient mining method for retrieving sequential patterns over online data streams." Journal of Information Science 2005.
> >
> >
> > [17] Park, Jongjin, et al. "SURF: Semi-supervised Reward Learning with Data Augmentation for Feedback-efficient Preference-based Reinforcement Learning." ICLR 2021.

---

> > > ### Author Response · Authors · 2023-11-19
> > > **Mid-point check-in**
> > >
> > > As we are stepping into the 2nd half of the discussion period, should the reviewer have any follow-ups, we will try out best to address them in time. If satisfied, we would greatly appreciate the reviewer to update the reviews/acknowledge our responses. We sincerely thank the reviewer again for the efforts devoted to the review process, allowing the work to be thoroughly evaluated and discussed.

---

> > > > ### Comment · Area_Chair_EGyB · 2023-11-20
> > > >
> > > > Dear WbKh,
> > > >
> > > > The author reviewer discussion period is ending soon this Wed. Does the author response clear your concerns w.r.t., e.g., the empirical results or there are still outstanding items that you would like more discussion?
> > > >
> > > > Thanks again for your service to the community.
> > > >
> > > > Best,
> > > > AC

---

> > > > > ### Comment · Reviewer_WbKh · 2023-11-20
> > > > > **Feedback on author response**
> > > > >
> > > > > I want to thank the authors for their efforts in addressing my concerns. Based on their answers, I have decided to update my review and raise my score.

---

> > > > > > ### Author Response · Authors · 2023-11-20
> > > > > > **Thank you**
> > > > > >
> > > > > > Thank you for acknowledging our responses and spending this amount of time and efforts to help us improve the presentation of the manuscript.

---

### Meta-Review · Area_Chair_EGyB · 2023-12-05

**Metareview:**

This work develops a method to augment trajectories in offline data from human involved systems to improve offline policy evaluation algorithms. Extensive empirical study are conducted in 34 different datasets and clear performance improvements are observed.

Strengths: Despite that data augmentation in RL is not new, trajectory level data augmentation in offline RL appears novel. The empirical study in this work is extensive, including many real world datasets. Various data augmentation techniques are tested and the performance improvement of the proposed method is significant, across different representative OPE algorithms.

Weakness: This work lacks a mathematical formulation of the problem. It would be nice to have a mathematical characterization of the data from human involved systems. Then this work could benefit from a theoretical analysis demonstrating how the proposed trajectory augmentation techniques interact with the data.

**Justification For Why Not Higher Score:**

Lack of theoretical analysis.

**Justification For Why Not Lower Score:**

The extensive empirical study clearly supports the efficacy of the novel trajectory augmentation idea.

---

### Decision · Program_Chairs · 2024-01-16

Accept (poster)